# Neural Graph Reasoning: A Survey on Complex Logical Query Answering

**Hongyu Ren**\* 1                                                   *hyren@cs.stanford.edu*

**Mikhail Galkin**\* 2                                            *mikhail.galkin@intel.com*

**Zhaocheng Zhu**³                                          *zhaocheng.zhu@umontreal.ca*

**Jure Leskovec**¹                                                    *jure@cs.stanford.edu*

**Michael Cochez**⁴                                                      *m.cochez@vu.nl*

\* **Equal contribution**
¹ *Stanford University* ² *Intel AI Lab* ³ *Mila - Québec AI Institute and Université de Montréal* ⁴ *Vrije Universiteit Amsterdam and Elsevier discovery lab, Amsterdam, the Netherlands*

**Reviewed on OpenReview:** *https://openreview.net/forum?id=xG8un9ZbqT*

## Abstract

Complex logical query answering (CLQA) is a recently emerged task of graph machine learning that goes beyond simple one-hop link prediction and solves the far more complex task of multi-hop logical reasoning over massive, potentially incomplete graphs. The task received significant traction in the community; numerous works expanded the field along theoretical and practical axes to tackle different types of complex queries and graph modalities with efficient systems. In this paper, we provide a holistic survey of CLQA with a detailed taxonomy studying the field from multiple angles, including graph types (modality, reasoning domain, background semantics), modeling aspects (encoder, processor, decoder), supported queries (operators, patterns, projected variables), datasets, evaluation metrics, and applications. Finally, we point out promising directions, unsolved problems and applications of CLQA for future research.

## Contents

## 1   Introduction

Graph databases (graph DBs) are key architectures to capture, organize and navigate structured relational information over real-world entities. Unlike traditional relational DBs storing information in tables with a rigid schema, graph DBs store information in the form of heterogeneous graphs, where nodes represent entities and edges represent relationships between entities. In graph DBs, a relation (i.e. heterogeneous connection between entities) is a first-class citizen. With the graph structure and a more flexible schema, graph DBs allow for an efficient and expressive way to handle higher-order relationships between distant

Figure 1: A complex logical query **a** and its execution over an incomplete graph **b**. Symbolic engines like SPARQL **c** perform edge traversal and retrieve an incomplete set of *easy* answers directly reachable in the graph, i.e., {`UofT`}. Neural query execution **d** recovers missing ground truth edges (dashed) and returns an additional set of *hard* answers {`UdeM`, `NYU`} unattainable by symbolic methods.

entities, especially navigating through multi-hop hierarchies. While traditional DBs require expensive join operations to retrieve information, graph DBs can directly traverse the graph and navigate through links more efficiently. Due to its capabilities, graph databases serve as the backbone of many critical industrial applications including question answering in virtual assistants (Flint, 2021; Ilyas et al., 2022), recommender systems in marketplaces (Dong, 2018; Hamad et al., 2018), social networking in mobile applications (Bronson et al., 2013), and fraud detection in financial industries (Tian et al., 2019; Pourhabibi et al., 2020).

One of the most important tasks of graph DBs is to perform complex query answering. The goal is to retrieve the answers of a given input query from the graph database. Given the query, graph DBs first translate and optimize the query into a more efficient graph traversal pattern with a query planner, and then execute the pattern on the graph database to retrieve the answers from the graph storage using the query executor. The storage compresses the graphs into symbolic indexes suitable for fast table lookups. Querying is thus fast and efficient under the assumption of completeness, i.e., stored graphs have no missing edges.

However, most real-world graphs are notoriously incomplete, e.g., in Freebase, 93.8% of people have no place of birth and 78.5% have no nationality (Mintz et al., 2009), and about 68% of people do not have any profession (West et al., 2014), while in Wikidata, about 50% of artists have no date of birth (Zhang et al., 2022a), and only 0.4% of known buildings have information about height (Ho et al., 2022). In light of incompleteness, naïvely traversing the graph to find answers leads to a significant miss of relevant results, and the issue further exacerbates with an increasing query complexity. This inherently hinders the application of graph databases. Link prediction aims to predict missing information, but is a challenging task. Prior works predict links by learning a latent representation of entities or links (Bordes et al., 2013; Yang et al., 2015; Trouillon et al., 2016; Sun et al., 2019) or mining rules (Galárraga et al., 2013; Xiong et al., 2017; Lin et al., 2018; Qu et al., 2021). While it is possible to use one-hop link predictors to materialize all predicted facts (above certain confidence threshold) and run deterministic query answering pipelines, the computational complexity of this operation is quadratic in the number of entities and is prohibitively expensive for any real-world graph. Furthermore, these approaches rank possible candidates for completion, meaning that they do not tell which of the completions could be traversed. Also reasoning can be used to complete specific information, but there is always a trade-off between possibly incomplete results and decidability – with a

denser graph, some SPARQL entailment regimes (Hawke et al., 2013) do not guarantee that query execution terminates in finite time.

On the other hand, recent advances in graph machine learning enabled expressive reasoning over large graphs in a latent space without facing decidability bottlenecks. The seminal work of Hamilton et al. (2018) on Graph Query Embedding (GQE) laid foundations of answering complex, database-like logical queries over incomplete KGs where inferring missing links during query execution is achieved via parameterization of entities, relations, and logical operators with learnable vector representations and neural networks. For the incomplete knowledge graph in (Fig. 1), given a complex query *"At what universities do the Turing Award winners in the field of Deep Learning work?"*, traditional symbolic graph DBs (SPARQL- or Cypher-like) would return only one answer (`UofT`), reachable by edge traversal. In contrast, neural query embedding parameterizes the graph and the query with learnable vectors in the embedding space. Neural query execution is akin to *traversing the graph and executing logical operators in the embedding space* that infers missing links and enriches the answer set with two more relevant answers, `UdeM` and `NYU`, unattainable by symbolic DBs.

Since then, the area has seen a surge of interest with numerous improvements of supported logical operators, query types, graph modalities, and modeling approaches. In our view, those improvements have been rather scattered, without an overall aim. There still lacks a unifying framework to organize the existing works and guide future research. To this end, we present one of the first holistic studies about the field. Conceptually, we devise the taxonomy of CLQA methods that includes various aspects of query answering and is supposed to be a bridge between data management and ML communities. The taxonomy (Section 3) classifies existing works along three main axes, *i.e.*, (i) **Graphs** (Section 4 – logical formalisms behind the underlying graph and its schema) (ii) **Modeling** (Section 5 – what are the neural approaches to answer queries) (iii) **Queries** (Section 6 – what queries can be answered). We then discuss **Datasets and Metrics** (Section 7 – how we measure the performance of query answering). Each of these dimensions is further divided into fine-grained aspects. Finally, we discuss CLQA applications (Section 8) and summarize open challenges for future research (Section 9).

**Related Work.** While there exist insightful surveys on general graph machine learning (Chami et al., 2022), simple link prediction in KGs (Ali et al., 2021; Chen et al., 2023), and logic-based link prediction (Zhang et al., 2022b; Delong et al., 2024), the complex query answering area remained uncovered so far. With our work, we close this gap and provide a holistic view on the state of affairs in this emerging field. We also elaborate on the similarities and differences between CLQA and KG-based Question Answering (KGQA) (a different subfield of NLP) in Appendix B.

## 2 Preliminaries

Here, we discuss the foundational terms and preliminaries often seen in the database and graph learning literature. The rest of the preliminaries is deferred to the Appendix A elaborating on the mapping to structured languages like SPARQL, formalizing approximate graph query answering, logical operators, and fuzzy logic. The full hierarchy of definitions is presented in Fig. 3.

### 2.1 Types of Graphs

Here we introduce different types of graphs relevant for this research area. Our definitions are adaptations from those in (Hogan et al., 2021, ch. 2). We begin by defining a set with all elements used in our graph

**Definition 2.1 (`Con`)** *`Con` is an infinite set of constants.*[1]

**Definition 2.2 ((Standard / Triple based) Knowledge Graph)** *We define a knowledge graph (KG) $\mathcal{G} = (\mathcal{E}, \mathcal{R}, \mathcal{S})$, where $\mathcal{E} \subset$ `Con` represents a set of nodes (entities), $\mathcal{R} \subset$ `Con` is a set of relations (edge types), and $\mathcal{S} \subset (\mathcal{E} \times \mathcal{R} \times \mathcal{E})$ is a set of edges (triples)[2]. Each edge $s \in \mathcal{S}$ on a KG $\mathcal{G}$ denotes a statement (or fact) $(e_s, r, e_o)$ in a triple format or $r(e_s, e_o)$ as a formula in first-order logic (FOL) form, where $e_s, e_o \in \mathcal{E}$ denote*

---

[1]In case the constants do not include representations for the real numbers, this set would be countably infinite.

[2]There might be an overlap between $\mathcal{E}$ and $\mathcal{R}$, i.e., some constants might be used as an edge and as a node.

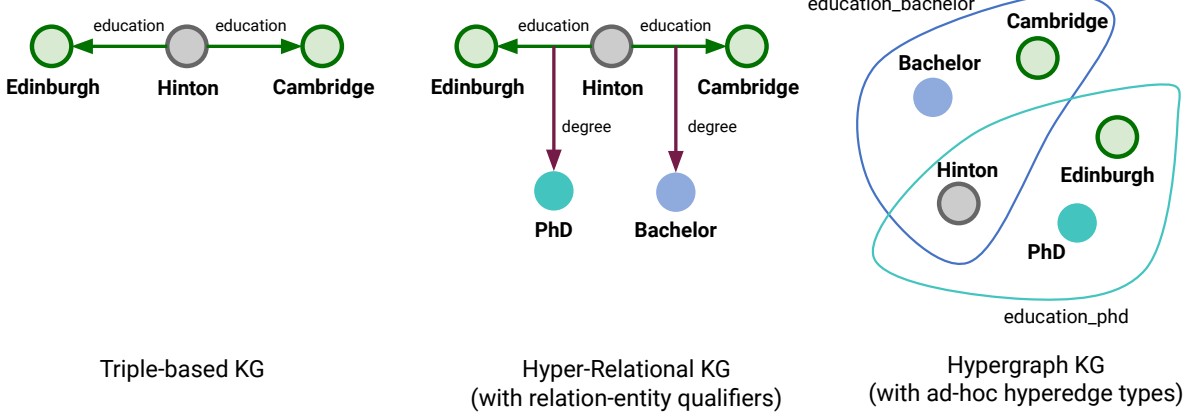

Figure 2: An example of KG modalities. Triple-only graphs have directed binary edges between nodes. Hyper-relational graphs allow key-value relation-entity attributes over edges. Hypergraphs consist of hyperedges composed of multiple nodes.

*the subject and object entities, and $r \in \mathcal{R}$ denotes the relationship between the subject and the object. Often, the graph is restricted such that the set of nodes and edges must be finite, we will make this assumption unless indicated otherwise.*

For example, one statement on the KG in Fig. 1 **b** is `(Hinton, university, UofT)` or `university(Hinton, UofT)`. Note all relations are binary on KGs, *i.e.*, each relation involves exactly two entities. This is also the choice made for the resource description framework (RDF) (Brickley et al., 2014) that defines a triple as a basic representation of a fact, yet RDF allows a broader definition of triples, where the object can be literals including numbers and timestamps (detailed in the following paragraphs).

We elaborate on hyper-relational KGs and hypergraph KGs in Appendix A.1.

## 2.2 Basic Graph Query Answering

We base our definition of basic graph queries on the one from (Hogan et al., 2021, section 2.2.1 basic graph patterns), but adapt it to our graph formalization. Other definitions can be found in Appendix A.2.

**Definition 2.3 (Term and Var)** *Given a knowledge graph $\mathcal{G} = (\mathcal{E}, \mathcal{R}, \mathcal{S})$ (or equivalent $\mathcal{G} = (\mathcal{E}, \mathcal{R}, \mathcal{S}, \mathcal{L})$), we define the set of variables $\mathtt{Var} = \{v_1, v_2, \dots\}$ which take values from $\mathtt{Con}$, but is strictly disjoint from it. We call the union of the variables and the constants the terms: $\mathtt{Term} = \mathtt{Con} \cup \mathtt{Var}$*

With these concepts in place a Basic Graph Query is defined as follows:

**Definition 2.4 (Basic Graph Query)** *A basic graph query is a 4-tuple $\mathcal{Q} = (\mathcal{E}', \mathcal{R}', \mathcal{S}', \overline{\mathcal{S}'})$ (equivalently a 5-tuple $\mathcal{Q} = (\mathcal{E}', \mathcal{R}', \mathcal{S}', \overline{\mathcal{S}'}, \mathcal{L}')$, with literals), with $\mathcal{E}' \subset \mathtt{Term}$ a set of node terms, $\mathcal{R}' \subset \mathtt{Term}$ a set of relation terms, and $\mathcal{S}', \overline{\mathcal{S}'} \subset \mathcal{E}' \times \mathcal{R}' \times \mathcal{E}'$, two sets of edges (or equivalent to how graph edges are defined with literals). The query looks like two (small) graphs; one formed by the edges in $\mathcal{S}'$, and another by the edges in $\overline{\mathcal{S}'}$. The former set includes edges that must be matched in the graph to obtain answers to the query, while the latter contains edges that **must not** be matched (i.e., atomic negation).*

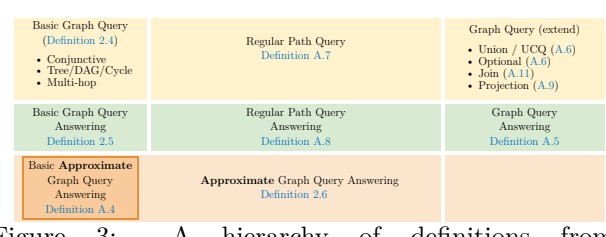

Figure 3: A hierarchy of definitions from Graph Query to Graph Query Answering and more general Approximate Graph Query Answering .

With $\mathtt{Var}_Q = (\mathcal{E}' \cup \mathcal{R}') \cap \mathtt{Var}$ (all variables in the query), the answer to the query is defined as follows.

**Definition 2.5 (Basic Graph Query Answering)** *Given a knowledge graph $\mathcal{G}$ and a query $\mathcal{Q}$ as defined above, an answer to the query is any mapping $\mu : \mathtt{Var}_Q \to \mathtt{Con}$ such that replacing each variable $v_i$ in the edges in $\mathcal{S}'$ with $\mu(v_i)$ results in an edge in $\mathcal{S}$, and replacing each variable $v_j$ in the edges in $\overline{\mathcal{S}'}$ with $\mu(v_j)$ results in an edge that is **not** in $\mathcal{S}$. The set of all answers to the query is the set of all such mappings.*

For triple-based KGs, each edge $(a, R, b)$ in the edge set $\mathcal{S}'$ (respectively $\overline{\mathcal{S}'}$) of a query can be seen as a relation projection $R(a, b)$ (resp. $\neg R(a, b)$), *i.e.*, as binary functions. Now, because the conjunction (*i.e.*, all) of these relation projections (resp. the negation) have to be true, we also call these *conjunctive* queries with atomic negation (CQ$_{\mathrm{neg}}$). The SPARQL query language defines basic graph patterns (BGPs). These closely resemble our basic graph query for RDF graphs, but with $\overline{\mathcal{S}'} = \emptyset$, *i.e.*, they do not support atomic negation but only conjunctive queries (CQ). We could analogously create CQ and CQ$_{\mathrm{neg}}$ classes for the other KG types.

In Fig. 1 **c** we provide an example of a Basic Graph Query expressed in the form of a SPARQL query. This corresponds to our formalism:

$$\mathcal{Q} = (\{\mathtt{TuringAward}, \mathit{?person}, \mathtt{DeepLearning}, \mathit{?uni}\}, \qquad\qquad (\texttt{\# Node Terms } \mathcal{E}')$$
$$\{\mathit{win}, \mathit{field}, \mathit{university}\}, \qquad\qquad (\texttt{\# Relation Terms } \mathcal{R}')$$
$$\{(\mathtt{TuringAward}, \mathit{win}, \mathit{?person}), (\mathtt{DeepLearning}, \mathit{field}, \mathit{?person}), (\mathit{?person}, \mathit{university}, \mathit{?uni})\}, \quad (\mathcal{S}')$$
$$\emptyset) \qquad\qquad (\overline{\mathcal{S}'})$$

A possible answer to this query is the following partial mapping: $\mu_1 = \{(\mathit{?person}, \mathtt{Hinton}), (\mathit{?uni}, \mathtt{UofT})\}$ This is also the only answer, so the set of all answers is $\{\mu_1\}$

If we want to exclude people who have worked together with $\mathtt{Welling}$, then we modify the query as follows:

$$\mathcal{Q} = (\{\mathtt{TuringAward}, \mathit{?person}, \mathtt{DeepLearning}, \mathit{?uni}\},$$
$$\{\mathit{win}, \mathit{field}, \mathit{university}\},$$
$$\{(\mathtt{TuringAward}, \mathit{win}, \mathit{?person}), (\mathtt{DeepLearning}, \mathit{field}, \mathit{?person}), (\mathit{?person}, \mathit{university}, \mathit{?uni})\},$$
$$\mathbf{\{(\mathit{?person}, \mathit{collab}, \mathtt{Welling})\}})$$

The key difference here is that we add $\{(\mathit{?person}, \mathit{collab}, \mathtt{Welling})\}$ to $\overline{\mathcal{S}'}$, resulting in the empty answer set.

## 2.3 Approximate Graph Query Answering

**Definition 2.6 (Approximate Graph Query Answering)** *Given a knowledge graph $\mathcal{G}$, subgraph of a complete, but not observable knowledge graph $\hat{\mathcal{G}}$, a query formalism, **any** graph query $\mathcal{Q}$ according to that formalism, and the scoring domain $\mathbb{R}$.*

*An approximate graph query answer to the query $\mathcal{Q}$ is a function $f$ which maps every possible mapping $(\mu : \mathtt{Var}_Q \to \mathtt{Con})$ to $\mathbb{R}$.*

Note there that the variables are not always mapped to nodes which occur in the graph. It is well possible that the query contains an aggregation function which results in a literal value.

In the neural logical query answering literature (starting from Hamilton et al. (2018); Ren et al. (2020)), we can find the concepts of *easy* and *hard* answers. This refers to whether the answers can be found by only having access to $\mathcal{G}$ or not.

**Definition 2.7 (Easy and Hard answers)** *Given a knowledge graph $\mathcal{G}$, subgraph of a larger unobservable graph $\hat{\mathcal{G}}$, and a query $\mathcal{Q}$. Easy and hard answers are defined in terms of exact query answering (Definition A.5). The set of **easy** answers is the intersection of the answers obtained from $\mathcal{G}$, and those from $\hat{\mathcal{G}}$. The set of **hard** answers is the set difference between the answers from $\hat{\mathcal{G}}$ and those from $\mathcal{G}$.*

Note the asymmetry in the definitions. Easy answers are those that can be found in both $\mathcal{G}$ and $\hat{\mathcal{G}}$. Hard answers are those that can be found only in $\hat{\mathcal{G}}$ but *not* in $\mathcal{G}$. For example, the query in Fig. 1 has one *easy* answer UofT as it is reachable by traversing the original graph $\mathcal{G}$ and two *hard* answers UdeM, NYU since they require predicting missing links (missing in $\mathcal{G}$ but true links in the non-observable full graph $\hat{\mathcal{G}}$). For Basic Graph Queries (Definition 2.4), all easy answers can also be found from $\hat{\mathcal{G}}$. However, for some more complex query types (*e.g.*, these which allow negation) there could be answers found in $\mathcal{G}$ which are not found in $\hat{\mathcal{G}}$. We call these answers **false positives** in the context of answering over $\hat{\mathcal{G}}$.

## 3 A Taxonomy of Query Reasoning Methods

In the following sections, we devise a taxonomy of query answering works. We categorize existing and envisioned approaches along three main directions: (i) **Graphs** – what is the underlying structure against which we answer queries; (ii) **Modeling** – how we answer queries and which inductive biases are employed; (iii) **Queries** – what we answer, what are the query structures and what are the expected answers. The taxonomy is presented in Fig. 4. In the following sections, we describe each direction in detail and illustrate them with examples covering the currently existing CLQA literature (more than 50 papers).

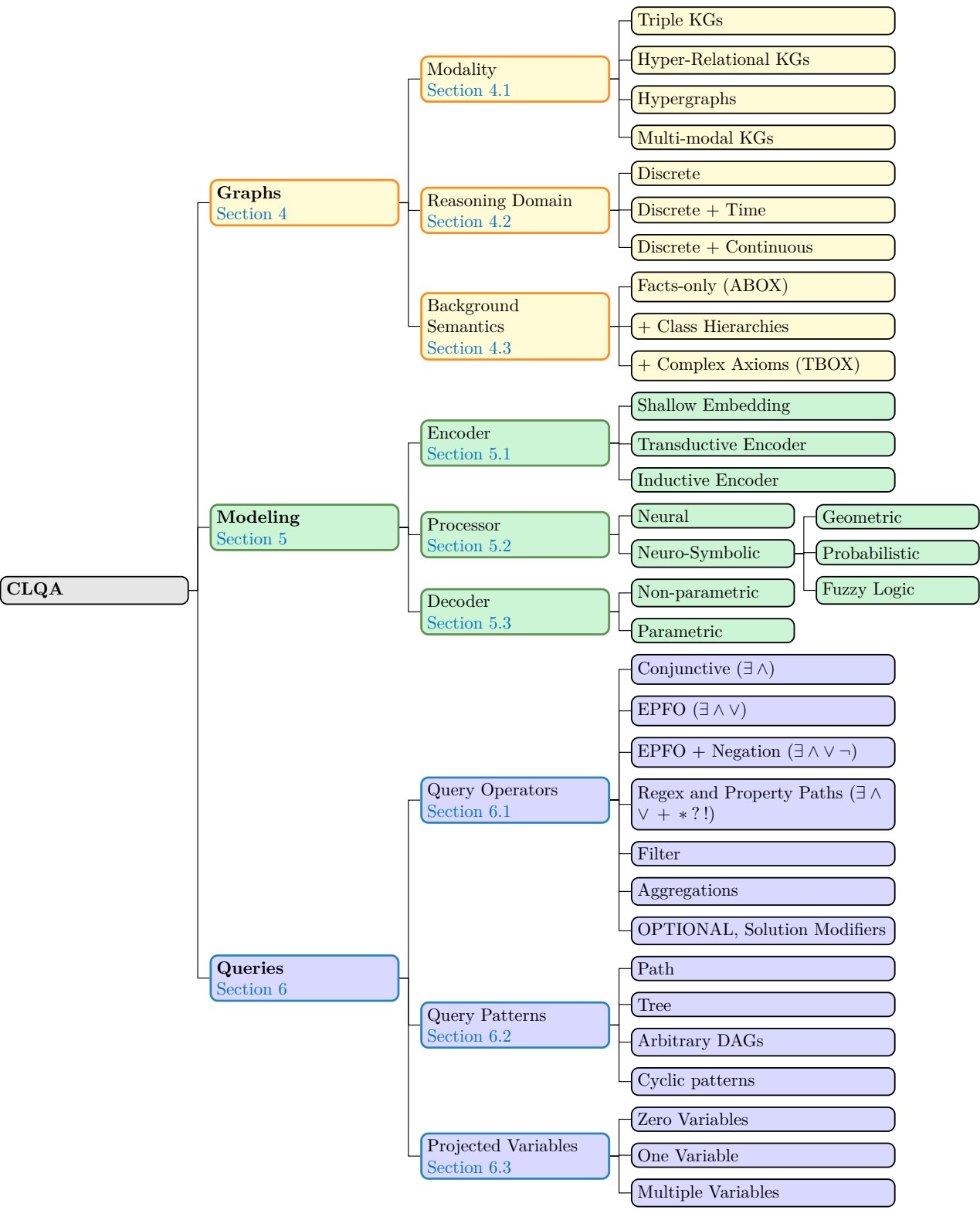

Figure 4: The Neural Query Engine Taxonomy consists of three main branches – Graphs, Modeling, and Queries. We describe each branch in more detail including prominent examples in the relevant sections.

## 4 Graphs

The *Graphs* category covers the underlying graph structure ($\mathcal{G}$ in Definition A.4) against which complex queries are sent and the answers are produced. Understanding the graph, its contents and modeling paradigms is crucial for designing query answering models. To this end, we propose to analyze the underlying graph from three aspects: *Modality*, *Reasoning Domain*, and *Background Semantics*.

### 4.1 Modality

We highlight four modalities common for KGs and graph databases: standard **triple-based KGs**, **hyper-relational KGs**, and **hypergraph KGs**. The difference among the three modalities is illustrated in Fig. 2. Additionally, we outline **multi-modal KGs** that contain not just a graph of nodes and edges, but also text, images, audio, video, and other data formats linked to the underlying graph explicitly or implicitly.

We categorize the literature along the Modality aspect in Table 1. To date, most query answering approaches operate solely on **triple-based** graphs. Among approaches supporting **hyper-relational** graphs, we are only aware of StarQE (Alivanistos et al., 2022) that incorporates entity-relation *qualifiers* over labeled edges and its extension NQE (Luo et al., 2023). We posit that the hyper-relational model might serve as a theoretical foundation of temporal query answering approaches since temporal attributes are in fact continuous key-value edge attributes. To date, we are not aware of complex query answering models supporting **hypergraphs** or **multi-modal** graphs. We foresee them as possible area of future research in the area.

Table 1: Complex Query Answering approaches categorized under *Modality*.

| Triple-only | Hyper-Relational | Hypergraph | Multi-modal |
|---|---|---|---|
| GQE (Hamilton et al., 2018), GQE w hash (Wang et al., 2019), CGA (Mai et al., 2019), TractOR (Friedman & Van den Broeck, 2020), Query2Box (Ren et al., 2020), BetaE (Ren & Leskovec, 2020), EmQL (Sun et al., 2020), MPQE (Daza & Cochez, 2020), Shv (Gebhart et al., 2023), Q2B Onto (Andresel et al., 2021), RotatE-Box (Adlakha et al., 2021), BiQE (Kotnis et al., 2021), HyPE (Choudhary et al., 2021b), NewLook (Liu et al., 2021), CQD (Arakelyan et al., 2021), PERM (Choudhary et al., 2021a), ConE (Zhang et al., 2021b), LogicE (Luus et al., 2021), MLPMix (Amayuelas et al., 2022), FuzzQE (Chen et al., 2022), GNN-QE (Zhu et al., 2022), GNNQ (Pflueger et al., 2022), SMORE (Ren et al., 2022), KGTrans (Liu et al., 2022), LinE (Huang et al., 2022b), Query2Particles (Bai et al., 2022), TAR (Tang et al., 2022), TeMP (Hu et al., 2022), FLEX (Lin et al., 2022), TFLEX (Lin et al., 2023), NodePiece-QE (Galkin et al., 2022b), ENeSy (Xu et al., 2022), GammaE (Yang et al., 2022a), NMP-QEM (Long et al., 2022), QTO (Bai et al., 2023c), SignalE (Wang et al., 2022), LMPNN (Wang et al., 2023e), Var2Vec (Wang et al., 2023a), CQD$^{\mathcal{A}}$ (Arakelyan et al., 2023), Query2Geom (Sardina et al., 2023), SQE (Bai et al., 2023b), RoConE (He et al., 2023), FIT (Yin et al., 2023b), LitCQD (Demir et al., 2023), CylE (Nguyen et al., 2023b), LARK (Choudhary & Reddy, 2023), WFRE (Wang et al., 2023d), BiDAG (Xu et al., 2023a), NRN (Bai et al., 2023a), Query2Triple (Xu et al., 2023b), SCoNe (Nguyen et al., 2023a), CQD Onto (Andresel et al., 2023), UnRavL (Cucumides et al., 2024), UltraQuery (Galkin et al., 2024) | StarQE (Alivanistos et al., 2022), NQE (Luo et al., 2023) | None | None |

### 4.2 Reasoning Domain

Following Definition 2.5, a query $\mathcal{Q}$ includes constants Con, variables Var, and returns *answers* as mappings $\mu : \text{Var}_Q \to \text{Con}$. By *Reasoning Domain* we understand the space of possible constants that query answering models can reason about. We highlight three common domains (*Discrete*, *Discrete + Time*, *Discrete + Continuous*), illustrate them in Fig. 5, and categorize existing works in Table 2. Each subsequent domain is a superset of the previous domains, *e.g.*, *Discrete + Continuous* includes the capabilities of *Discrete* and *Discrete + Time* and expands the space to continuous inputs and outputs.

In the **Discrete** domain, constants, variables, and answers can be entities $\mathcal{E}$ and (or) relation types $\mathcal{R}$ of the KG, $\text{Con} \subseteq \mathcal{E} \cup \mathcal{R}$, $\text{Var} \subseteq \mathcal{E} \cup \mathcal{R}$, $\mu \subseteq \mathcal{E} \cup \mathcal{R}$. That is, queries may only contain entities (or relations) and their answers are only entities (or relations). For example, a query $?x : \text{education}(\text{Hinton}, x)$ in Fig. 5 can

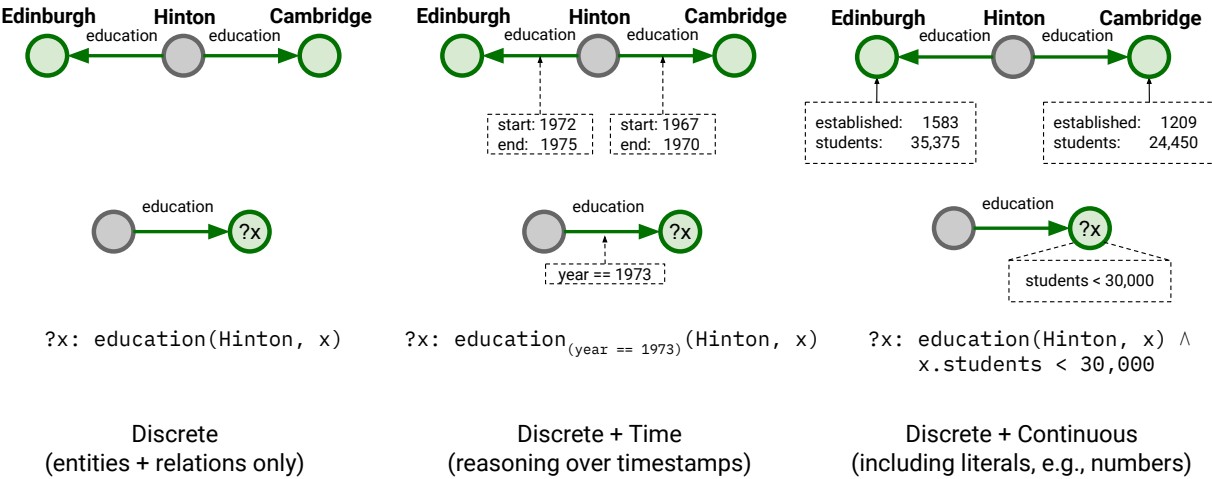

Figure 5: Reasoning Domains. The *Discrete* domain only allows entities and relations as constants, variables, and answers. The *Discrete + Time* domain extends the space to discrete timestamps and time-specific operators. The *Discrete + Continuous* domain allows continuous inputs (literals) and outputs.

only return two entities {Edinburgh, Cambridge} as answers. Conceptually, the framework allows relation types $r \in \mathcal{R}$ to be variables as well. For example, a SPARQL graph pattern {Hinton ?r Cambridge} – or $?r : r(\text{Hinton}, \text{Cambridge})$ in the functional form – that returns all relation types between two nodes Hinton and Cambridge. However, to the best of our knowledge, all current approximate query answering literature and datasets limit the queries such that they do never have variables in the relation position, *i.e.*, $\mathcal{R}' \subset \text{Con}$. (we discuss queries structure in more detail in Section 6). To date, most of the literature in the field belongs to the *Discrete* reasoning domain (Table 2).

Some nodes and edges might have *timestamps* from a set of discrete timestamps $t \in \mathcal{TS}$ indicating a validity period of a certain statement. In a more general case, certain subgraphs might be timestamped. We define the **Discrete + Time** domain when queries include temporal data. In this domain, the set of constants is extended with the set of timestamps, i.e., $\mathcal{TS} \subset \text{Con}$, and relation projections may be instantiated with a certain timestamp $R_t(a, b)$. With such a graph, one can extend set of queries and the set of possible answers to include time information. An example of such a work is this is the Temporal Feature-Logic Embedding Framework (TFLEX) by Lin et al. (2023) that defines additional operators **before**, **after**, **between** over edges with discrete timestamps.

For instance (Fig. 5), given a timestamped graph and a query $?x : \text{education}_{\text{year}==1973}(\text{Hinton}, x)$, the answer set includes only Edinburgh as the timestamp 1973 falls into the validity period of only one edge.

Finally, the most expressive domain is **Discrete + Continuous** that enables reasoning over continuous inputs (such as numbers, texts, continuous timestamps) often available as node and edge attributes or *literals*. Formally, for numerical data, the space of constants is extended with real numbers $\mathbb{R} \subset \text{Con}$. Also the query formalism is extended to allow reasoning over the real numbers. An example query in Fig. 5 $?x : \text{education}(\text{Hinton}, x) \wedge x.\text{students} < 30000$ includes a conjunctive term $x.\text{students} < 30000$ that requires numerical reasoning over the **students** attribute of a variable $x$ to produce the answer Cambridge. In a similar fashion, extending the answer set to continuous outputs can be framed as a regression task. To date, LitCQD (Demir et al., 2023) and NRN (Bai et al., 2023a) are the only approaches supporting numerical literals (in $\mathbb{R}$) as query constants and potential query answers. NRN, however, treats literals as discrete entities in the graph. LitCQD employs TransEA (Wu & Wang, 2018) (n addition to the ComplEx-based (Lacroix et al., 2018) link predictor) as a regressor to predict numerical values of entities' attributes. Incorporating literals into queries, LitCQD defines filtering operators **less than**, **greater than**, **equal** implemented as exponential functions of predicted and target values. For example, the term $x.\text{students} < 30000$ is represented as a conjuction $students(x, C) \wedge lt(C, 30000)$ of a relation projection and the less-than

filter. The ability to reason over continuous data is crucial for query answering given that most real-world KGs heavily rely on literals.

Table 2: Complex Query Answering approaches categorized under *Reasoning Domain*.

| Discrete | Discrete + Time | Discrete + Continuous |
|---|---|---|
| GQE (Hamilton et al., 2018), GQE w hash (Wang et al., 2019), CGA (Mai et al., 2019), TractOR (Friedman & Van den Broeck, 2020), Query2Box (Ren et al., 2020), BetaE (Ren & Leskovec, 2020), EmQL (Sun et al., 2020), MPQE (Daza & Cochez, 2020), Shv (Gebhart et al., 2023), Q2B Onto (Andresel et al., 2021), RotatE-Box (Adlakha et al., 2021), BiQE (Kotnis et al., 2021), HyPE (Choudhary et al., 2021b), NewLook (Liu et al., 2021), CQD (Arakelyan et al., 2021), PERM (Choudhary et al., 2021a), ConE (Zhang et al., 2021b), LogicE (Luus et al., 2021), MLPMix (Amayuelas et al., 2022), FuzzQE (Chen et al., 2022), GNN-QE (Zhu et al., 2022), GNNQ (Pflueger et al., 2022), SMORE (Ren et al., 2022), KGTrans (Liu et al., 2022), LinE (Huang et al., 2022b), Query2Particles (Bai et al., 2022), TAR (Tang et al., 2022), TeMP (Hu et al., 2022), FLEX (Lin et al., 2022), NodePiece-QE (Galkin et al., 2022b), ENeSy (Xu et al., 2022), GammaE (Yang et al., 2022a), NMP-QEM (Long et al., 2022), StarQE (Alivanistos et al., 2022), QTO (Bai et al., 2023c), SignalE (Wang et al., 2022), LMPNN (Wang et al., 2023e), NQE (Luo et al., 2023), Var2Vec (Wang et al., 2023a), CQD$^{\mathcal{A}}$ (Arakelyan et al., 2023), Query2Geom (Sardina et al., 2023), SQE (Bai et al., 2023b), RoConE (He et al., 2023), FIT (Yin et al., 2023b), CylE (Nguyen et al., 2023b), LARK (Choudhary & Reddy, 2023), WFRE (Wang et al., 2023d), BiDAG (Xu et al., 2023a), Query2Triple (Xu et al., 2023b), SCoNe (Nguyen et al., 2023a), CQD Onto (Andresel et al., 2023), UnRavL (Cucumides et al., 2024), UltraQuery (Galkin et al., 2024) | TFLEX (Lin et al., 2023) | LitCQD (Demir et al., 2023), NRN (Bai et al., 2023a) |

## 4.3 Background Semantics

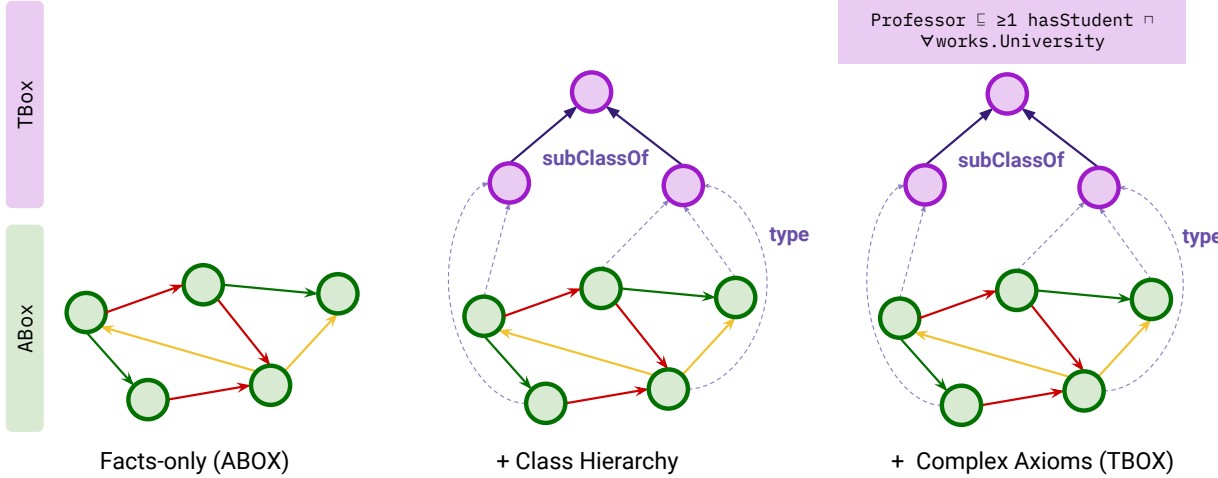

Figure 6: Background Semantics. *Facts-only* are graphs that only have assertions (ABox) and have no higher-level schema. *Class Hierarchy* introduces node types (classes) and hierarchical relationships between classes. Finally, *Complex Axioms* add even more complex logical rules (TBox), for example, governed by certain OWL profiles, *e.g.*, *A professor is someone who has one or more students and works at university.*

Relational databases often contain a *schema*, that is, a specification of how tables and columns are organized that gives a high-level overview of the database content. In graphs databases, schemas exist as well and, for instance, node types are common in the Labeled Property Graph (LPG) paradigm. RDF graphs employ the standards based on *Description Logics* (Baader et al., 2003). As incorporating schema is crucial for

designing effective query answering ML models, we introduce *Background Semantics* (Fig. 6) as the notion of additional schema information available on top of plain facts (statements).

**Facts-only.** In the simplest case, there is no background schema such that a KG consists of statements (facts) only, that is, a KG follows Definition 2.2, $\mathcal{G} = (\mathcal{E}, \mathcal{R}, \mathcal{S})$. In terms of description logics, the graph only has *assertions* (ABox). Queries, depending on the Reasoning Domain (Section 4.2), involve only entities, relations, and literals. The original GQE (Hamilton et al., 2018) focused on facts-only graphs and the majority of subsequent query answering approaches (Table 3) operate exclusively on schema-less KGs.

**Class Hierarchy.** Classes of entities, or node types, are a natural addition to a facts-only graph as a basic schema. Using Definition 2.2 with a set of types $\mathcal{T}$, a graph $\mathcal{G}$ is defined as $\mathcal{G} = (\mathcal{E}, \mathcal{R}, \mathcal{S})$, where $\mathcal{E} = \bar{\mathcal{E}} \cup \mathcal{T}$ (with $\bar{\mathcal{E}} \cap \mathcal{T} = \emptyset$). In other words, the types can be used as normal entities. To indicate that an entity has a type, a specially labeled edge can be used (e.g., `rdf:type`), or a labeling function can be defined; both options have equivalent expressive power. Because types are nodes themselves, it is possible to specify hierarchical relationships between classes using RDF Schema (RDFS) (Brickley et al., 2014)[3]. In LPG, it is more common to only have a labeling function, and not allow type hierarchies. Practically, edges involving types might be present physically in a KG or be considered as an additional input to a particular model.

To date, we are aware of three query answering approaches (Table 3) that incorporate entity types. Contextual Graph Attention (CGA, Mai et al. (2019)) only uses types for entity embedding initialization and requires each entity to have only one type. A similar technique was used in the evaluation of MPQE, but only to initialize the representation of variables (for which types were assumed given). The queries are not conditioned by entity types and the answer set still includes entities only. That is, constants in queries and those bound to variables are never types.

In Type-Aware Message Passing (TEMP, Hu et al. (2022)), type embeddings are used to enrich entity and relation representations that are later sent to a downstream query answering model. Each node might have several types. In the inductive scenario (we elaborate on inference scenarios in Section 5.1) with unseen nodes at inference time, type and relation embeddings are learned *invariants* that transfer across training and inference entities. Queries, and bound variables are still limited to non-types only ($\text{im}(\mu) \subseteq \bar{\mathcal{E}}$).

The TBox and ABox Neural Reasoner (TAR, Tang et al. (2022)) incorporates types and their hierarchy to improve predictions over entities. They also introduce the task of predicting types of answer entities, that is, $\mu(t) \subseteq (\bar{\mathcal{E}} \cup \mathcal{T})$, for one specific variable called $t$. Other variables cannot bind to types and constants in the query cannot be types either. The class hierarchy in TAR is used in three auxiliary losses besides the original entity prediction, that is, *concept retrieval* – prediction of the answer set of types, *subsumption* – predicting which type is a subclass of another type, and *instantiation* – predicting a type for each entity.

A natural next step for the *Class Hierarchy* family of approaches is to incorporate types in queries in the form of constants and variables.

**Complex Axioms.** Finally, a schema might contain not just a class hierarchy but a set of more complex axioms involving, for example, a hierarchy of relations, restrictions on relations, or composite classes. Such a complex schema can now be treated as an *ontology* $\mathcal{O}$ and we extend the definition of the graph to include it: $\mathcal{G} = (\mathcal{E}, \mathcal{R}, \mathcal{S}, \mathcal{O})$. In Fig. 6, the axiom `Professor` $\sqsubseteq\ \geq 1\ hasStudent \sqcap \forall works.$`University` describes that *A professor is someone who has one or more students and works at university.* In terms of description logics, a graph has an additional *terminology* component (TBox). The expressiveness of the TBox directly affects the complexity of symbolic reasoning engines up to exponential (EXPTIME) for most expressive fragments.

In graph representation learning, incorporating complex ontological axioms is non-trivial even for simple link prediction models (Zhang et al., 2022b). In the query answering literature, the only attempt to include complex axioms is taken by Andresel et al. (2021). In Q2B Onto (O2B), an extension of Query2Box (Q2B, Ren et al. (2020)), the set of considered complex axioms belongs to the *DL-Lite$_{\mathcal{R}}$* fragment and supports the hierarchy of classes (*subclasses*), the hierarchy of relations (*subproperties*), as well as *range* and *domain* of relations. The model architecture is not directly conditioned on the axioms and remains the original

---

[3]RDFS has more expressive means (*e.g.*, a hierarchy of relations) but we leave them to *Complex Axioms*

Query2Box. Instead, the axioms affect the graph structure, query sampling, and an auxiliary loss, that is, *query rewriting* mechanisms are used to materialize more answers to original queries as if executed against the complete graph (*deductive closure*) akin to data augmentation. During optimization, an auxiliary regularization loss aims at including a specialized query box $q$ into the more general version of this query $q'$.

Still, even the expensive procedure of incorporating complex axioms in query sampling in O2B benefits mostly the *deductive* capabilities of query answering, that is, inferring answers that are already implied by the graph $\mathcal{G}$ and ontology $\mathcal{O}$, and does not improve the *generalization* capabilities when missing edges cannot be inferred by ontological axioms. We elaborate on *deductive*, *generalization*, and other setups in Section 7.

Another avenue for future work is a better understanding of theoretical expressiveness of Graph Neural Network (GNN) encoders when applied to multi-relational KGs. Initial works on non-relational graphs (Barceló et al., 2020) map the expressiveness to the $\text{FOC}_2$ subset of FOL with two variables and counting quantifiers, and to $\text{FOC}_B$ (Luo et al., 2022) for hypergraphs of maximum arity $B$. In relational graphs, Barceló et al. (2022) quantified the expressiveness of relational GNNs in terms of the *relational Weisfeiler-Leman* (RWL) test proving that RWL is more expressive than classical WL test (Weisfeiler & Leman, 1968) and that common relational GNN architectures like R-GCN (Schlichtkrull et al., 2018) and CompGCN (Vashishth et al., 2020) are bounded by 1-RWL. Using RWL, Huang et al. (2023) derive that the family of GNNs conditioned on the query node, such as Neural Bellman-Ford Networks (Zhu et al., 2021), are bounded by the asymmetric local 2-RWL and expressive as $\text{rGFO}_{\text{cnt}}^3$, restricted guarded first-order logic fragment with three variables and counting. Concurrently, Gao et al. (2023) study KGs as double permutation equivariant structures (to permuting nodes and edge types) and map their expressiveness to universally quantified entity-relation (UQER) Horn clauses. However, it is still an open question if there exists GNNs architectures that can capture OWL-like axioms and leverage them as an inductive bias in complex query answering.

Table 3: Complex Query Answering approaches categorized under *Background Semantics* .

| Facts-only (ABOX) | + Class Hierarchy | + Complex Axioms (TBOX) |
|---|---|---|
| GQE (Hamilton et al., 2018), GQE w hash (Wang et al., 2019), TractOR (Friedman & Van den Broeck, 2020), Query2Box (Ren et al., 2020), BetaE (Ren & Leskovec, 2020), EmQL (Sun et al., 2020), Shv (Gebhart et al., 2023), RotatE-Box (Adlakha et al., 2021), MPQE (Daza & Cochez, 2020), BiQE (Kotnis et al., 2021), HyPE (Choudhary et al., 2021b), NewLook (Liu et al., 2021), CQD (Arakelyan et al., 2021), PERM (Choudhary et al., 2021a), ConE (Zhang et al., 2021b), LogicE (Luus et al., 2021), MLP-Mix (Amayuelas et al., 2022), FuzzQE (Chen et al., 2022), GNN-QE (Zhu et al., 2022), GNNQ (Pflueger et al., 2022), SMORE (Ren et al., 2022), KGTrans (Liu et al., 2022), LinE (Huang et al., 2022b), Query2Particles (Bai et al., 2022), FLEX (Lin et al., 2022), TFLEX (Lin et al., 2023), NodePiece-QE (Galkin et al., 2022b), ENeSy (Xu et al., 2022), GammaE (Yang et al., 2022a), NMP-QEM (Long et al., 2022), StarQE (Alivanistos et al., 2022), QTO (Bai et al., 2023c), SignalE (Wang et al., 2022), LMPNN (Wang et al., 2023e), NQE (Luo et al., 2023), Var2Vec (Wang et al., 2023a), CQD$^{\mathcal{A}}$ (Arakelyan et al., 2023), Query2Geom (Sardina et al., 2023), SQE (Bai et al., 2023b), RoConE (He et al., 2023), FIT (Yin et al., 2023b), LitCQD (Demir et al., 2023), CylE (Nguyen et al., 2023b), LARK (Choudhary & Reddy, 2023), WFRE (Wang et al., 2023d), BiDAG (Xu et al., 2023a), NRN (Bai et al., 2023a), Query2Triple (Xu et al., 2023b), SCoNe (Nguyen et al., 2023a), UnRavL (Cucumides et al., 2024), UltraQuery (Galkin et al., 2024) | CGA (Mai et al., 2019), TeMP (Hu et al., 2022), TAR (Tang et al., 2022) | Q2B Onto (Andresel et al., 2021), CQD Onto (Andresel et al., 2023) |

## 5   Modeling

In this section, we discuss the literature from the perspective of *Modeling*. Following the common methodology (Battaglia et al., 2018), we segment the *Modeling* methods through the lens of *Encoder-Processor-Decoder* modules (illustrated in Fig. 7). (1) The *Encoder* ENC() takes an input query $q$, target graph $\mathcal{G}$ with its entities and relations, and auxiliary inputs (*e.g.*, node, edge, graph features) to build their representations in

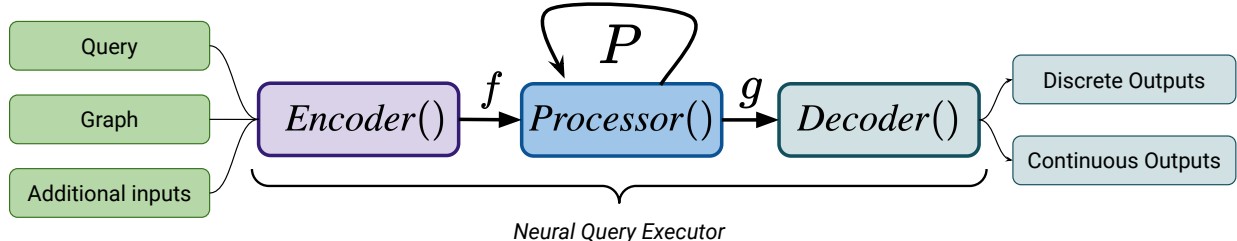

Figure 7: *Neural Query Execution* through the *Encoder-Processor-Decoder* modules. Encoder function $f$ builds representations of inputs (query, target graph, auxiliary data) in the latent space. Processor $P$ executes the query with its logical operators against the graph conditioned on other inputs. Decoder function $g$ builds requested outputs that might be discrete or continuous.

the latent space. (2) The *Processor P* leverages the chosen inductive biases to process representations of the query with its logical operators in the latent or symbolic space. (3) The *Decoder* DEC() takes the processed latents and builds desired outputs such as a distribution over discrete entities or regression predictions in case of continuous tasks. Generally, encoder, processor, and decoder can be parameterized with a neural network $\theta$ or be non-parametric. Finally, we analyze computational complexity of existing processors.

## 5.1 Encoder

We start the modeling section with encoders, *i.e.*, how different methods encode and represent entities and relations from the KG. There are three different categories, *Shallow Embedding*, *Transductive Encoder*, and *Inductive Encoder* each representing a different way of producing the neural representation of the entities/relations. Different encoding methods are suitable in different inference setups (details in Section 7.4), and may further require different logical operator methods (details in Section 5.2). Fig. 8 illustrates the three common encoding approaches.

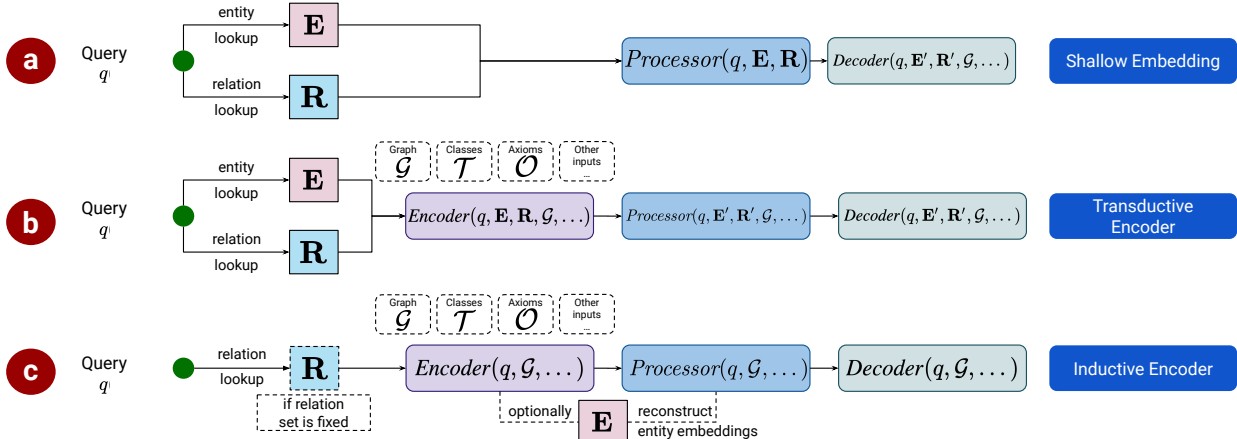

Figure 8: Categorization of *Encoders*. **a** Shallow encoders perform entity and relation embedding lookup and send them to the processor. **b** Transductive encoders additionally enrich the representations with query, graph, classes, or other latents. **c** Inductive encoders do not need learnable entity embeddings.

**Shallow Embeddings.** The first line of approaches encodes each entity/relation on the graph as a low-dimensional vector, and thus we achieve an entity embedding matrix $\boldsymbol{E}$ and a relation embedding matrix $\boldsymbol{R}$. The shape of the entity embedding matrix is $|\mathcal{E}| \times d$ ($|\mathcal{R}| \times d$), where $d$ is the dimension of the embedding. Shallow embedding methods assume independence of the representation of all the nodes on the graph.

Table 4: Complex Query Answering approaches categorized under *Encoder*.

| Shallow Embedding | Transductive Encoder | Inductive Encoder |
| --- | --- | --- |
| GQE (Hamilton et al., 2018), GQE w hash (Wang et al., 2019), CGA (Mai et al., 2019), TractOR (Friedman & Van den Broeck, 2020), Query2Box (Ren et al., 2020), BetaE (Ren & Leskovec, 2020), EmQL (Sun et al., 2020), Shv (Gebhart et al., 2023), Q2B Onto (Andresel et al., 2021), RotatE-Box (Adlakha et al., 2021), HyPE (Choudhary et al., 2021b), NewLook (Liu et al., 2021), CQD (Arakelyan et al., 2021), PERM (Choudhary et al., 2021a), ConE (Zhang et al., 2021b), LogicE (Luus et al., 2021), FuzzQE (Chen et al., 2022), SMORE (Ren et al., 2022), LinE (Huang et al., 2022b), Query2Particles (Bai et al., 2022), TAR (Tang et al., 2022), FLEX (Lin et al., 2022), TFLEX (Lin et al., 2023), GammaE (Yang et al., 2022a), NMP-QEM (Long et al., 2022), QTO (Bai et al., 2023c), SignalE (Wang et al., 2022), Var2Vec (Wang et al., 2023a), CQD$^{\mathcal{A}}$ (Arakelyan et al., 2023), Query2Geom (Sardina et al., 2023), RoConE (He et al., 2023), FIT (Yin et al., 2023b), LitCQD (Demir et al., 2023), CylE (Nguyen et al., 2023b), WFRE (Wang et al., 2023d), NRN (Bai et al., 2023a), SCoNe (Nguyen et al., 2023a), CQD Onto (Andresel et al., 2023) | MPQE (Daza & Cochez, 2020), BiQE (Kotnis et al., 2021), KGTrans (Liu et al., 2022), StarQE (Alivanistos et al., 2022), MLPMix (Amayuelas et al., 2022), ENeSy (Xu et al., 2022), LMPNN (Wang et al., 2023e), NQE (Luo et al., 2023), SQE (Bai et al., 2023b), BiDAG (Xu et al., 2023a), Query2Triple (Xu et al., 2023b), | NodePiece-QE (Galkin et al., 2022b), GNN-QE (Zhu et al., 2022), GNNQ (Pflueger et al., 2022), TeMP (Hu et al., 2022), UnRavL (Cucumides et al., 2024), UltraQuery (Galkin et al., 2024) |

This independence assumption gives the model much freedom, free parameters to learn. Such modeling origins from the KG completion literature, where the idea is to learn the entity and relation embedding matrices by optimizing a pre-defined distance/score function over all edges on the graph, *e.g.*, a triplet fact $\texttt{dist}(\mathbf{e_s}, \mathbf{r}, \mathbf{e_o})$. The majority of query answering literature follows the same paradigm with various different embedding spaces and distance functions to learn the entity and relation embedding matrices. Multiple embedding spaces have been proposed. For example, GQE (Hamilton et al., 2018) and Query2Box (Ren et al., 2020) embed into $\mathbb{R}^d$ (point vector in the Euclidean space); FuzzQE (Chen et al., 2022) embeds into the space of real numbers in range $[0, 1]$ (fuzzy logic score); BetaE (Ren & Leskovec, 2020) uses Beta distribution, a probabilistic embedding space; ConE Zhang et al. (2021b) on the other hand embeds entities as a point on a unit circle. Each design choice motivates the inductive bias for executing logical operators (as we show in Section 5.2). Some approaches employ shallow entity and relation embeddings already pre-trained on a simple link prediction task and just apply on top of them a query answering decoder with non-parametric logical operators. For example, CQD (Arakelyan et al., 2021), LMPNN (Wang et al., 2023e), Var2Vec (Wang et al., 2023a), and CQD$^{\mathcal{A}}$ (Arakelyan et al., 2023) take pre-trained embeddings in the complex space $\mathbb{C}^d$ and apply non-parametric *t-norms* and *t-conorms* to model intersection and union, respectively. QTO (Bai et al., 2023c) goes even further and fully materializes scores of all possible triples in one $[0, 1]^{|\mathcal{R}| \times |\mathcal{E}| \times |\mathcal{E}|}$ matrix given pre-trained entity and relation embeddings at preprocessing stage.

Despite being the mainstream design choice, the downside of shallow methods is that (1) shallow embeddings do not use any inductive bias and prior knowledge of the entity or its neighboring structure since the parameters of all entities/relations are free parameters learned from scratch; (2) they are not applicable in the inductive inference setting since these methods do not have a representation/embedding for those unseen novel entities by design. One possible solution is to randomly initialize one embedding vector for a novel entity and finetune the embedding vector by sampling queries involving the novel entity (detailed in Section 7.3). However, such a solution requires gradient steps during inference, rendering it not ideal.

**Transductive Encoder.** Similar to shallow embedding methods, transductive encoder methods learn the same entity embedding matrix $\mathcal{E}$. Besides, they learn an additional encoder $\textsc{Enc}_\theta(q, \boldsymbol{E}, \boldsymbol{R}, \dots)$ (parameterized with $\theta$) on top of the query $q$, entity and relation embedding matrices (and, optionally, other available inputs). The goal is to apply the encoder to the embeddings of entities in the query $q$ in order to capture dependencies between neighboring entities in the graph. Specifically, the additional encoder may take several rows of the feature matrix as input and further apply transformations. For example, BiQE (Kotnis et al., 2021) and kgTransformer (Liu et al., 2022) linearize a query graph $\mathcal{G}_q$ into a sequence and apply a Transformer (Vaswani et al., 2017) encoder that attends to all other embeddings in the query and obtain the final representation of the [MASK] token as the target query. MPQE (Daza & Cochez, 2020) and StarQE (Ali-

vanistos et al., 2022) run a message passing GNN architecture on top of the query graph $\mathcal{G}_q$ to enrich entity and relation embeddings and extract the final node representation as the query embedding. These methods share similar benefit and disadvantage of the shallow embeddings. Namely, there are many free parameters in the method to train. Unlike the shallow embeddings, the additional encoder leverages relational inductive bias between an entity and its neighboring entities or other entities in a query, allowing for a better learned entity representation and generalization capacity. However, since at its core the method is still based on the large look-up matrix of entity embeddings, it still exhibits the same downside that all such methods cannot be directly applied to an inductive setting where we may observe new entities.

**Inductive Encoder.** In order to address the aforementioned challenges of shallow embeddings and transductive encoders, inductive encoder methods aim to avoid learning an embedding matrix $\boldsymbol{E}$ for a fixed number of entities. Instead, inductive representations are often calculated by leveraging certain *invariances*, that is, the features that remain the same when transferred onto different graphs with new entities at inference time. As we describe in Section 7.4, inductive encoders might employ different invariances albeit the majority of inductive encoders rely on the assumption of the fixed set of relation types $\mathcal{R}$. Formally, following Definition 2.4, given a complex query $\mathcal{Q} = (\mathcal{E}', \mathcal{R}', \mathcal{S}', \bar{\mathcal{S}}')$ composed of entity and relation terms $\mathcal{E}', \mathcal{R}'$ (that, in turn, contain constants `Con` and variables `Var`), relation projections $R(a,b) \in \mathcal{S}$ (and, optionally, in $\bar{\mathcal{S}}'$), a target graph $\mathcal{G}$ (and, optionally, other inputs), inductive encoders learn a conditional representation function $\text{ENC}_\theta(e|\mathcal{E}', \mathcal{R}', \mathcal{G}, \dots)$ for each entity $e \in \mathcal{E}$. Galkin et al. (2022b) devise two families of inductive representations, *i.e.*, (1) inductive *node representations* and (2) inductive *relational structure* representations.

Inductive **node representation** approaches parameterize $\text{ENC}_\theta$ as a function of a fixed-size invariant vocabulary. For instance, NodePiece-QE (Galkin et al., 2022b) employs the invariant vocabulary of relation types and parameterizes each entity through the set of incident relations. TeMP (Hu et al., 2022) employs the invariant vocabulary of entity types and class hierarchy and injects their representations into entity representations. Inductive node representation approaches reconstruct embeddings of new entities and can be used as a drop-in replacement of shallow lookup tables paired with any *processor* method, *e.g.*, NodePiece-QE used CQD as the processor while TeMP was probed with GQE, Query2Box, BetaE, and LogicE processors.

Inductive **relational structure** representation methods parameterize $\text{ENC}_\theta$ as a function of the relative relational structure that only requires learning of relation embeddings and uses relations as invariants. Such methods often employ various *labeling tricks* (Zhang et al., 2021a) to label constants (anchor entities) of the input query $\mathcal{Q}$ such that after the message passing procedure all other nodes would encode a graph structure relative to starting nodes. In particular, GNN-QE (Zhu et al., 2022) labels anchor nodes with the embedding vector of the queried relations, *e.g.*, for a projection query $(h, r, ?)$ a node $h$ will be initialized with the embedding of relation $r$, whereas all other nodes are initialized with the zero vector. In this way, GNN-QE learns only relation embeddings $\boldsymbol{R}$ and GNN weights. GNNQ (Pflueger et al., 2022) represents a query with its variables and relations as a hypergraph and learns a relational structure through applying graph convolutions on hyperedges. Hyperedges are parameterized with multi-hot feature vectors of participating relations, so the only learnable parameters are GNN weights.

Still, there exists a set of open problems for inductive models. As the majority of inductive methods rely on learning relation embeddings, they cannot be easily used in setups where at inference time KGs are updated with new, unseen relation types, that is, relations are not invariant. This fact might require exploration of novel invariances and featurization strategies (Huang et al., 2022a; Gao et al., 2023; Chen et al., 2023). To date, the only fully-inductive CLQA model capable of generalizing to new, unseen graphs with new sets of entities and relations at inference time is UltraQuery (Galkin et al., 2024) that leverages the invariance of *relational structure*. In particular, relational representations (for both seen and new unseen relation types) are obtained through the graph of relational interactions and the conditional message passing GNN encoder which is independent from relation identities and thus generalizable to any multi-relational graph. Inductive models are more expensive to train in terms of both time and memory than shallow models and cannot yet be easily extended to large-scale graphs. We conjecture that inductive encoders will be in the focus of the future work in CLQA as generalization to unseen entities and graphs at inference time without re-training is crucial for updatability. Furthermore, updatability might increase the role of *continual learning* (Thrun, 1995; Ring, 1998) and amplify the negative effects of *catastrophic forgetting* (McCloskey & Cohen, 1989)

that have to be addressed by the encoders. Larger inference graphs also present a major *size generalization* issue (Yehudai et al., 2021; Buffelli et al., 2022; Zhou et al., 2022) when performance of GNNs trained on small graphs decreases when running inference on much larger graphs. The phenomenon has been observed by Galkin et al. (2022b) in the inductive complex query answering setup.

## 5.2 Processor

Having encoded the query and other available inputs, the *Processor P* executes the query in the latent (or symbolic) space against the input graph. Recall that a query $q$ is defined as $q(\mathcal{E}', \mathcal{R}', \mathcal{S}, \bar{\mathcal{S}})$ where $\mathcal{E}'$ and $\mathcal{R}'$ terms include constants Con and variables Var, statements in $\mathcal{S}$ and $\bar{\mathcal{S}}$ include relation projections $R(a, b)$, and logical operators *ops* over the variables. We define *Processor P* as a collection of modules that perform relation projections $R(a, b)$ given constants Con and logical operators $ops \subseteq \{\land, \lor, \neg, \dots\}$ over variables Var (we elaborate on the logical operators in Section 6.1). Depending on the chosen inductive biases and parameterization strategies behind those modules, we categorize *Processors* into *Neural* and *Neuro-Symbolic* (Table 5a). Furthermore, we break down the Neuro-Symbolic processors into *Geometric*, *Probabilistic*, and *Fuzzy Logic* (Table 5b). Note that in this section we omit pure entity encoder approaches like TeMP (Hu et al., 2022), NodePiece-QE (Galkin et al., 2022b), and NRN (Bai et al., 2023a) (for numerical literals only) that can be paired with any neural or neuro-symbolic processor. To describe processor models more formally, we denote **e** as an entity vector, **r** as a relation vector, and **q** as the query embedding that is often a function of **e** and **r**. We use $\mathcal{G}_q$ as the query graph.

Table 5: Categorization of Query Processors. Table 5a provides a general view on Neural, Symbolic, Neuro-Symbolic methods as well as encoders that can be paired with any processor. Table 5b further breaks down neuro-symbolic processors.

(a) Complex Query Answering approaches categorized under the *Processor* type.

| Any Processor | Neural | Neuro-Symbolic |
|---|---|---|
| TeMP (Hu et al., 2022), NodePiece-QE (Galkin et al., 2022b), NRN (Bai et al., 2023a) | GQE (Hamilton et al., 2018), GQE w/ hashing (Wang et al., 2019), CGA (Mai et al., 2019), BiQE (Kotnis et al., 2021), MPQE (Daza & Cochez, 2020), StarQE (Alivanistos et al., 2022), MLPMix (Amayuelas et al., 2022), Query2Particles (Bai et al., 2022), KGTrans (Liu et al., 2022), RotatE-m, DistMult-m, ComplEx-m (Ren et al., 2022), GNNQ (Pflueger et al., 2022), SignalE (Wang et al., 2022), LMPNN (Wang et al., 2023e), SQE (Bai et al., 2023b), LARK (Choudhary & Reddy, 2023), BiDAG (Xu et al., 2023a), Query2Triple (Xu et al., 2023b) | Table 5b |

(b) Neuro-symbolic Processors

| Geometric [Neuro-Symbolic] | Probabilistic [Neuro-Symbolic] | Fuzzy Logic [Neuro-Symbolic] |
|---|---|---|
| Query2Box (Ren et al., 2020), Query2Onto (Andresel et al., 2021), RotatE-Box (Adlakha et al., 2021), NewLook (Liu et al., 2021), Knowledge Sheaves (Gebhart et al., 2023), HypE (Choudhary et al., 2021b), ConE (Zhang et al., 2021b), Query2Geom (Sardina et al., 2023), CylE (Nguyen et al., 2023b), RoConE (He et al., 2023), SCoNe (Nguyen et al., 2023a) | BetaE (Ren & Leskovec, 2020), PERM (Choudhary et al., 2021a), LinE (Huang et al., 2022b), GammaE (Yang et al., 2022a), NMP-QEM (Long et al., 2022) | EmQL (Sun et al., 2020), TractOR (Friedman & Van den Broeck, 2020), CQD (Arakelyan et al., 2021), LogicE (Luus et al., 2021), FuzzQE (Chen et al., 2022), TAR (Tang et al., 2022), FLEX (Lin et al., 2022), TFLEX (Lin et al., 2023), UnRavL (Cucumides et al., 2024), ENeSy (Xu et al., 2022), QTO (Bai et al., 2023c), NQE (Luo et al., 2023), Var2Vec (Wang et al., 2023a), $\text{CQD}^{\mathcal{A}}$ (Arakelyan et al., 2023), FIT (Yin et al., 2023b), LitCQD (Demir et al., 2023), WFRE (Wang et al., 2023d), CQD Onto (Andresel et al., 2023), UnRavL (Cucumides et al., 2024), UltraQuery (Galkin et al., 2024) |

**Neural Processors.** Neural processors execute relation projections and logical operators directly in the latent space $\mathbb{R}^d$ parameterizing them with neural networks. To date, most existing purely neural approaches

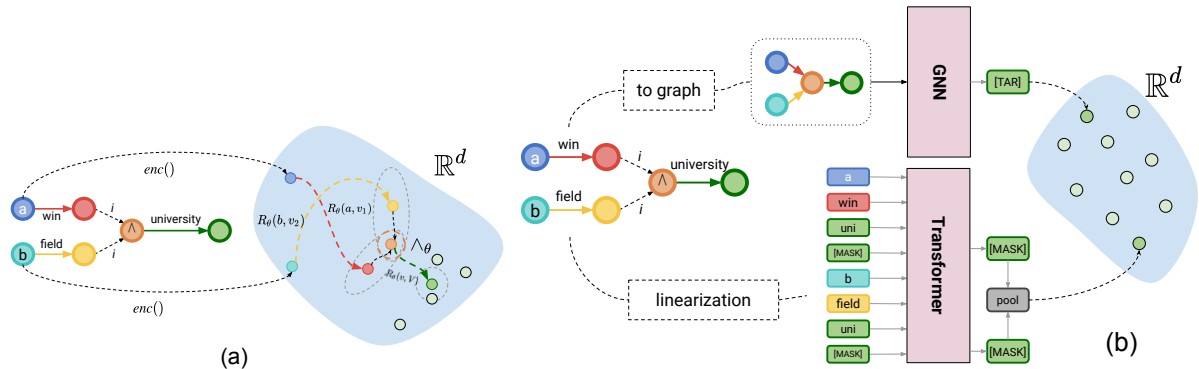

Figure 9: Neural Processors. (a) Relation projections $R_\theta(a, b)$ and logical operators (non-parametric or parameterized with $\theta$) are executed sequentially in the latent space; (b) a query is encoded to a graph or linearized to a sequence and passed through the encoder (GNN or Transformer, respectively). A pooled representation denotes the query embedding.

operate exclusively on the query graph $\mathcal{G}_q$ only executing operators within a single query and do not condition the execution process on the full underlying graph structure $\mathcal{G}$. Since the query processing is performed in the latent space with neural networks where Union ($\vee$) and Negation ($\neg$) are not well-defined, the majority of neural processors implement only relation projection ($R(a, b)$) and intersection ($\wedge$) operators. We aggregate the characteristics of neural processors as to their embedding space, the way of executing relation projection, logical operators, and the final decoding distance function in Table 6. We illustrate the difference between two families of neural processors (sequential execution and joint query encoding) in Fig. 9.

The original GQE (Hamilton et al., 2018) is the first example of the neural processor. That is, queries $\mathbf{q}$, entities $\mathbf{e}$, and relations $\mathbf{r}$ are vectors in $\mathbb{R}^d$. Query embedding starts with embeddings of constants $\mathcal{C}$ (anchor nodes $\mathbf{e}$) and they get progressively refined through relation projection and intersection, *i.e.*, it is common to assume that query embedding at the initial step 0 is equivalent to embedding(s) of anchor node(s), $\mathbf{q}^{(0)} = \mathbf{e}$. Relation projection is executed in the latent space with the translation function $\mathbf{q} + \mathbf{r}$, and intersection is modeled with the permutation-invariant DeepSet (Zaheer et al., 2017) neural network. Several follow-up works improved GQE to work with hashed binary vectors $\{+1, -1\}^d$ (Wang et al., 2019) or replaced DeepSet with self-attention and translation-based projection to a matrix-vector product (Mai et al., 2019). Recently, Ren et al. (2022) proposed DistMult-m, ComplEx-m, and RotatE-m, extensions of simple link prediction models for complex queries that, inspired by GQE, perform relation projection by the respective composition function and model the intersection operator with DeepSet and, optionally, L2 norm.

The other line of works apply neural encoders to whole query graphs $\mathcal{G}_q$ without explicit execution of logical operators. Depending on the query graph representation, such encoders are often GNNs, Transformers, or MLPs. It is assumed that neural encoders can implicitly capture logical operators in the latent space during optimization. For instance, MPQE (Daza & Cochez, 2020), StarQE (Alivanistos et al., 2022), and LMPNN (Wang et al., 2023e) represent queries as relational graphs (optionally, hyper-relational graphs for StarQE) where each edge is a relation projection and intersection is modeled as two incoming projections to the same variable node. All constants $\mathcal{C}$ and known relation types are initialized from the respective embedding matrices. All variable nodes in all query graphs are initialized with the same learnable `[VAR]` feature vector while all target nodes are initialized with the same `[TAR]` vector. Then, the query graph is passed through a GNN encoder (R-GCN (Schlichtkrull et al., 2018) for MPQE, StarE (Galkin et al., 2020) for StarQE, GIN (Xu et al., 2019) for LMPNN), and the final state of the `[TAR]` target node is considered the final query embedding ready for decoding. A recent LMPNN extends query graph encoding with an additional edge feature indicating whether a given projection $R(a, b)$ has a negation or not and derives a closed-form solution for the merged projection and negation operator for the ComplEx composition function. A different approach is taken by GNNQ (Pflueger et al., 2022) that frames query answering as a subgraph classification task. That is, an input query is not directly executed over a given graph $\mathcal{G}$, but, instead, the task is to classify whether a given precomputed subgraph $\mathcal{G}' \subset \mathcal{G}$ satisfies a given conjunctive query.

For that, GNNQ first augments the graph with Datalog-derived triples and converts the subgraph to a hypergraph where only hyperedges are parameterized with learnable vectors. On the one hand, this strategy allows GNNQ to be inductive and not learn entity embeddings. On the other hand, GNNQ is limited to conjunctive queries only and extensions to union and negation queries are not defined.

A more exotic approach by Gebhart et al. (2023) is based on the sheaf theory and algebraic topology (Hansen & Ghrist, 2019). There, a graph is represented as a cellular sheaf and conjunctive queries are modeled as chains of relations (0-cochains). A sheaf is induced over the query graph and relevant answers should be consistent with the induced sheaf and entity embeddings. The optimization problem is a harmonic extension of a 0-cochain using *sheaf Laplacian* and *Schur complement* of the sheaf Laplacian. Conceptually, this approach merges execution of projection and intersection operators as functions over topological structures.

Considering Transformer encoders, BiQE (Kotnis et al., 2021), kgTransformer (Liu et al., 2022), and SQE (Bai et al., 2023b) linearize a conjunctive query graph into a sequence of relational paths composed of entity constants $\mathcal{C}$ and relation tokens. The order of tokens in paths and intersections of paths are marked with positional encodings. The target node (present in many paths) is marked with the `[MASK]` token (optionally, kgTransformer also annotates existentially quantified variables with `[MASK]`). SQE does not model variables explicitly but instead relies on auxiliary *bracket* tokens that separate branches of the computation graph. Passing the sequence through the Transformer encoder, the final query embedding is the aggregated representation of the target node. BiQE only supports conjunctive queries while kgTransformer converts queries with unions to the Disjunctive Normal Form (DNF) with post-processing of score distributions (we elaborate on query rewritings and normal forms in Section 6.1). SQE explicitly includes all operator tokens into the linearized sequence and thus supports negations.

Finally, MLPMix (Amayuelas et al., 2022) sequentially executes operations of the query where projection, intersection, and negation operators are modeled as separate learnable MLPs. Union queries are converted to DNF such that they can be answered with projection and intersection operators with the final post-processing of scores as a union operator. Similarly, Query2Particles (Bai et al., 2022) represents each query as a set of vectors in the embedding space and models projection, intersection, and negation operators as attention over the set of particles followed by an MLP. Union is a concatenation of query particles.

Table 6: Neural Processors. Most methods implement only Relation Projection and Intersection operators. The top part of models execute a query sequentially, the bottom part encode the whole query graph $\mathcal{G}_q$.

| Model | Embedding Space | Relation Projection | Intersection | Union | Negation | Distance |
|---|---|---|---|---|---|---|
| GQE | $\mathbf{q},\mathbf{e},\mathbf{r}\in\mathbb{R}^d$ | $\mathbf{q}+\mathbf{r}$ | $\texttt{DeepSet}(\{\mathbf{q_i}\})$ | - | - | $\|\mathbf{q}-\mathbf{e}\|$ |
| GQE+hashing | $\mathbf{q},\mathbf{e}\in\{\pm1\}^d,\mathbf{r}\in\mathbb{R}^d$ | $\text{sgn}(\mathbf{q}+\mathbf{r})$ | $\text{sgn}(\texttt{DeepSet}(\{\mathbf{q_i}\}))$ | - | - | $-cos(\mathbf{q},\mathbf{e})$ |
| CGA | $\mathbf{q},\mathbf{e},\mathbf{r}\in\mathbb{R}^d$ | $\mathbf{W}_r\mathbf{q}$ | $\texttt{SelfAttn}(\{\mathbf{q_i}\})$ | - | - | $-cos(\mathbf{q},\mathbf{e})$ |
| RotatE-m | $\mathbf{q},\mathbf{e},\mathbf{r}\in\mathbb{C}^d$ | $\mathbf{q}\circ\mathbf{r}$ | $\texttt{DeepSet}(\{\mathbf{q_i}\})$ | - | - | $\|\mathbf{q}-\mathbf{r}\|$ |
| DisMult-m | $\mathbf{q},\mathbf{e},\mathbf{r}\in\mathbb{R}^d$ | $\texttt{L2Norm}(\mathbf{q}\circ\mathbf{r})$ | $\texttt{L2Norm}(\texttt{DeepSet}(\{\mathbf{q_i}\}))$ | - | - | $-\langle\mathbf{q},\mathbf{e}\rangle$ |
| ComplEx-m | $\mathbf{q},\mathbf{e},\mathbf{r}\in\mathbb{C}^d$ | $\texttt{L2Norm}(\mathbf{q}\circ\mathbf{r})$ | $\texttt{L2Norm}(\texttt{DeepSet}(\{\mathbf{q_i}\}))$ | - | - | $-\text{Re}(\langle\mathbf{q},\bar{\mathbf{e}}\rangle)$ |
| Query2Particles | $\mathbf{q},\mathbf{e}\in\mathbb{R}^{d\times K},\mathbf{r}\in\mathbb{R}^d$ | $f(\mathbf{q},\mathbf{r})$ $f$ is neural gates | $\texttt{MLP}(\texttt{Attn}([\mathbf{q_1},\mathbf{q_2}]))$ | $[\mathbf{q}^1,\ldots,\mathbf{q}^N]$ | $\texttt{MLP}(\texttt{Attn}(\mathbf{q}))$ | $-\max_k\langle\mathbf{q}^k,\mathbf{e}\rangle$ |
| SignalE | $\mathbf{q},\mathbf{e},\mathbf{r}=[\mathbf{z},\mathbf{v}]$ $\mathbf{z}\in\mathbb{C}^d,\mathbf{v}=\texttt{IDFT}(\mathbf{z})\in\mathbb{R}^d$ | $\mathbf{z}_q\circ\mathbf{z}_r$ | $\texttt{SelfAttn}(\{\mathbf{z}_{q_i}\})$ | DNF | $(1-\mathbf{z}_q^{\text{amp}},\mathbf{z}_q^{\text{phase}})$ | $\beta\|\mathbf{v}_q-\mathbf{v}_e\|_2+\|\mathbf{z}_q-\mathbf{z}_e\|_2$ |
| MLPMix | $\mathbf{q},\mathbf{e},\mathbf{r}\in\mathbb{R}^d$ | $\texttt{MLP}_{\texttt{mix}}(\mathbf{q},\mathbf{r})$ | $\texttt{MLP}_{\texttt{mix}}(\mathbf{q_1},\mathbf{q_2})$ | DNF | $\texttt{MLP}(\mathbf{q})$ | $\|\mathbf{q}-\mathbf{e}\|$ |
| MPQE | $\mathbf{q},\mathbf{e},\mathbf{r}\in\mathbb{R}^d$ | $\mathbf{W}_r\mathbf{q}$ | $\texttt{RGCN}(\mathbf{q},\mathcal{G}_q)$ | - | - | $cos(\mathbf{q},\mathbf{e})$ |
| StarQE | $\mathbf{q},\mathbf{e},\mathbf{r}\in\mathbb{R}^d$ | $f(\mathbf{q},g(\mathbf{r},\mathbf{h}_{\text{quals}}))$ | $\texttt{StarE}(\mathbf{q},\mathcal{G}_q)$ | - | - | $\texttt{dot}(\mathbf{q},\mathbf{e})$ |
| GNNQ | $\mathbf{q},\mathbf{r}\in\mathbb{R}^d$ | Hypergraph $\texttt{RGCN}(\mathcal{G}')$ | | - | - | BCE |
| LMPNN | $\mathbf{q},\mathbf{e},\mathbf{r}\in\mathbb{C}^d$ | $\rho(\mathbf{q},\mathbf{r},\text{dir},\text{neg})$ | $\texttt{GIN}(\mathbf{q},\mathcal{G}_q)$ | DNF | $\rho(\mathbf{q},\mathbf{r},\text{dir},\text{neg})$ | $cos(\mathbf{q},\mathbf{e})$ |
| BiQE | $\mathbf{q},\mathbf{e},\mathbf{r}\in\mathbb{R}^d$ | Transformer($\mathbf{q}$, linearized $\mathcal{G}_q$) | | - | - | CE |
| kgTransformer | $\mathbf{q},\mathbf{e},\mathbf{r}\in\mathbb{R}^d$ | Transformer($\mathbf{q}$, linearized $\mathcal{G}_q$) | | DNF | - | $\texttt{dot}(\mathbf{q},\mathbf{e})$ |
| SQE | $\mathbf{q},\mathbf{e},\mathbf{r}\in\mathbb{R}^d$ | LSTM / Transformer($\mathbf{q}$, linearized $\mathcal{G}_q$ with operator tokens) | | | | $\texttt{dot}(\mathbf{q},\mathbf{e})$ |
| Sheaves | $\mathbf{q},\mathbf{e}\in\mathbb{R}^d,\mathcal{F}_r\in\mathbb{R}^{d\times d}$ | $f(\mathbf{q},\mathcal{G}_q$ as cochain) | | - | - | $\|\mathcal{F}_r\mathbf{q}-\mathcal{F}_r\mathbf{e}\|$ |

**Neuro-Symbolic Processors.** In contrast to purely neural and symbolic models, we define *neuro-symbolic* processors as those who (1) explicitly design logic modules (or neural logical operators) that simulate the real logic/set operations, or rely on various kinds of fuzzy logic formulations to provide a probabilistic view of the query execution process, and (2) execute relation traversal in the latent space. The key difference between neuro-symbolic processors and the previous two is that neuro-symbolic processors explicitly model

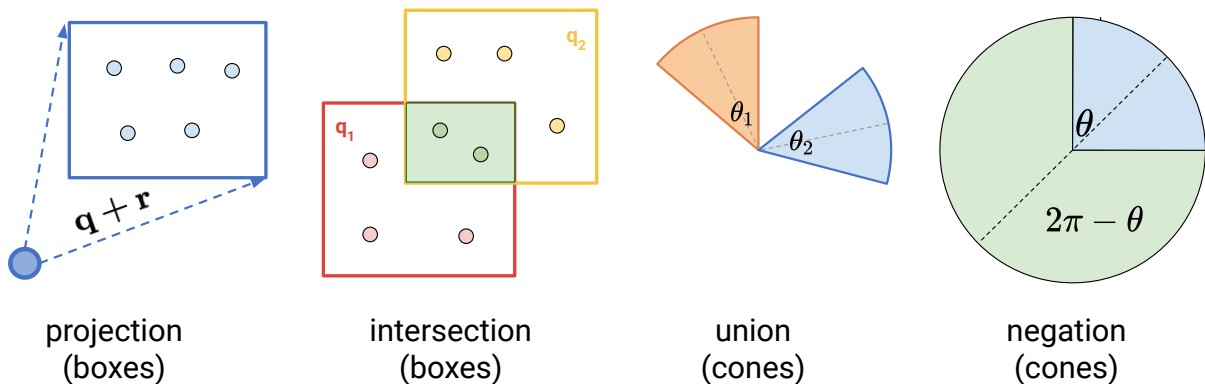

Figure 10: Geometric Processors and their inductive biases.

the logical operations with strong inductive bias so that the processing / execution is better aligned with the symbolic operation (*e.g.*, by imposing restrictions on the embedding space) and more interpretable. We further segment these methods into the following categories.

**Geometric Processors.** Geometric processors design an entity/query embedding space with different geometric intuitions and further customize neuro-symbolic operators that directly simulate their logical counterparts with similar properties (as illustrated in Fig. 10). We aggregate the characteristics of geometric models as to their embedding space and inductive biases for logical operators in Table 7.

Query2Box (Ren et al., 2020) embeds queries **q** as hyper-rectangles (high-dimensional boxes) in the Euclidean space. To achieve that, entities **e** and relations **r** are embedded as points in the Euclidean space where each relation has an additional learnable offset vector $\mathbf{r_o}$ (entities' offsets are zeros). The projection operator is modeled as an element-wise summation $\mathbf{q} + \mathbf{r}$ of centers and offsets of the query and relation, that is, the initial box is obtained by projecting the original anchor node embedding **e** (with zero offset) with the relation embedding **r** and relation offset $\mathbf{r_o}$. Accordingly, an attention-based neuro-intersection operator is designed to simulate the set intersection of the query boxes in the Euclidean space. The operator is closed, permutation invariant and aligns well with the intuition that the size of the intersected set is smaller than that of all input sets. The union operator is achieved via DNF, that is, union is the final step of concatenating results of operand boxes. Several works extend Query2Box, *i.e.*, Query2Onto (Andresel et al., 2021) attempts to model complex ontological axioms by materializing entailed triples and enforcing hierarchical relationship using inclusion of the box embeddings; RotatE-Box (Adlakha et al., 2021) designs an additional rotation-based Kleene plus (+) operator denoting relational paths (we elaborate on the Kleene plus operator in Section 6.1); NewLook (Liu et al., 2021) adds symbolic lookup from the adjacency tensor[4] to the operators, modifies projection with MLPs and models the *difference* operator as attention over centers and offsets (note that the difference operator is a particular case of the *2in* negation query, we elaborate on that in Section 6.1); Query2Geom (Sardina et al., 2023) replaces an attention-based intersection operator with a simple non-parametric closed-form geometric intersection of boxes.

HypE (Choudhary et al., 2021b) extends the idea of Query2Box and embeds a query as a hyperboloid (two parallel pairs of arc-aligned horocycles) in a Poincaré hyperball to better capture the hierarchical information. A similar attention-based neuro-intersection operator is designed for the hyperboloid embeddings with the goal to shrink the limits with DeepSets.

ConE (Zhang et al., 2021b), on the other hand, embeds queries on the surface of a set of unit circles. Each query is represented as a cone section and the benefit is that in most cases the intersection of cones is still a cone, and the negation/complement of a cone is also a cone thanks to the angular space bounded by $2\pi$. Based on this intuition, they design geometric neuro-intersection and negation operators. RoConE (He et al.,

---

[4]Incorrect implementation led to the major test set leakage and incorrect reported results.

Table 7: Geometric Neuro-Symbolic Processors. $\odot$ denotes element-wise multiplication, $\oplus_c$ denotes Möbius addition, $\odot_c$ denotes Möbius scalar product. $\texttt{DS}$ denotes DeepSets neural network. NewLook only partially implements negation as the difference operator.

| Model | Embedding Space | Relation Projection | Intersection | Union | Negation | Distance |
|---|---|---|---|---|---|---|
| Query2Box Query2Onto | $\mathbf{q}, \mathbf{r} \in \mathbb{R}^{2d}, \mathbf{e} \in \mathbb{R}^d$ | $\mathbf{q} + \mathbf{r}$ | $\mathbf{q}_c = \texttt{Attn}(\{\mathbf{q_c^i}\})$ $\mathbf{q_o} = \min(\{\mathbf{q_o^i}\}) \odot \sigma(\texttt{DS}(\{\mathbf{q_o^i}\}))$ | DNF | - | $\mathrm{d_{out}} + \alpha \mathrm{d_{in}}$ |
| Query2Geom | $\mathbf{q}, \mathbf{r} \in \mathbb{R}^{2d}, \mathbf{e} \in \mathbb{R}^d$ | $\mathbf{q} + \mathbf{r}$ | $\mathbf{q}_c = \frac{1}{2}((\mathbf{q_c^i} + \mathbf{q_o^i}) + (\mathbf{q_c^j} - \mathbf{q_o^j}))$ $\mathbf{q}_o = \mathbf{q}_c - (\mathbf{q_c^i} - \mathbf{q_o^i})$ | DNF | - | $\mathrm{d_{out}} + \alpha \mathrm{d_{in}}$ |
| RotatE-Box | $\mathbf{q}, \mathbf{r} \in \mathbb{C}^{2d}, \mathbf{e} \in \mathbb{C}^d$ | $(\mathbf{q_c} \circ \mathbf{r_c}, \mathbf{q_o} + \mathbf{r_o})$ | - | DNF / $\texttt{DS}(\{\mathbf{q}^i\})$ | - | $\mathrm{d_{out}} + \alpha \mathrm{d_{in}}$ |
| NewLook | $\mathbf{q}, \mathbf{r} \in \mathbb{R}^{2d}, \mathbf{e} \in \mathbb{R}^d$ $\mathbf{x} \in \mathcal{T}^{\|\mathcal{R}\| \times \|\mathcal{E}\| \times \|\mathcal{E}\|}$ | $\texttt{MLP}[\texttt{MLP}(\mathbf{q_c} + \mathbf{r_c}) \parallel$ $\texttt{MLP}(\mathbf{r_o}) \parallel \mathbf{x}_t]$ | $\mathbf{q_c} = \texttt{Attn}(\{\mathbf{q_c^i}, \mathbf{x}^i\})$ $\mathbf{q_o} = \min(\{\mathbf{q_o^i}\}) \odot \sigma(\texttt{DS}(\{\mathbf{q_o^i}\}))$ | DNF | $\texttt{Attn}(\{\mathbf{q_c^i}\})$ $\texttt{Attn}(\{\mathbf{q_o^i}, \mathbf{x}^i\})$ | $\mathrm{d_{out}} + \alpha \mathrm{d_{in}}$ |
| HypE | $\mathbf{q}, \mathbf{r} \in \mathbb{R}^{2d}, \mathbf{e} \in \mathbb{R}^d$ | $\mathbf{q} \oplus_c \mathbf{r}$ | $\mathbf{q_c} = \texttt{Attn}(\{\mathbf{q_c^i}\})$ $\mathbf{q_o} = \min(\{\mathbf{q_o^i}\}) \odot_c \sigma(\texttt{DS}(\{\mathbf{q_o^i}\}))$ | DNF | - | $\mathrm{d_{out}} + \alpha \mathrm{d_{in}}$ |
| ConE RoConE | $\mathbf{q}, \mathbf{r} = (\boldsymbol{\theta}_{\mathrm{ax}}, \boldsymbol{\theta}_{\mathrm{ap}})$ $\mathbf{e} = (\boldsymbol{\theta}_{\mathrm{ax}}, \mathbf{0})$ $\boldsymbol{\theta}_{\mathrm{ax}} \in [-\pi, \pi)^d$ $\boldsymbol{\theta}_{\mathrm{ap}} \in [0, 2\pi]^d$ | $g(\texttt{MLP}(\mathbf{q} + \mathbf{r}))$ $g$ gates $\boldsymbol{\theta}_{\mathrm{ax}}$ and $\boldsymbol{\theta}_{\mathrm{ap}}$ $\mathbf{q} \circ \mathbf{r}$ | $\boldsymbol{\theta}_{\mathrm{ax}} = \texttt{SemanticAvg}(\mathbf{q}_1, \dots, \mathbf{q}_n)$ $\boldsymbol{\theta}_{\mathrm{ap}} = \texttt{CardMin}(\mathbf{q}_1, \dots, \mathbf{q}_n)$ | DNF/DM | $\boldsymbol{\theta}_{\mathrm{ax}} = \boldsymbol{\theta}_{\mathrm{ax}} \pm \pi$ $\boldsymbol{\theta}_{\mathrm{ap}} = 2\pi - \boldsymbol{\theta}_{\mathrm{ap}}$ | $\mathrm{d_{out}} + \alpha \mathrm{d_{in}}$ |
| CylE | Same as ConE + $\boldsymbol{\theta}_{\mathrm{he}} \in (-\pi, \pi)^d$ | $g(\texttt{MLP}(\mathbf{q} + \mathbf{r}))$ | $\mathbf{q} = \texttt{Attn}(\{\mathbf{q}^i\})$ | DNF | Same as ConE | $\mathrm{d_{out}} + \alpha \mathrm{d_{in}}$ |

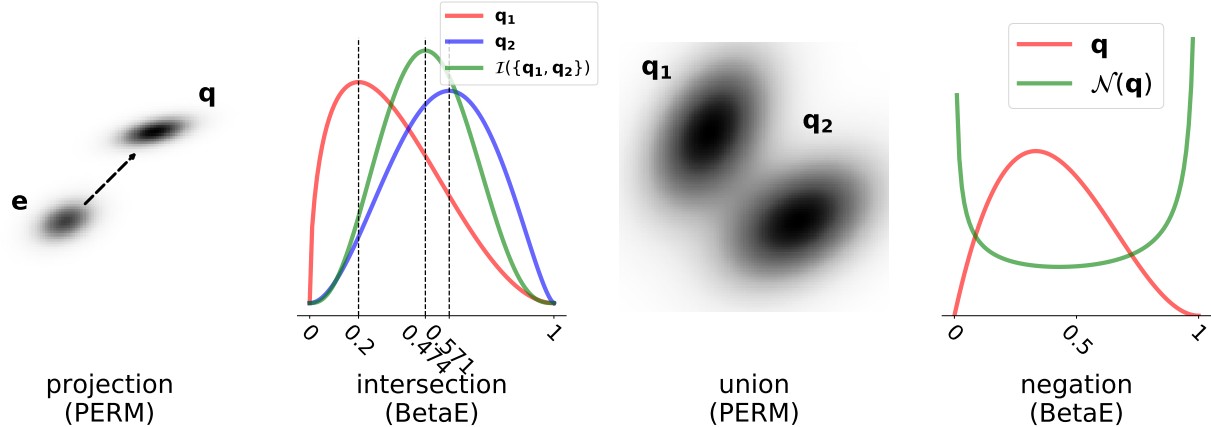

Figure 11: Probabilistic Processors and their inductive biases.

2023) replaces the projection operator of ConE with the angular rotation in the complex plane. CylE (Nguyen et al., 2023b) turns cones into cylinders by adding an additional height dimension.

To sum up, the geometric-based processors are often designed with a strong geometric prior such that properties of the logical/set operations can be better simulated or satisfied.

**Probabilistic Processors.** Instead of a geometric embedding space, probabilistic processors aim to model the query and the logic/set operations in a probabilistic space. Some examples of implementing logical operators in a probabilistic space are illustrated in Fig. 11. The aggregated characteristics are presented in Table 8.

BetaE (Ren & Leskovec, 2020) builds upon the Beta distribution and embeds entities/queries as high-dimensional Beta distribution with learnable parameters. The benefit is that one can design a parameterized neuro-intersection operator over two Beta embeddings where the output is still a Beta embedding with more concentrated density function. A neuro-negation operator can be designed by simply taking the reciprocal of the parameters in order to flip the density. PERM (Choudhary et al., 2021a) looks at the Gaussian distribution space and embeds queries as a multivariate Gaussian distribution. Since the product of Gaussian probability density functions (PDFs) is still a Gaussian PDF, the neuro-intersection operator accordingly calculates the parameters of the Gaussian embedding of the intersected set. NMP-QEM (Long et al., 2022)

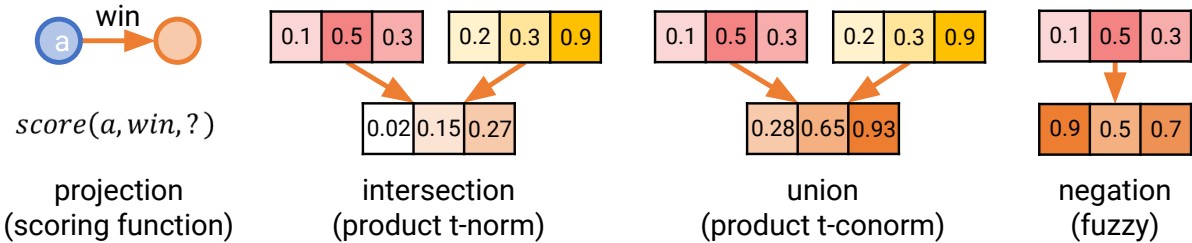

Figure 12: Fuzzy-Logic Processors and fuzzy logical operators.

develops this idea further and represents a query as a mixture of Gaussians where logical operators are modeled with MLP or attention over distribution parameters. LinE (Huang et al., 2022b) transforms the Beta distribution into a discrete sequence of values. A similar neuro-negation operator is introduced by taking the reciprocal as BetaE while designing a new neuro-intersection/union operator by taking element-wise min/max. GammaE (Yang et al., 2022a) replaces Beta distribution with Gamma distribution as entity and query embedding space. Parameterizing logical operators with operations over mixtures of Gamma distributions, union and negation become closed, do not need DNF or DM transformations, and can be executed sequentially along the query computation graph. Overall, probabilistic processors are similar to geometric processors since they are all inspired by certain properties of the probability and geometry used for embeddings and customize neuro-logic operators.

Table 8: Probabilistic Neuro-Symbolic Processors.

| Model | Embedding Space | Relation Projection | Intersection | Union | Negation | Distance |
|---|---|---|---|---|---|---|
| BetaE | $\mathbf{q}, \mathbf{e} \in \mathbb{R}^{2d}$ | $\text{MLP}_r(\mathbf{q})$ | $\mathbf{q} = [(\sum w_i \alpha_i, \sum w_i \beta_i)]$ | DNF/DM | $\frac{1}{\mathbf{q}}$ | $\text{KL}(\mathbf{e}; \mathbf{q})$ |
| PERM | $\mathbf{q}, \mathbf{e}, \mathbf{r} = \mathcal{N}(\boldsymbol{\mu}, \Sigma)$ | $\mathcal{N}(\boldsymbol{\mu}_q + \boldsymbol{\mu}_r, (\Sigma_q^{-1} + \Sigma_r^{-1})^{-1})$ | $\Sigma_q^{-1} = \Sigma_{q_1}^{-1} + \Sigma_{q_2}^{-1}$ $\boldsymbol{\mu}_q = \Sigma_q(\Sigma_{q_2}^{-1}\boldsymbol{\mu}_1 + \Sigma_{q_1}^{-1}\boldsymbol{\mu}_2)$ | $\text{Attn}(\{\mathbf{q}_i\})$ | - | $(\boldsymbol{\mu}_q - \boldsymbol{\mu}_e)^T \Sigma_q^{-1} (\boldsymbol{\mu}_q - \boldsymbol{\mu}_e)$ |
| LinE | $\mathbf{q}, \mathbf{e} \in \mathbb{R}^{k \times h}$ | $\text{MLP}_r(\mathbf{q})$ | $\min\{q_1, q_2\}$ | $\max\{q_1, q_2\}$ | $[\frac{1}{p_1^q}, \ldots, \frac{1}{p_k^q}]^h$ | $\|\mathbf{q} - \mathbf{e}\|_2^2$ |
| GammaE | $\mathbf{q}, \mathbf{e} \in \mathbb{R}^{2d}$ | $\text{MLP}_r(\mathbf{q})$ | $\mathbf{q} = [(\sum w_i \alpha_i, \sum w_i \beta_i)]$ | $\text{Attn}(\{\mathbf{q}_i\})$ | $[\frac{1}{\alpha}, \beta]$ | $\text{KL}(\mathbf{e}; \mathbf{q})$ |
| NMP-QEM | $\mathbf{e} \in \mathbb{R}^d$ $\mathbf{q} = \sum_{i=1}^{K} \omega_i \mathcal{N}(\boldsymbol{\mu}_i, \Sigma_i)$ | $\text{MLP}_r(\mathbf{q})$ | $\text{Attn}(\{\mathbf{q}_i\})$ | DNF | $\text{MLP}(\mathbf{q})$ | $\|\mathbf{e} - \sum_{i=1}^{K} \omega_i^q \boldsymbol{\mu}_i^q\|$ |

**Fuzzy-Logic Processors.** Unlike the methods above, fuzzy-logic processors directly model all logical operations using existing fuzzy logic theory (Klement et al., 2013; van Krieken et al., 2022) where intersection can be expressed via *t-norms* and union via corresponding *t-conorms* (Section A.4). In such a way, fuzzy-logic processors avoid the need to manually design or learn neural logical operators as in the previous two processors but rather directly use established fuzzy operators commonly expressed as differentiable, element-wise algebraic operators over vectors (Fig. 12). While intersection, union, and negation are non-parametric, the projection operator might still be parameterized with a neural network. The aggregated characteristics of fuzzy-logic processors are presented in Table 9. Generally, fuzzy processors aim to combine execution in *embedding space* (vectors) with *entity space* (symbols). The described methods are different in designing such a combination.

One of the first fuzzy processors is EmQL (Sun et al., 2020) that imbues entity and relation embeddings with a count-min sketch (Cormode & Muthukrishnan, 2005). There, projection, intersection, and union are performed both in the embedding space and in the symbolic sketch space, *e.g.*, intersection is modeled as an element-wise multiplication and union is an element-wise summation of two sketches. CQD (Arakelyan et al., 2021) scores each atomic formula in a query with a pretrained neural link predictor and uses t-norms and t-conorms to compute the final score of a query directly in the *embedding space*. CQD does not train any neural logical operators and only requires pretraining the entity and relation embeddings with one-hop links such that the projection operator is equivalent to *top-k* results of the chosen scoring function, *e.g.*, ComplEx (Lacroix et al., 2018). The idea was then extended in several directions: Query Tree Optimization (QTO) (Bai et al., 2023c) added a look-up from the materialized tensor of scores of all possible triples

$\mathbf{M} \in [0, 1]^{|\mathcal{R}| \times |\mathcal{E}| \times |\mathcal{E}|}$ to the relation projection step; CQD$^{\mathcal{A}}$ (Arakelyan et al., 2023) and Var2Vec (Wang et al., 2023a) learn an additional linear transformation of the entity-relation concatenation $\mathbf{W}[\mathbf{q}, \mathbf{r}]$; Fuzzy Inference with Truth Value (FIT) (Yin et al., 2023b) added several query rewriting steps to tackle DAG and cyclic queries, LitCQD (Demir et al., 2023) added a jointly trained regression model on entities' numerical attributes to include numerical literals into queries and answers. CQD Onto (Andresel et al., 2023) further adds the support for ontological axioms.

LogicE (Luus et al., 2021) designs logic embeddings for each entity with a list of *lower bound – upper bound* pairs in range $[0, 1]$, which can be interpreted as a uniform distribution between the lower and upper bound. LogicE executes negation and conjunction with continuous t-norms over the lower and upper bounds. FuzzQE (Chen et al., 2022), TAR (Tang et al., 2022), and WFRE (Wang et al., 2023d) embed query to a high-dimensional fuzzy space $[0, 1]^d$ and similarly use Gödel t-norm and Łukasiewicz t-norm to model disjunction, conjunction and negation. FuzzQE and WFRE model relation projection as a relation-specific MLP whereas TAR uses a geometric translation (element-wise sum). FLEX (Lin et al., 2022) and TFLEX (Lin et al., 2023) embed a query as a mixture of feature and logic embedding. For the logic part, both methods use the real logical operations in vector logic (Mizraji, 2008). TFLEX adds a temporal module conditioning logical operators on the time embedding.

GNN-QE (Zhu et al., 2022) models the likelihood of all entities for each relation projection step with a graph neural network NBFNet (Zhu et al., 2021). It further adopts product logic to directly model the set operations (intersection, union, and negation) over the fuzzy set obtained after a relation projection. GNN-QE employs a node labeling technique where a starting node is initialized with the relation vector (while other nodes are initialized with zeros). This allows GNN-QE to be inductive and not rely on trainable entity embeddings. UnRavL (Cucumides et al., 2024) extends the framework to queries that might not have a starting anchor node. ENeSy (Xu et al., 2022), on the other hand, maintains both vector and symbolic representations for queries, entities, and relations (where symbolic relations are encoded into $\mathbf{M}_r$ sparse adjacency matrices). Logical operators are executed first in the neural space, *e.g.*, relation projection is RotatE composition function (Sun et al., 2019), and then get intertwined with symbolic representations. Logical operators in the symbolic space employ a generalized version of the product logic and corresponding t-(co)norms.

In summary, fuzzy-logic processors directly rely on established fuzzy logic formalisms to perform all the logical operations in the query and avoid manually designing and learning neural operators in (possibly) unbounded embedding space. The fuzzy logic space is continuous but bounded within $[0, 1]$ – this is both the advantage and weakness of such processors. The bounded space is beneficial for closed logical operators as their output values still belong to the same bounded space. On the other hand, most of the known t-norms (and corresponding t-conorms) still lead to vanishing gradients and only the Product logic norms are stable (van Krieken et al., 2022; Badreddine et al., 2022). Another caveat is designing an effective and differentiable interaction mechanism between the fuzzy space $[0, 1]^d$ and unbounded embedding space $\mathbb{R}^d$ (or $\mathbb{C}^d$) where relation representations are often initialized from. That is, re-scaling and squashing of vector values when processing a computation graph might lead to noisy gradients and unstable training which is observed, for instance, by GNN-QE that has to turn off gradients from all but last projection step.

### 5.3 Decoder

The goal of decoding is to obtain the final set of answers or a ranking of all the entities. It is the final step of the query answering task after processing. Here we categorize the methods into two buckets: *non-parametric* and *parametric*. Parametric methods require a parameterized method to score an entity (or predict a regression target from the processed latents) while non-parametric methods can directly measure the similarity (or distance) between a pair of query and entity on the graph. Most of the methods belong to the non-parametric category as shown in the *Distance* column of processor tables Table 6, Table 7, Table 8, Table 9. For instance, geometric models (Ren et al., 2020; Andresel et al., 2021; Adlakha et al., 2021; Choudhary et al., 2021b; Zhang et al., 2021b) pre-define a distance function between the representation of the query and that of an entity. Commonly employed distance functions are L1 (Hamilton et al., 2018; Ren et al., 2022; Gebhart et al., 2023; Amayuelas et al., 2022; Tang et al., 2022; Xu et al., 2022; Long et al., 2022), L2 (Huang et al., 2022b; Chen et al., 2022), or their variations (Luus et al., 2021; Lin et al., 2022;

Table 9: Fuzzy Neuro-Symbolic Processors.

| Model | Embedding Space | Relation Projection | Intersection | Union | Negation | Distance |
|---|---|---|---|---|---|---|
| EmQL | $\mathbf{e}, \mathbf{r} \in \mathbb{R}^d, \mathbf{q} \in \mathbb{R}^{3d}$ 
 $\mathbf{b}$ count-min sketch | $\mathtt{MIPS}(\mathbf{q}, [\mathbf{r}, \mathbf{e}_h, \mathbf{e}_t])$ | $(\mathbf{q_1} + \mathbf{q_2})/2$ 
 $\mathbf{b_1} \odot \mathbf{b_2}$ | $(\mathbf{q_1} + \mathbf{q_2})/2$ 
 $\mathbf{b_1} + \mathbf{b_2}$ | - | $\mathtt{dot}(\mathbf{q}, \mathbf{e})$ |
| CQD 
 LitCQD | $\mathbf{q}, \mathbf{e}, \mathbf{r} \in \mathbb{C}^d$ | $\min_{\mathbf{q_2}} d(\mathbf{q_1}, \mathbf{r}, \mathbf{q_2})$ or 
 $\mathrm{top}_k(d(\mathbf{q_1}, \mathbf{r}, \mathbf{e_k}))$ | Product: $\mathbf{q_1} \cdot \mathbf{q_2}$ 
 Gödel: $\min(\mathbf{q_1}, \mathbf{q_2})$ | $\mathbf{q_1} + \mathbf{q_2} - \mathbf{q_1} \cdot \mathbf{q_2}$ 
 $\max(\mathbf{q_1}, \mathbf{q_2})$ | - | $\mathtt{dist}(\mathbf{q}, \mathbf{e})$ |
| CQD$^{\mathcal{A}}$ | $\mathbf{q}, \mathbf{e}, \mathbf{r} \in \mathbb{C}^d$ 
 $\mathbf{W} \in \mathbb{R}^{2 \times 2d}$ | $\theta = \mathbf{W}[\mathbf{q_1}, \mathbf{r}]$ 
 $\mathrm{top}_k[\rho_\theta(d(\mathbf{q_1}, \mathbf{r}, \mathbf{e_k}))]$ | Product: $\mathbf{q_1} \cdot \mathbf{q_2}$ 
 Gödel: $\min(\mathbf{q_1}, \mathbf{q_2})$ | $\mathbf{q_1} + \mathbf{q_2} - \mathbf{q_1} \cdot \mathbf{q_2}$ 
 $\max(\mathbf{q_1}, \mathbf{q_2})$ | $\mathbf{1} - \mathbf{q}$ 
 $(1 + \cos(\pi\mathbf{q}))/2$ | $\mathtt{dist}(\mathbf{q}, \mathbf{e})$ |
| Var2Vec | $\mathbf{q}, \mathbf{e}, \mathbf{r} \in \mathbb{C}^d$ 
 $\mathbf{W} \in \mathbb{R}^{d \times 2d}$ | $\mathbf{q_2} = \mathbf{W}[\mathbf{q_1}, \mathbf{r}]$ 
 $d(\mathbf{q_1}, \mathbf{r}, \mathbf{q_2})$ | Product: $\mathbf{q_1} \cdot \mathbf{q_2}$ 
 Gödel: $\min(\mathbf{q_1}, \mathbf{q_2})$ | $\mathbf{q_1} + \mathbf{q_2} - \mathbf{q_1} \cdot \mathbf{q_2}$ 
 $\max(\mathbf{q_1}, \mathbf{q_2})$ | $1 - \mathbf{q}$ | $\mathtt{dist}(\mathbf{q}, \mathbf{e})$ |
| QTO 
 FIT | $\mathbf{q}, \mathbf{e}, \mathbf{r} \in \mathbb{C}^d$ 
 $\mathbf{M} \in [0,1]^{\|\mathcal{R}\| \times \|\mathcal{E}\| \times \|\mathcal{E}\|}$ | $d(\mathbf{q_1}, \mathbf{r}, \mathbf{q_2})$ 
 $\mathrm{row}_{q1}(\mathbf{M}_r)$ | $\mathbf{q_1} \cdot \mathbf{q_2}$ | $\mathbf{q_1} + \mathbf{q_2} - \mathbf{q_1} \cdot \mathbf{q_2}$ | $d(\mathbf{q_1}, \mathbf{r}, \mathbf{q_2})$ 
 $\mathrm{row}_{q1}(\mathbf{1} - \mathbf{M}_r)$ | $\mathtt{dist}(\mathbf{q}, \mathbf{e})$ |
| LogicE | $\mathbf{q}, \mathbf{e} = ([l_i, u_i],$ 
 $l_i, u_i \in [0,1])_{i=1}^d$ 
 $\mathbf{r} \in \mathbb{R}^d$ | $\sigma(\max(0, \max(0,$ 
 $[\mathbf{r}, \mathbf{q}]\mathbf{F}_1)\mathbf{F}_2)\mathbf{F}_3)$ | $([\top(l_i^{(1)}, \ldots, l_i^{(n)}),$ 
 $\top(u_i^{(1)}, \ldots, u_i^{(n)})])$ 
 $i = 1 \ldots d$ | DM | $([1 - l_i,$ 
 $1 - u_i])$ 
 $i = 1 \ldots d$ | $\|\mathbf{q} - \mathbf{e}\|_1$ |
| FuzzQE | $\mathbf{q}, \mathbf{e} \in [0,1]^d$ | $\sigma(\mathtt{MLP}_r(\mathbf{q}))$ | Product: $\mathbf{q_1} \cdot \mathbf{q_2}$ 
 Gödel: $\min(\mathbf{q_1}, \mathbf{q_2})$ | $\mathbf{q_1} + \mathbf{q_2} - \mathbf{q_1} \cdot \mathbf{q_2}$ 
 $\max(\mathbf{q_1}, \mathbf{q_2})$ | $\mathbf{1} - \mathbf{q}$ 
 - | $\mathtt{dot}(\mathbf{q}, \mathbf{e})$ |
| WFRE | $\mathbf{q}, \mathbf{e} \in [0,1]^d$ | $\sigma(\mathtt{MLP}_r(\mathbf{q}))$ | $\min(\mathbf{q_1}, \mathbf{q_2})$ | $\max(\mathbf{q_1}, \mathbf{q_2})$ | $\mathbf{1} - \mathbf{q}$ | $\mathtt{WFR}(\mathbf{q}, \mathbf{e})$ |
| TAR | $\mathbf{q}, \mathbf{e}, \mathbf{r} \in \mathbb{R}^d$ | $\mathbf{q} + \mathbf{r}$ | $\mathtt{Attn}(\mathbf{q_1}, \mathbf{q_2})$ | $\max(\mathbf{q_1}, \mathbf{q_2})$ | $\mathbf{1} - \mathbf{q}$ | $\|\mathbf{q} - \mathbf{e}\|$ |
| GNN-QE | $\mathbf{q} \in \mathbb{R}^{\|\mathcal{E}\|}, \mathbf{r} \in \mathbb{R}^d$ | $\sigma(\mathtt{GNN}(\mathbf{q}, \mathcal{G}))$ | $\mathbf{q_1} \cdot \mathbf{q_2}$ | $\mathbf{q_1} + \mathbf{q_2} - \mathbf{q_1} \cdot \mathbf{q_2}$ | $\mathbf{1} - \mathbf{q}$ | $\mathtt{BCE}(\mathbf{q})$ |
| FLEX | $\mathbf{q} = (\theta_f, \theta_l)$ 
 $\theta_f \in [-L, L]^d$ 
 $\mathbf{e} = (\theta_f, \mathbf{0})$ 
 $\theta_l \in [0,1]^d$ | $g(\mathtt{MLP}([\theta_f + \theta_{f,r};$ 
 $\theta_l + \theta_{l,r}]))$ 

 $g$ gates $\theta_f, \theta_e$. | $\theta_f = \sum_i \mathbf{a}_i \theta_{q_i, f}$ 

 $\theta_l = \Pi_i \theta_{q_i, l}$ | $\theta_f = \sum_i \mathbf{a}_i \theta_{q_i, f}$ 
 $\theta_l = \sum_i \theta_{q_i, l} -$ 
 $\sum_{1 \le i < j \le n} \theta_{q_i, l} \theta_{q_j, l} +$ 
 $\cdots + (-1)^{n-1} \Pi_i \theta_{q_i, l}$ | $\theta_f = L \cdot \tanh($ 
 $\mathtt{MLP}([\theta_f; \theta_l]))$ 

 $\theta_l = 1 - \theta_l$ | $\|\theta_f^e - \theta_f^q\|_1 +$ 
 $\theta_l^q$ |
| TFLEX | $\mathbf{q}, \mathbf{r} = (\mathbf{q}_f^e, \mathbf{q}_l^e, \mathbf{q}_f^t, \mathbf{q}_l^t)$ 
 $\mathbf{q}_f^e, \mathbf{q}_f^t, \mathbf{e}_f^e \in \mathbb{R}^d$ 

 $\mathbf{e} = (\mathbf{e}_f^e, \mathbf{0}, \mathbf{0}, \mathbf{0})$ 
 $\mathbf{q}_l^e, \mathbf{q}_l^t \in [0,1]^d$ | Entity: $g(\mathtt{MLP}($ 
 $\mathbf{q} + \mathbf{r} + \mathbf{t}))$ 

 Time: $g(\mathtt{MLP}($ 
 $\mathbf{q_1} + \mathbf{r} + \mathbf{q_2}))$ | $\sum_i \alpha \mathbf{q}_{i,f}^e, \bigotimes_i(\{\mathbf{q}_{i,l}^e\}),$ 
 $\sum_i \beta \mathbf{q}_{i,f}^t, \bigotimes_i(\{\mathbf{q}_{i,l}^t\})$ 

 $\sum_i \alpha \mathbf{q}_{i,f}^e, \bigotimes_i(\{\mathbf{q}_{i,l}^e\}),$ 
 $\sum_i \beta \mathbf{q}_{i,f}^t, \bigotimes_i(\{\mathbf{q}_{i,l}^t\})$ | $\sum_i \alpha \mathbf{q}_{i,f}^e, \bigoplus_i(\{\mathbf{q}_{i,l}^e\}),$ 
 $\sum_i \beta \mathbf{q}_{i,f}^t, \bigotimes_i(\{\mathbf{q}_{i,l}^t\})$ 

 $\sum_i \alpha \mathbf{q}_{i,f}^e, \bigoplus_i(\{\mathbf{q}_{i,l}^e\}),$ 
 $\sum_i \beta \mathbf{q}_{i,f}^t, \bigotimes_i(\{\mathbf{q}_{i,l}^t\})$ | $f_{\mathrm{not}}^e(\mathbf{q}_f^e),$ 
 $\ominus(\mathbf{q}_l^e), \mathbf{q}_f^t, \mathbf{q}_l^t$ 

 $\mathbf{q}_f^e, \mathbf{q}_l^e,$ 
 $f_{\mathrm{not}}^t(\mathbf{q}_f^t), \ominus(\mathbf{q}_l^t)$ | $\|\mathbf{e}_f^e - \mathbf{q}_f^e\|_1 +$ 
 $\mathbf{q}_f^e\|_1 + \mathbf{q}_l^e$ 

 $\|\mathbf{t}_f^t - \mathbf{q}_f^t\|_1 +$ 
 $\mathbf{q}_l^t$ |
| ENeSy | $\mathbf{q}, \mathbf{e}, \mathbf{r} \in \mathbb{C}^d$ 
 $\mathbf{p}_q, \mathbf{p}_e \in \{0,1\}^{\|\mathcal{E}\|}$ 
 $\mathbf{M}_r \in \{0,1\}^{\|\mathcal{E}\| \times \|\mathcal{E}\|}$ | Neural: $\mathbf{q} \circ \mathbf{r}$ 
 Symb: $g(\mathbf{p}_q \mathbf{M}_r)^\top$ 
 $g = \mathbf{x}/\mathrm{sum}(\mathbf{x})$ | $g(\mathbf{p_1} \cdot \mathbf{p_2})$ | $g(\mathbf{p_1} + \mathbf{p_2} - \mathbf{p_1} \cdot \mathbf{p_2})$ | $g(\frac{\alpha}{\|\mathcal{E}\|} - \mathbf{p})$ | $\|\mathbf{q} - \mathbf{e}\|$ |
| NQE | $\mathbf{q}, \mathbf{e}, \mathbf{r} \in [0,1]^d$ | $\sigma(\mathtt{Trf}(\mathbf{q}, \mathcal{G}_q))$ | $\mathbf{q_1} \cdot \mathbf{q_2}$ | $\mathbf{q_1} + \mathbf{q_2} - \mathbf{q_1} \cdot \mathbf{q_2}$ | $\mathbf{1} - \mathbf{q}$ | $\mathtt{dot}(\mathbf{q}, \mathbf{e})$ |

2023), cosine similarity (Wang et al., 2019; Mai et al., 2019; Daza & Cochez, 2020; Wang et al., 2023e), dot product (Sun et al., 2020; Ren et al., 2022; Alivanistos et al., 2022; Liu et al., 2022; Bai et al., 2022; Luo et al., 2023), or naturally model the likelihood of all the entities without the need of a distance function (Arakelyan et al., 2021; Kotnis et al., 2021; Zhu et al., 2022; Pflueger et al., 2022; Bai et al., 2023c; Arakelyan et al., 2023; Wang et al., 2023a). Probabilistic models often employ KL divergence (Ren & Leskovec, 2020; Yang et al., 2022a) or Mahalanobis distance (Choudhary et al., 2021a). A rather exotic approach is Wasserstein-Fisher-Rao metric (WFR) used in WFRE Wang et al. (2023d). WFR is an optimal transport (OT) metric between distributions. Due to its computational complexity, the metric is approximated by the 1D convolution-based Sinkhorn optimization routine.

One important direction (orthogonal to the distance function) that current methods largely ignore is how to perform efficient answer entity retrieval over extremely large graphs with billions of entities. A scalable and approximate nearest neighbor (ANN) search algorithm is necessary. Existing frameworks including FAISS (Johnson et al., 2019) or ScaNN (Guo et al., 2020) provide scalable implementations of ANN. However, ANN is limited to L1, L2 and cosine distance and mostly optimized for CPUs. It is still an open research problem how to design efficient scalable ANN search algorithms for more complex distance functions such as KL divergences so that we can retrieve with much better efficiency for different CLQA methods with different distance functions (preferably, using GPUs).

We conjecture that parametric decoders are to gain more traction in numerical tasks on top of plain entity retrieval for query answering. Such tasks might involve numerical and categorical features on node-, edge-, and graph levels, *e.g.*, training a regressor to predict numerical values for node attributes like *age*, *length*, etc. Besides, a parametric decoder gives new opportunities to generalize to inductive settings where we may have unseen entities during evaluation. SE-KGE (Mai et al., 2020) takes a step in this direction by predicting geospatial coordinates of query targets. LitCQD (Demir et al., 2023) expands the set of query variables and answers to continuous numerical values by joint training of a link prediction model between entities and a regression model on top of entities' attributes with numerical values.

## 5.4 Computation Complexity

Here we analyze the time complexity of different query reasoning models categorized by different operations, including relation projection, intersection, union, negation and answer retrieval after obtaining the representation of the query. We list the asymptotic complexity in Table 10 for methods that perform stepwise encoding of the query graph, and Table 11 for methods (mostly GNN-based) that encode the whole query graph simultaneously including projection and other logic operations.

Table 10: Time complexity of each operation on $\mathcal{G} = (\mathcal{E}, \mathcal{R}, \mathcal{S})$. Across all methods, we denote the embedding dimension and hidden dimension of MLPs as $d$, number of layers of MLPs/GNNs as $l$, number of branches in an intersection operation as $i$, number of branches in a union operation as $u$.

| Model | Projection | Intersection | Negation | Union | Answer Retrieval | Definitions |
|---|---|---|---|---|---|---|
| GQE GQE+hashing RotatE-m Distmult-m ComplEx-m | $\mathcal{O}(d)$ | $\mathcal{O}(ild^2)$ | - | - | $\mathcal{O}(|\mathcal{E}|d)$ | - |
| CGA | $\mathcal{O}(d^2)$ | $\mathcal{O}(id + d^2)$ | - | - | $\mathcal{O}(|\mathcal{E}|d)$ | - |
| Query2Particles | $\mathcal{O}(Kd^2)$ | $\mathcal{O}(iKd^2)$ | $\mathcal{O}(Kd^2)$ | DNF | $\mathcal{O}(|\mathcal{E}|dK)$ | $K$ : #particles. |
| SignalE | $\mathcal{O}(d)$ | $\mathcal{O}(ild^2)$ | $\mathcal{O}(d)$ | DNF | $\mathcal{O}(|\mathcal{E}|d)$ | - |
| MLPMix | $\mathcal{O}(ld^2)$ | $\mathcal{O}(ild^2)$ | $\mathcal{O}(ld^2)$ | DNF | $\mathcal{O}(|\mathcal{E}|d)$ | - |
| Query2Box Query2Onto | $\mathcal{O}(d)$ | $\mathcal{O}(ild^2)$ | - | DNF | $\mathcal{O}(|\mathcal{E}|d)$ | - |
| Query2Geom | $\mathcal{O}(d)$ | $\mathcal{O}(d)$ | - | DNF | $\mathcal{O}(|\mathcal{E}|d)$ | - |
| RotatE-Box | $\mathcal{O}(d)$ | - | - | DNF / $\mathcal{O}(uld^2)$ | $\mathcal{O}(|\mathcal{E}|d)$ | - |
| NewLook | $\mathcal{O}(|\mathcal{E}|d + ld^2)$ | $\mathcal{O}(i(|\mathcal{E}| + ld^2))$ | $\mathcal{O}(ld^2)$ | DNF | $\mathcal{O}(|\mathcal{E}|d)$ | - |
| HypE | $\mathcal{O}(d)$ | $\mathcal{O}(ild^2)$ | - | DNF | $\mathcal{O}(|\mathcal{E}|d)$ | - |
| ConE/BetaE | $\mathcal{O}(ld^2)$ | $\mathcal{O}(ild^2)$ | $\mathcal{O}(d)$ | DNF | $\mathcal{O}(|\mathcal{E}|d)$ | - |
| PERM | $\mathcal{O}(d^2)$ | $\mathcal{O}(id^3)$ | - | DNF | $\mathcal{O}(|\mathcal{E}|d^2)$ | - |
| LinE | $\mathcal{O}(ld^2)$ | $\mathcal{O}(d)$ | $\mathcal{O}(d)$ | $\mathcal{O}(d)$ | $\mathcal{O}(|\mathcal{E}|d)$ | - |
| GammaE | $\mathcal{O}(ld^2)$ | $\mathcal{O}(ild^2)$ | $\mathcal{O}(d)$ | $\mathcal{O}(uld^2)$ | $\mathcal{O}(|\mathcal{E}|d)$ | - |
| NMP-QEM | $\mathcal{O}(Kld^2)$ | $\mathcal{O}(iK^2d)$ | $\mathcal{O}(Kld^2)$ | DNF | $\mathcal{O}(|\mathcal{E}|dK)$ | $K$ : # centers |
| EmQL | $\mathcal{O}(|\mathcal{E}|d)$ | $\mathcal{O}(d)$ | - | $\mathcal{O}(d)$ | $\mathcal{O}(|\mathcal{E}|d)$ | - |
| CQD-CO | $\mathcal{O}(|\mathcal{E}|d)$ | Opt | - | Opt | $\mathcal{O}(|\mathcal{E}|d)$ | - |
| CQD-Beam | $\mathcal{O}(|\mathcal{E}|d)$ | $\mathcal{O}(|\mathcal{E}|kd)$ | - | $\mathcal{O}(|\mathcal{E}|kd)$ | $\mathcal{O}(|\mathcal{E}|d)$ | - |
| QTO | $\mathcal{O}(\max_k |T^*(v_k) > 0||\mathcal{E}|)$ | $\mathcal{O}(|\mathcal{E}|)$ | $\mathcal{O}(|\mathcal{E}|)$ | $\mathcal{O}(|\mathcal{E}|)$ | $\mathcal{O}(|\mathcal{E}|)$ | $T^*(v)$ : the maximum truth value for the subquery rooted at node $v$. |
| LogicE | $\mathcal{O}(ld^2)$ | $\mathcal{O}(d)$ | $\mathcal{O}(d)$ | $\mathcal{O}(d)$ | $\mathcal{O}(|\mathcal{E}|d)$ | - |
| FuzzQE | $\mathcal{O}(d^2)$ | $\mathcal{O}(d)$ | $\mathcal{O}(d)$ | $\mathcal{O}(d)$ | $\mathcal{O}(|\mathcal{E}|d)$ | - |
| TAR | $\mathcal{O}(d)$ | $\mathcal{O}(d)$ | $\mathcal{O}(d)$ | $\mathcal{O}(d)$ | $\mathcal{O}(|\mathcal{E}|d)$ | - |
| GNN-QE | $\mathcal{O}(|\mathcal{E}|d^2 + |\mathcal{S}|d)$ | $\mathcal{O}(|\mathcal{E}|)$ | $\mathcal{O}(|\mathcal{E}|)$ | $\mathcal{O}(|\mathcal{E}|)$ | $\mathcal{O}(|\mathcal{E}|)$ | - |
| FLEX | $\mathcal{O}(ld^2)$ | $\mathcal{O}(ild^2)$ | $\mathcal{O}(ld^2)$ | $\mathcal{O}(uld^2 + 2^u d)$ | $\mathcal{O}(|\mathcal{E}|d)$ | - |
| TFLEX | $\mathcal{O}(ld^2)$ | $\mathcal{O}(ild^2)$ | $\mathcal{O}(ld^2)$ | $\mathcal{O}(uld^2)$ | $\mathcal{O}(|\mathcal{E}|d)$ | - |

Table 11: Time complexity of answering a query for methods that directly encode the query graph. Besides the operators defined in Table 10, we denote $n_q$ as average degree of the query graph $\mathcal{G}_q = (\mathcal{E}_q, \mathcal{R}_q, \mathcal{S}_q)$.

| Model | Projection | Intersection | Negation | Union | Answer Retrieval |
|---|---|---|---|---|---|
| MPQE / GNNQ | $\mathcal{O}(d^2 n_q l |\mathcal{E}_q|)$ | | - | - | $\mathcal{O}(|\mathcal{E}|d)$ |
| StarQE | $\mathcal{O}(d^2 n_q l |\mathcal{E}_q| + |\mathcal{S}_q||qp|d)$ | | - | - | $\mathcal{O}(|\mathcal{E}|d)$ |
| LMPNN | $\mathcal{O}(d^2 n_q l |\mathcal{E}_q|)$ | | | DNF | $\mathcal{O}(|\mathcal{E}|d)$ |
| BiQE | $\mathcal{O}((|\mathcal{E}_q|d^2 + |\mathcal{E}_q|^2 d)l)$ | | - | - | $\mathcal{O}(|\mathcal{E}|d)$ |
| kgTransformer | $\mathcal{O}((d^2 n_q |\mathcal{E}_q| + |\mathcal{E}_q|n_q^2 d)l)$ | | - | | $\mathcal{O}(|\mathcal{E}|d)$ |
| SQE | $\mathcal{O}((|\mathcal{E}_q|d^2 + |\mathcal{E}_q|^2 d)l)$ | | | | $\mathcal{O}(|\mathcal{E}|d)$ |

# 6 Queries

The third direction to segment the methods is from the queries point of view. Under the queries category, we have three subcategories: *Query Operators*, *Query Patterns*, and *Projected Variables*. For query operators, methods have different operator expressiveness, which means the set of query operators one model is able to handle including existential quantification (∃), conjunction (∧), disjunction (∨), negation (¬), Kleene plus (+), filter and various aggregation operators. For query patterns, we refer to the structure/pattern of the (optimized) query plan, ranging from paths and trees to arbitrary directed acyclic graphs (DAGs) and cyclic patterns. As to projected variables (by *projected* we refer to target variables that have to be bound to particular graph elements like entity or relation), queries might have a different number (zero or more) of target variables. We are interested in the complexity of such projections as binding of two and more variables involves relational algebra (Codd, 1970) and might result in a Cartesian product of all retrieved answers.

## 6.1 Query Operators

Different query processors have different expressiveness in terms of operators a method can handle. Throughout all the works, we compiled Table 12 that classifies all methods based on the supported operators.

Table 12: Query answering processors and supported query operators. Processors supporting unions ∨ also support projection and intersection (∧). Models supporting negation (¬) also support unions, projections, and intersections.

| Projection and Intersection (∧) | Union (∨) | Negation (¬) | Kleene + | Filter & Aggr |
|---|---|---|---|---|
| GQE (Hamilton et al., 2018) | Query2Box (Ren et al., 2020) | BetaE (Ren & Leskovec, 2020) | RotatE-Box | LitCQD (numbers) |
| GQE hashed (Wang et al., 2019) | EmQL (Sun et al., 2020) | ConE (Zhang et al., 2021b) | (Adlakha et al., 2021) | (Demir et al., 2023) |
| CGA (Mai et al., 2019) | Query2Onto (Andresel et al., 2021) | LogicE (Luus et al., 2021) | | |
| TractOR (Friedman & Van den Broeck, 2020) | HypE (Choudhary et al., 2021b) | MLPMix (Amayuelas et al., 2022) | | |
| MPQE (Daza & Cochez, 2020) | NewLook (Liu et al., 2021) | Query2Particles (Bai et al., 2022) | | |
| BiQE (Kotnis et al., 2021) | PERM (Choudhary et al., 2021a) | LinE (Huang et al., 2022b) | | |
| Sheaves (Gebhart et al., 2023) | CQD (Arakelyan et al., 2021) | GammaE (Yang et al., 2022a) | | |
| StarQE (Alivanistos et al., 2022) | kgTransformer (Liu et al., 2022) | NMP-QEM (Long et al., 2022) | | |
| RotatE-m, DistMult-m, | Query2Geom (Sardina et al., 2023) | FuzzQE (Chen et al., 2022) | | |
| ComplEx-m (Ren et al., 2022) | LitCQD (Demir et al., 2023) | TAR (Tang et al., 2022) | | |
| GNNQ (Pflueger et al., 2022) | BiDAG (Xu et al., 2023a) | GNN-QE (Zhu et al., 2022) | | |
| | NRN (Bai et al., 2023a) | FLEX (Lin et al., 2022) | | |
| | CQD Onto (Andresel et al., 2023) | TFLEX (Lin et al., 2023) | | |
| | | ENeSy (Xu et al., 2022) | | |
| | | QTO (Bai et al., 2023c) | | |
| | | SignalE (Wang et al., 2022) | | |
| | | LMPNN (Wang et al., 2023e) | | |
| | | NQE (Luo et al., 2023) | | |
| | | Var2Vec (Wang et al., 2023a) | | |
| | | CQD$^A$ (Arakelyan et al., 2023) | | |
| | | SQE (Bai et al., 2023b) | | |
| | | RoConE (He et al., 2023) | | |
| | | CylE (Nguyen et al., 2023b) | | |
| | | FIT (Yin et al., 2023b) | | |
| | | WFRE (Wang et al., 2023d) | | |
| | | LARK (Choudhary & Reddy, 2023) | | |
| | | SCoNe (Nguyen et al., 2023a) | | |
| | | Query2Triple (Xu et al., 2023b) | | |
| | | UnRavL (Cucumides et al., 2024) | | |
| | | UltraQuery (Galkin et al., 2024) | | |

We start from simplest conjunctive queries that involve only existential quantification (∃) and conjunction (∧) and gradually increase the complexity of supported operators.

**Existential Quantification (∃).** When ∃ appears in a query, this means that there exists at least one existentially quantified variable. For example, given a query "At what universities do the Turing Award winners work?" and its logical form $q = U_? . \exists V : win(\texttt{TuringAward}, V) \wedge university(V, U_?)$, here $V$ is the existentially quantified variable. Query processors model existential quantification by using a relation projection operator. Mapping to query languages like SPARQL, relation projection is equivalent to a *triple pattern* with one variable, *e.g.*, {TuringAward win ?v} (Fig. 13). Generally, as introduced in Section 5.2, query embedding methods embed a query in a bottom-up fashion, starting with the embedding of the anchors

(leaf nodes) and gradually traversing the query tree up to the root. In such a way, query embedding methods (*e.g.*, geometric or probabilistic) explicitly obtain an embedding for the existentially quantified variables. The embeddings/representations of these variables are calculated by a relation projection function implemented as shallow vector operations (Hamilton et al., 2018; Ren et al., 2020; Choudhary et al., 2021b; Arakelyan et al., 2021; Ren et al., 2022) or deep neural nets (Ren & Leskovec, 2020; Zhang et al., 2021b; Bai et al., 2022; Amayuelas et al., 2022). Another set of methods based on GNNs and Transformers directly assigns a learnable initial embedding for the existentially quantified variables. These embeddings are then updated through several message passing layers over the query plan (Daza & Cochez, 2020; Alivanistos et al., 2022; Pflueger et al., 2022; Wang et al., 2023e) or attention layers over the serialized query plan (Kotnis et al., 2021; Liu et al., 2022).

The major drawback of existing neural query processors is the assumption of at least one anchor entity in the query from which the answering process starts and relation projections can be executed. It remains an open challenge and an avenue for future work to support queries without anchor entities, *e.g.*, $q_1 = U_?.\exists v_1, v_2 : win(v_1, v_2) \land university(v_2, U_?)$, and queries where relations are variables, *e.g.*, $q_2 = r_? : r_?(\texttt{TuringAward}, \texttt{Bengio})$, that can be framed as the relation prediction task.

**Universal Quantification ($\forall$).** A universally quantified variable $\forall x.P(x)$ means that a logical formula $P(x)$ holds for all possible $x$. Usually, the universal quantifier does not appear in facts-only ABox graphs without types and complex axioms (see Section 4.3) as it would imply that some entity is connected to all other entities. For example, $\forall V.win(\texttt{TuringAward}, V)$ without other constraints implies that all entities are connected to $\texttt{TuringAward}$ by the *win* relation (which does not occur in practice). However, universal quantifiers are more useful when paired with the class hierarchy (unary relations), *e.g.*, $\exists ?paper, \forall r \in \texttt{Researcher} : \texttt{Researcher}(r) \land authored(r, ?paper)$ means that the *authored* relation projection would be applied only to entities of class $\texttt{Researcher}$.

Currently, existing CLQA approaches do not support universal quantifiers explicitly nor the datasets include queries with the $\forall$ quantifier. Still, using the basic identity $\forall x.P(x) \equiv \neg(\exists x.\neg P(x))$ it is possible to model the universal quantifier by any approach supporting existential quantification $\exists$ and negation $\neg$. By default, we assume the closed-world assumption. We leave the implications of universal quantification pertaining to the open-world assumption (OWA) out of the scope of this work.

**Conjunction ($\land$).** We elaborate on the differences of those query patterns in the following Section 6.2 and emphasize our focus on more complex intersection queries going beyond simpler path-like queries.

Query processors, as described in Section 5.2, employ different parameterizations of conjunctions as permutation-invariant set functions. A family of neural processors (Hamilton et al., 2018; Wang et al., 2019; Ren et al., 2022) often resort to the DeepSet architecture (Zaheer et al., 2017) that first projects each set element independently and then pools representations together with a permutation-invariant function (*e.g.*, sum, mean) followed by an MLP. Alternatively, (Mai et al., 2019; Bai et al., 2022) self-attention, can serve as a replacement of the DeepSet where set elements are weighted with the attention operation. The other family of neural processors combine projection and intersection by processing the whole query graph with GNNs (Daza & Cochez, 2020; Alivanistos et al., 2022; Pflueger et al., 2022; Wang et al., 2023e) or with the Transformer over linearized query sequences (Kotnis et al., 2021; Liu et al., 2022). Geometric processors (Ren et al., 2020; Liu et al., 2021; Choudhary et al., 2021b; Zhang et al., 2021b; He et al., 2023; Nguyen et al., 2023b) implement conjunction as the attention-based average of centroids and offsets of respective geometric objects (boxes, hyperboloids, cones, or cylinders). Probabilistic processors (Ren & Leskovec, 2020; Choudhary et al., 2021a; Huang et al., 2022b; Yang et al., 2022a) implement intersection as a weighted sum of parametric distributions that represent queries and variables.

Fuzzy-logic processors (Arakelyan et al., 2021; Luus et al., 2021; Chen et al., 2022; Zhu et al., 2022; Wang et al., 2023d; Yin et al., 2023b; Demir et al., 2023) commonly resort to *t-norms*, generalized versions of conjunctions in the continuous $[0, 1]$ space and corresponding *t-conorms* for modeling unions (Section A.4). Often, due to the absence of a principal study, the choice of the fuzzy logic is a hyperparameter. We posit that such a study is an important avenue for future works in fuzzy processors. More exotic neuro-symbolic methods for modeling conjunctions include element-wise product of count-min sketches (Sun et al., 2020) or

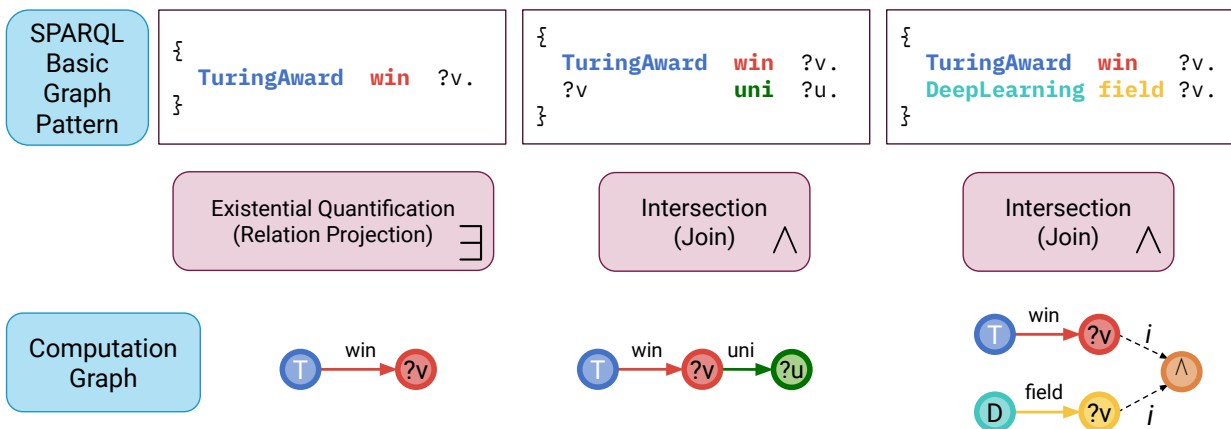

Figure 13: Query operators (relation projection and intersection), corresponding SPARQL basic graph patterns (BGP), and their computation graphs. Relation Projection (left) corresponds to a triple pattern.

as a weighted sum in the *feature logic* (Lin et al., 2022; 2023). Finally, some processors (Tang et al., 2022; Xu et al., 2022) perform conjunctions both in the embedding and symbolic space with neural and fuzzy operators.

We note that certain neural query processors that embed queries directly (Hamilton et al., 2018; Wang et al., 2019; Mai et al., 2019; Friedman & Van den Broeck, 2020; Ren et al., 2022) or via GNN/Transformer encoder over a query graph (Daza & Cochez, 2020; Kotnis et al., 2021; Gebhart et al., 2023; Alivanistos et al., 2022; Das et al., 2022; Pflueger et al., 2022) support only projections and intersections, that is, their extensions to more complex logical operators are non-trivial and might require changing the underlying assumptions of modeling entities, variables, and queries. In some cases, support of unions might be enabled when re-writing a query to the disjunctive normal form (discussed below).

**Disjunction ($\vee$).** Query processors implement the disjunction operator in several ways. However, modeling disjunction is notoriously hard since it requires modeling any powerset of entities on the graph in a vector space. Before delving into details about different ways of modeling disjunction, we first refer the readers to the Theorem 1 in Query2Box (Ren et al., 2020). The theorem proves that we need the VC dimension of the function class of the distance function to be around the number of entities on the graph.

The theorem shows that in order to accurately model *any* EPFO query with the existing framework, the complexity of the distance function measured by the VC dimension needs to be as large as the number of KG entities. This implies that if we use common distance functions based on hyper-plane, Euclidean sphere, or axis-aligned rectangle,[5] their parameter dimensionality needs to be at least $\Theta(|\mathcal{E}|)$ for real KGs. In other words, the dimensionality of the logical query embeddings needs to be $\Theta(|\mathcal{E}|)$, which is not low-dimensional; thus not scalable to large KGs and not generalizable in the presence of unobserved KG edges.

The first idea proposed in Query2Box (Ren et al., 2020) is that given a model has defined a distance function between a query representation and the entity representation, then a query can be transformed (or re-written) into its equivalent *disjunctive normal form* (DNF), *i.e.*, a disjunction of conjunctive queries. For example, we can safely convert a query $(A \vee B) \wedge (C \vee D)$ to $((A \wedge C) \vee (A \wedge D) \vee (B \wedge C) \vee (B \wedge D))$, where $A, B, C, D$ are atomic formulas. In such a way, we only need to process disjunction $\vee$ at the very last step. For models that have defined a distance function between the query representation and entity representation $d(\mathbf{q}, \mathbf{e})$ (such as geometric processors (Ren et al., 2020; Andresel et al., 2021; Adlakha et al., 2021; Choudhary et al., 2021b; Zhang et al., 2021b; Sardina et al., 2023; He et al., 2023; Nguyen et al., 2023b) and some neural processors (Liu et al., 2022; Amayuelas et al., 2022; Wang et al., 2023e), the idea of using DNF to handle disjunction is to (1) embed each atomic formula / conjunctive query in the DNF into a vector $\mathbf{q_i}$, (2)

---

[5]For the detailed VC dimensions of these function classes, see Vapnik (2013). Crucially, their VC dimensions are all linear with respect to the number of parameters $d$.

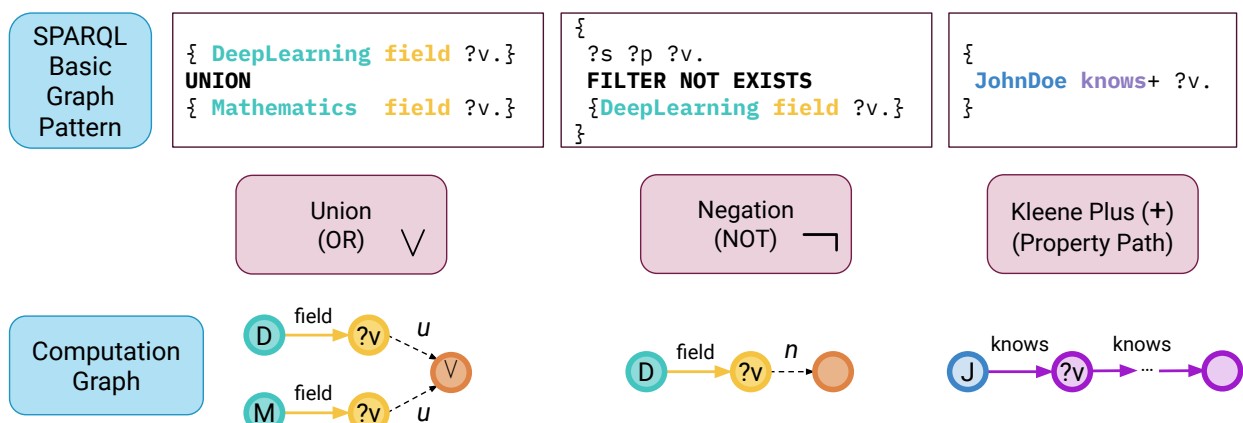

Figure 14: Query operators (union, negation, Kleene plus), corresponding SPARQL basic graph patterns (BGP), and their computation graphs.

calculate the distance between the representation/embedding of each atomic formula / conjunctive query and the entity $d(\mathbf{q_i}, \mathbf{e})$, (3) take the minimum of the distances $min_i(d(\mathbf{q_i}, \mathbf{e}))$. The intuition is that since disjunction models the union operation, as long as the node is close to one atomic formula / conjunctive query, it should be close to the whole query. Potentially, many neural processors with the defined distance function and originally supporting only intersection and projection can be extended to supporting unions with DNF. One notable downside of this modeling is that it is exponentially expensive (to the number of disjunctions) in the worst case when converting a query to its DNF.

Another category of mostly probabilistic processors (Choudhary et al., 2021a; Yang et al., 2022a) proposes a neural disjunction operator implemented with the permutation-invariant attention over the input set with the *closure* assumption that the result of attention weighting union remains in the same probabilistic space as its inputs. Such models design a more black-box framework to handle the disjunction operation under the strong closure assumption that might not be true in all cases.

The third way of modeling disjunction is based on the De Morgan's laws (Ren & Leskovec, 2020). According to the De Morgan's laws (DM), the disjunction is equivalent to the negation of the conjunction of the negation of the statements making up the disjunction, *i.e.*, $A \vee B = \neg(\neg A \wedge \neg B)$. For methods that can handle the negation operator (detailed in the following paragraph), they model disjunction by using three negation operations and one conjunction operation. DM conversion was explicitly probed in probabilistic (Ren & Leskovec, 2020), geometric (Zhang et al., 2021b), and fuzzy (Luus et al., 2021) processors.

Finally, most fuzzy-logic (Arakelyan et al., 2021; Chen et al., 2022; Tang et al., 2022; Zhu et al., 2022; Bai et al., 2023c; Arakelyan et al., 2023; Wang et al., 2023a;d; Yin et al., 2023b; Demir et al., 2023) processors employ *t-conorms*, generalized versions of disjunctions in the continuous $[0, 1]$ space (Section A.4). More exotic versions of neuro-symbolic disjunctions include element-wise summation of count-min sketches (Sun et al., 2020), feature logic operations (Lin et al., 2022; 2023), as well as performing a union in both embedding and symbolic spaces (Tang et al., 2022; Xu et al., 2022) with fuzzy operators.

**Negation ($\neg$).** For negation operation, the goal is to model the complement set, *i.e.*, the answers $\mathcal{A}_q$ to a query $q = V_? : \neg r(v, V_?)$ are the exact complement of the answers $\mathcal{A}_{q'}$ to query $q' = V_? : r(v, V_?)$: $\mathcal{A}_q = \mathcal{V}/\mathcal{A}_{q'}$. Correspondingly, negation in SPARQL can be implemented with `FILTER NOT EXISTS` or `MINUS` clauses. For example (Fig. 14), a logical formula with negation $\neg field(\texttt{DeepLearning}, \texttt{V})$ is equivalent to the SPARQL BGP `{?s ?p ?v.  FILTER NOT EXISTS {DeepLearning field ?v}}` where `{?s ?p ?v}` models the *universe set* ($\mathbf{1}$) of all facts that gets filtered by the triple pattern.

Modeling the *universe set* ($\mathbf{1}$) and its complement is the key problem when designing a negation operator in neural query processors, *e.g.*, an arbitrary real $\mathbb{R}$ or complex $\mathbb{C}$ space is unbounded such that $\mathbf{1}$ is not

defined. For that reason, many neural processors do not support the negation operator. Still, there exist several approaches to handle negation.

The first line of works (Bai et al., 2022; Amayuelas et al., 2022; Long et al., 2022) designs a purely neural MLP-based negation operator over the query representation avoiding the universe set altogether. Similarly, a token of the negation operator can be included into the linearized query representation (Bai et al., 2023b) to be encoded with Transformer or recurrent network. A step aside from purely neural operators is taken by GNN-based processors (Wang et al., 2023e) that treat a negation edge as a new edge type during message passing over the query computation graph.

The second line is customized to different embedding spaces and aims to simulate the calculation of the universe and complement in the embedding space, *e.g.*, using geometric cones (Zhang et al., 2021b; He et al., 2023) or cylinders (Nguyen et al., 2023b), parameters are angles $\theta$ such that the space (and, hence, $\mathbf{1}$) is bounded to $2\pi$ and the complement is straight $2\pi - \theta$. Probabilistic methods (Ren & Leskovec, 2020; Huang et al., 2022b; Yang et al., 2022a) naturally represent negation as an inverse of distribution parameters.

Thirdly, fuzzy logic processors explicitly model the universe set $\mathbf{1}$ and the complement over the same real valued logic space. For instance, LogicE (Luus et al., 2021), FuzzQE (Chen et al., 2022), WFRE (Wang et al., 2023d) restrict the query embedding space to the range $[0,1]^d$ where each query $\mathbf{q} \in [0,1]^d$ is a vector. This way, the universe $\mathbf{1}$ is represented with a vector of all ones (in the embedding space $\mathbf{1}^d$) and negation is simply $\mathbf{1} - \mathbf{q}$. TAR (Tang et al., 2022), GNN-QE (Zhu et al., 2022), FIT (Yin et al., 2023b) operate over fuzzy sets where each entity has a corresponding scalar $\mathbf{q} \in [0,1]$ in the bounded range. Therefore, the universe $\mathbf{1}$ can still be a vector of all ones (in the entity space $\mathbf{1}^{|\mathcal{E}|}$) and negation is $\mathbf{1} - \mathbf{q}$. ENeSy (Xu et al., 2022) defines the universe as the uniform distribution over the entity space with each element weighting $\frac{\alpha}{|\mathcal{E}|}$ ($\alpha$ is a hyperparameter). CQD$^{\mathcal{A}}$ (Arakelyan et al., 2023) employs a strict cosine fuzzy negation $\frac{1}{2}(1 + \cos(\pi\mathbf{q}))$ over scalar scores $\mathbf{q}$. More exotic processors (Lin et al., 2022; 2023) employ feature logic for modeling negation. We also note that the *difference* operator introduced in Liu et al. (2021) is in fact a common intersection-negation (*2in*) query pattern used in all standard benchmarks (Section 7).

**Kleene Plus (+) and Property Paths.** Kleene Plus is an operator that applies compositionally and recursively to any regular expression (RegEx) that denotes *one or more* occurrence of the specified pattern. Regular expressions exhibit a direct connection to *property paths* in SPARQL. We defined a very basic regular graph query in Definition A.7, here we generalize that further to property paths. To define property paths more formally, given a set of relations $\mathcal{R}$ and operators $\{+, *, ?, !, \hat{}, /, |\}$, a property path $p$ can be obtained from the recursive grammar $p ::= r \mid p^+ \mid p^* \mid p? \mid !p \mid \hat{p} \mid p_1/p_2 \mid \text{“}p_1|p_2\text{”}$ Here, $r$ is any element of $\mathcal{R}$, $+$ is a Kleene Plus denoting *one or more* occurrences, $*$ is a Kleene Star denoting *zero or more* occurrences, ? denotes *zero or one* occurences, ! denotes negation of the relation or path, $\hat{p}$ traverses an edge of type $p$ in the opposite direction, $p_1/p_2$ is a sequence of relations (corresponds to *relation projection*), and $p_1|p_2$ denotes an alternative path of $p_1$ or $p_2$ (corresponds to a *union* operation). For example (Fig. 14), an expression with Kleene plus $knows(\texttt{JohnDoe}, V)^+$ can be represented as a SPARQL property path $\{\texttt{JohnDoe knows+ ?v.}\}$.

Property paths are non-trivial to model for neural query processors due to compositionalilty and recursive nature. To the best of our knowledge, RotatE-Box (Adlakha et al., 2021) is the only geometric processor that handles the Kleene Plus operator implementing the subset of operators $\{+, /, |\}$. RotatE-Box provides two ways to handle Kleene Plus. The first method is to define a $\mathbf{r}^+$ embedding for each relation $r \in \mathcal{R}$, note this is independent and separate from the regular relation embedding for $r$; another way is to use a trainable matrix to transform the relation embedding $\mathbf{r}$ to the $\mathbf{r}^+$ embedding. Note the two methods do not support Kleene Plus over paths. RotatE-Box also implements relation projection (as a rotation in the complex space) and union (with DeepSets or DNF) but does not support the intersection operator.

We hypothesize that better support of the property paths vocabulary might be one of main focuses in future neural query processors. Particularly for Kleene Plus, some unresolved issues include supporting idempotence $((r^+)^+ = r^+)$ and infinite union of sets $(r^+ = r|(r/r)|(r/r/r)\dots)$.

**Filter.** Filter is an operation that can be inserted in a SPARQL query. It takes any expression of boolean type as input and aims to filter the results based on the boolean value, *i.e.*, only the results rendered `True`

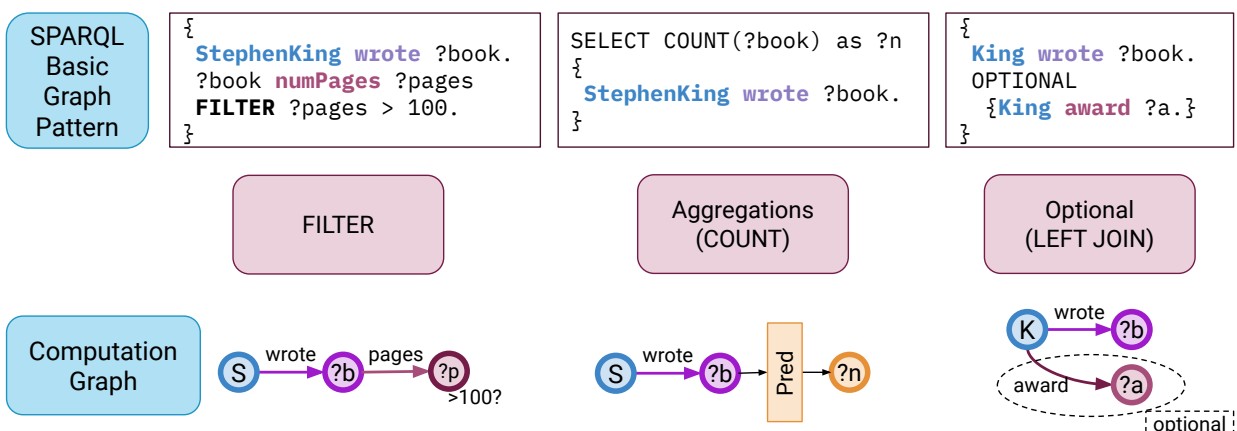

Figure 15: Query operators (`Filter`, `Count` Aggregation, `Optional`), corresponding SPARQL basic graph patterns (BGP), and their computation graphs.

under the expression will be returned. The boolean expression can thus be seen as a condition that the answers to the query should follow. For the filter operator, we can do filter on values/literals/attributes, *e.g.*, $\text{Filter}(V_{\text{date}} \geq \text{"}2000-01-01\text{"}\&\&V_{\text{date}} \leq \text{"}2000-12-31\text{"})$ means we would like to filter dates not in the year 2000; $\text{Filter}(\text{LANG}(V_{\text{book}}) = \text{"}en\text{"})$ means we would like to filter books not written in English, $\text{Filter}(?pages > 100)$ means returning the books that have more than 100 pages (as illustrated in Fig. 15). To the best of our knowledge, there does not exist a reasoning model that claims to handle all possible Filters, which leaves room for future work on this direction. However, the first attempt towards handling filters is taken by LitCQD (Demir et al., 2023) that allows **greater than**, **less than**, and **equal** filtering operators over numerical values and treats them as conjunctive terms to the main logical query, *e.g.*, *lt(?pages, 100)*. Such terms are processed by a jointly trained regression model.

We envision several possibilities to support filtering in neural query engines: (1) the simplest option used by Thorne et al. (2021a;b) in natural language engines is to defer filtering to the postprocessing stage when the set of candidate nodes is identified and their attributes can be extracted by a lookup. (2) Filtering often implies reasoning over literal values and numerical node attributes, that is, processors supporting continuous values (as described in Section 4.1) might be able to perform filtering in the latent space by attaching, for instance, a parametric regressor decoder (Section 5.3) when predicting ?pages > 100.

**Aggregation.** Aggregation is a set of operators in SPARQL queries including `COUNT` (return the number of elements), `MIN`, `MAX`, `SUM`, `AVG` (return the minimum / maximum / sum / average value of all elements), `SAMPLE` (return any sample from the set). For example (Fig. 15), given a triple pattern `{StephenKing wrote ?book.}`, the clause `COUNT (?book) as ?n` returns the total number of books written by `StephenKing`.

Most aggregation operators require reasoning over sets of numerical values/literals. Such symbolic operations have long been considered a challenge for neural models (Hendrycks et al., 2021). How to design a better representation for numerical values / literals requires remains an open question. Some neural query processors (Ren et al., 2020; Ren & Leskovec, 2020; Zhang et al., 2021b; Zhu et al., 2022), however, have the means to estimate the cardinality of the answer set (including predicted hard answers) that directly corresponds to the `COUNT` aggregation over the target projected variable (assumed to be an entity, not a literal). For example, GNN-QE (Zhu et al., 2022) returns a fuzzy set, *i.e.*, a scalar likelihood value for each entity, that, after thresholding, has low mean absolute percentage error (MAPE) of the number of ground truth answers. Answer cardinality estimation is thus obtained as a byproduct of the neural query processor without tailored predictors. Alternatively, when models cannot predict the exact count, Spearman's rank correlation is a surrogate metric to evaluate the correlation between model predictions and the exact count. Spearman's rank correlation and MAPE of the number of ground truth answers are common metrics to evaluate the performance of neural query processors and we elaborate on the metrics in Section 7.5. LitCQD (Demir

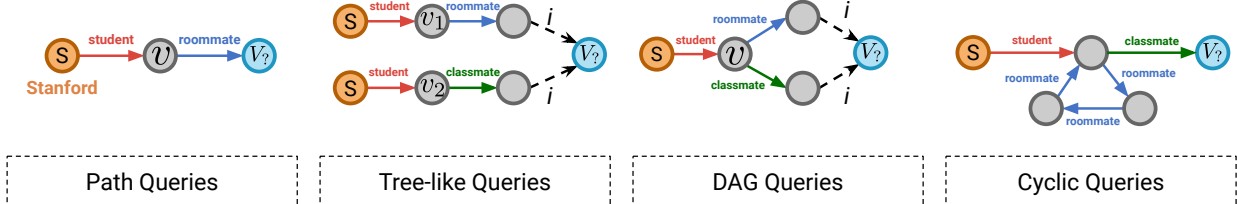

Figure 16: Query patterns: path, tree-like, DAG, and cyclic queries. A DAG query has two branches from the intermediate variable, a cyclic query contains a 3-cycle. Existing neural query processors support path and tree-like patterns.

et al., 2023) extends the answer set to numerical literals and by default implements the `AVG` aggregation of predicted values when a query has multiple correct target entities. The target metrics are typical regression metrics like mean absolute error (MAE) or mean squared error (MSE).

**Optional and Solution Modifiers.** SPARQL offers many features yet to be incorporated into neural query engines to extend their expressiveness. Some of those common features include the `OPTIONAL` clause that is essentially a `LEFT JOIN` operator. For example (Fig. 15), given a triple pattern {`King wrote ?book.`} that returns books, the optional clause {`King wrote ?book.  OPTIONAL {King award ?a.}}` enriches the answer set with any existing awards received by `King`. Importantly, if there are no bindings to the optional clause, the query still returns the values of `?book`. In the query's computation graph, the optional clause corresponds to the optional branch of a relation projection. A particular challenge for neural query processors operating on incomplete graphs is that the absence of the queried edge in the graph does not mean that there are no bindings – instead, the edge might be missing and might be predicted during query processing.

Solution modifiers, *e.g.*, `GROUP BY`, `ORDER BY`, `LIMIT`, apply further postprocessing of projected (returned) results and are particularly important when projecting several variables in the query. So far, all existing neural query processors are tailored for only one return variable. We elaborate on this matter in Section 6.3.

**A General Note on Incompleteness.** Finally, we would like to stress out that neural query engines performing all the described operators (**Projection**, **Intersection**, **Union**, **Negation**, **Property Paths**, **Filters**, **Aggregations**, **Optionals**, and **Modifiers**) assume the underlying graph is incomplete and queries might have some missing answers to be predicted, hence, all the operators should incorporate predicted *hard answers* in addition to *easy answers* reachable by graph traversal as in symbolic graph databases. Evaluation of query performance with those operators in light of incompleteness is still an open challenge (we elaborate on that in Section 7.5), *e.g.*, having an *Optional* clause, it might be unclear when there is no true answer (even predicted ones are in fact false) or a model is not able to predict them.

## 6.2   Query Patterns

Here, we introduce several types of query patterns commonly used in practical tasks and sort them in the increasing order of complexity. Starting with chain-like *Path* queries known in the literature for years, we move to *Tree-Structured* queries (the main supported pattern in modern CLQA systems). Then, we overview *DAG* and *cyclic* patterns which currently are not supported by any neural query answering system and represent a solid avenue for future work.

**Path Queries.** As introduced in Section A.3, previous literature starts with path queries (*aka* multi-hop queries), where the goal is simply to go beyond one-hop queries such as $q = V_?.r(v, V_?)$, where $r \in \mathcal{R}, v \in \mathcal{V}$ and $V_?$ represents the answer variable. As shown in Fig. 22 and Fig. 16, there is no logical operator such as branch intersection or union involved. Therefore, in order to answer such a query, we simply find or infer the neighbors of the entity $v$ with relation $r$. Path queries are a natural extension of one-hop queries. Formally, we denote a path query as follows. $q_{\text{path}} = V_?.\exists V_1, \dots, V_{k-1} : r_1(v, V_1) \wedge r_2(V_1, V_2) \wedge \cdots \wedge r_k(V_{k-1}, V_?)$, where

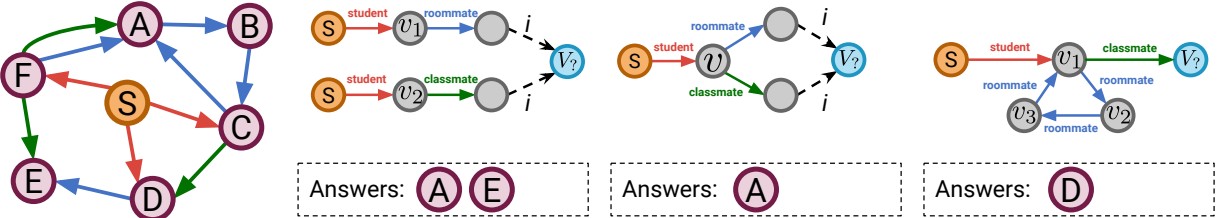

Figure 17: Answers to example tree-like, DAG, and cyclic query patterns given a toy graph. Note the difference in the answer set to the tree-like and DAG queries – in the DAG query, a variable $v$ must have two outgoing edges from the same node.

$r_i \in \mathcal{R}, \forall i \in [1, k]$, $v \in \mathcal{V}$, $V_i$ are all existentially quantified variables. We denote a $k$-hop path query if it has $k$ atomic formulas. The query plan of a $k$-hop path query is a chain of length $k$ starting from the anchor entity. For example (Fig. 16), a 2-hop path query is $V_?.\exists v : student(\texttt{Stanford}, v) \wedge roommate(v, V_?)$ where $\texttt{Stanford}$ is the starting anchor node, $v$ is a tail variable of the first projection $student$ and at the same time is the head variable of the second projection $roommate$ thus forming a chain.

As shown in the definition, in order to handle path queries, it is necessary to develop a method to handle existential quantification $\exists$ and conjunction $\wedge$ operators. Several query reasoning methods (Guu et al., 2015; Das et al., 2017) aim to answer the path queries with sequence models using either chainable KG embeddings (*e.g.*, TransE) in Guu et al. (2015) or LSTM in Das et al. (2017). These methods initiated one of the first efforts that use embeddings and neural methods to answer multi-hop path queries. We acknowledge the efforts in this domain but emphasize their limitations in terms of query expressiveness and, therefore, focus our attention in this work on more expressive query answering methods that operate on tree-like and more complex patterns.

**Tree-Structured Queries.** Path queries only have one anchor entity and one answer variable. Such queries have limited expressiveness and are far away from real-world query complexity seen in the logs (Malyshev et al., 2018) of real-world KGs like Wikidata. One direct extension to increase the expressiveness and complexity is to support tree-structured (tree-like) queries. Tree-like queries may have multiple anchor entities, and different branches (from different anchors) will merge at the final single answer node, thus forming a tree structured query plan. Such merge can be achieved by intersection, union, or negation operators. For example, as shown in Fig. 1, the query plan of "At what universities do the Turing Award winners in the field of Deep Learning work?" is not a path but a tree. Alternatively, the example in Fig. 16 depicts a query $q = V_?, \exists v_1, v_2 : student(\texttt{Stanford}, v_1) \wedge roommate(v_1, V_?) \wedge student(\texttt{Stanford}, v_2) \wedge classmate(v_2, V_?)$ that consists of two branches of 2-hop path queries joined by the intersection operator at the end.

Tree-like queries pose more challenges to the previous models that are only able to handle path (multi-hop) queries since a sequence model no longer applies to tree-structured execution plans with logical operators. In light of the challenges, neural and neuro-symbolic query processors (described in Section 5) are designed to execute more complex query patterns. These processors design neural set/logic operators and do a bottom-up traversal of the tree up to the single root node.

**Arbitrary DAGs.** Based on tree-structured queries, one can further increase the complexity of the query pattern to arbitrary directed acyclic graphs (DAGs). The key difference between the two types of queries is that for DAG-structured queries, one variable node in the query plan (that represents a set of entities) may be split and routed to different reasoning paths, while the number of branches/reasoning paths in the query plan always decreases from the anchor nodes to the answer node. We show one example in Fig. 16 and in Fig. 17. Consider the tree-like query from the previous paragraph $q_1 = V_?, \exists v_1, v_2 : student(\texttt{Stanford}, v_1) \wedge roommate(v_1, V_?) \wedge student(\texttt{Stanford}, v_2) \wedge classmate(v_2, V_?)$ and the DAG query $q_2 = V_?, \exists v : student(\texttt{Stanford}, v) \wedge roommate(v, V_?) \wedge classmate(v, V_?)$. The two queries search for $V_?$ who are *roommate* and *classmate* with $\texttt{Stanford}$ students. However, the answer sets of the two queries are different (illustrated in Fig. 17). That is, the answer to the DAG query $V_{q_2} = \{\texttt{A}\}$ is the subset of the answers to

the tree-like query $V_{q_1} = \{\mathtt{A},\mathtt{E}\}$ because the answers to $q_2$ have to be both *roommate* and *classmate* with the **same** Stanford student in the intermediate variable $v$. On the other hand, the two branches of the tree-like query $q_1$ are independent such that intermediate variables $v_1$ and $v_2$ need not be the same entities, hence, the query has more valid intermediate answers and more correct answers. To the best of our knowledge, there still does not exist a neural query processor that can faithfully handle any DAG query. Although BiQE (Kotnis et al., 2021) claims to support DAG queries, the mined dataset consists of tree-like queries. FIT (Yin et al., 2023b) processes DAG queries by rewriting and decomposing the query into fragments executable by CQD-like inference mechanism. Nevertheless, we hypothesize that, potentially, processors with message passing or Transformer architectures that consider the entire query graph structure $\mathcal{G}_q$ may be capable of handling DAG queries and leave this question for future work.

**Cyclic Queries.** Cyclic queries are more complex than DAG-structured queries. A cycle in a query naturally entails no particular order to traverse the query plan. An example of the cyclic query is illustrated in Fig. 16 and Fig. 17: $q = V_?, \exists v_1, v_2, v_3 : student(\mathtt{Stanford}, v_1) \land roommate(v_1, v_2) \land roommate(v_2, v_3) \land roommate(v_3, v_1) \land classmate(v_1, V_?)$. In $q$, three variables form a triangle cycle $roommate(v_1, v_2) \land roommate(v_2, v_3) \land roommate(v_3, V_?)$. Given a graph in Fig. 17, the cycle starts and ends at node $\mathtt{C}$, hence the only correct answer is obtained after performing the *classmate* relation projection from $\mathtt{C}$ ending in $\mathtt{D}$, $V_q = \{\mathtt{D}\}$.

Reasoning methods and query processors that assume a particular traversal or node ordering on the query plan, therefore, cannot faithfully answer cyclic queries. Yin et al. (2023b) attempt to answer cyclic queries by cutting query edges according to pre-defined decomposition rules. UnRavL (Cucumides et al., 2024) approximates cyclic queries by a set of tree-like queries with some theoretical guarantees.) It remains an open question how to effectively model a query with cyclic structures. Moreover, cyclic structures often appear when processing queries with regular expressions and *property paths* (Section 6.1). We posit that supporting cycles might be a necessary condition to fully enable property paths in neural query engines.

## 6.3 Projected Variables

By *projected variables* we understand target query variables that have to be bound to particular graph elements such as entity, relation, or literals. For example, a query in Fig. 1 $q = V_?.\exists v : win(\mathtt{TuringAward}, v) \land field(\mathtt{DeepLearning}, v) \land university(v, V_?)$ has one projected variable $V_?$ that can be bound to three answer nodes in the graph, $V_? = \{\mathtt{UofT}, \mathtt{UdeM}, \mathtt{NYU}\}$. In the SPARQL literature (Hawke et al., 2013), the `SELECT` query specifies which existentially quantified variables to project as final answers. The pairs of projected variables and answers form *bindings* as the result of the `SELECT` query. Generally, queries might have zero, one, or multiple projected variables, and we align our categorization with this notion. Examples of such queries and their possible answers are provided in Fig. 18. Currently, most neural query processors focus on the setting where queries have only one answer variable – the leaf node of the computation graph, as shown in Fig. 18 (center).

**Zero Projected Variables.** Queries with zero projected variables (Boolean queries) do not return any bindings but rather probe the graph on the presence of a certain subgraph or relational pattern where the answer is Boolean `True` or `False`. In SPARQL, the equivalent of zero-variable queries is the `ASK` clause. Zero-variable queries might have all entities and relations instantiated with constants, *e.g.*, $q = student(\mathtt{S}, \mathtt{D}) \land roommate(\mathtt{D}, \mathtt{E})$ as in Fig. 18 (left) is equivalent to the SPARQL query `ASK WHERE {S student D. D roommate E.}`. The query probes whether a graph contains a particular subgraph (path) induced by the constants. Such a path exists, so the answer is $q = \{\mathtt{True}\}$.

Alternatively, zero-variable queries might have existentially quantified variables that are never projected (up to the cases where all subjects, predicates, or objects are variables). For example, a query $q_1 = \exists v_1, v_2, v_3 : student(v_1, v_2) \land roommate(v_2, v_3)$ probes whether there exist any nodes forming a relational path $v_1 \xrightarrow{student} v_2 \xrightarrow{roommate} v_3$. In a general case, a query $q_2 = \exists p, s, o : p(s, o)$ asks if a graph contains at least one edge.

We note that in the main considered setting with incomplete graphs and missing edges zero-variable queries are still non-trivial to answer. Particularly, a subfield of *neural subgraph matching* (Rex et al., 2020; Huang

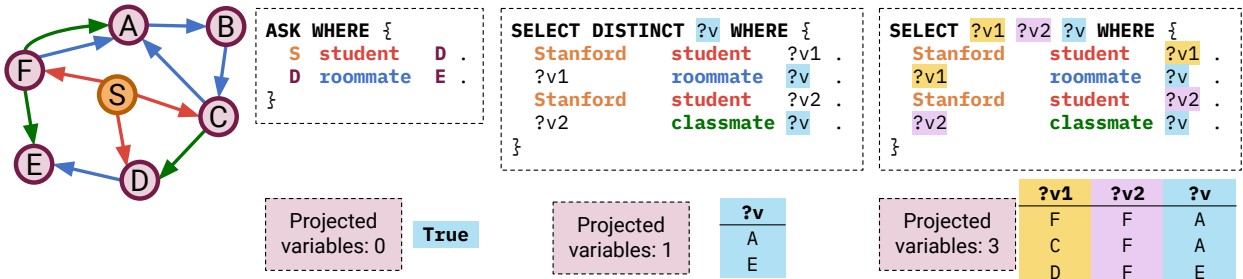

Figure 18: Projected variables of the tree-like query from Fig. 17. Current neural query processors support the single-variable `DISTINCT` mode (center) whereas queries might have zero return variables akin to a subgraph matching Boolean `ASK` query (left) or multiple projected variables (right) that imply returning intermediate answers and form output tuples.

et al., 2022a) implies having incomplete graphs. We hypothesize such approaches might be found useful for neural query processors to support answering zero-variable queries.

**One Projected Variable.** Queries with one projected variable return bindings for one (of possibly many) existentially quantified variable. In SPARQL, the projected variable is specified in the `SELECT` clause, *e.g.*, `SELECT DISTINCT ?v` in Fig. 18 (center). Although SPARQL allows projecting variables from any part of a query, most neural query engines covered in Section 5 follow the task formulation of GQE (Hamilton et al., 2018) and allow the projected target variable to be only the **leaf node** of the query computation graph. This limitation is illustrated in Fig. 18 (center) where the target variable $?v$ is the leaf node of the query graph and has two bindings $v = \{A, E\}$.

It is worth noting that existing neural query processors are designed to return a *unique set* of answers to the input query, *i.e.*, it corresponds to the `SELECT DISTINCT` clause in SPARQL. In contrast, the default `SELECT` returns *multisets* with possible duplicates. For example, the same query in Fig. 18 (center) without `DISTINCT` would have bindings $v = \{A, A, E\}$ as there exist two matching graph patterns ending in A. Implementing non-`DISTINCT` query answering remains an open challenge.

Most neural query processors have a notion of intermediate variables and model their distribution in the entity space. For instance, having a defined distance function, geometric processors (Ren et al., 2020; Choudhary et al., 2021b) can find nearest entities as intermediate variables. Similarly, fuzzy-logic processors operating on fuzzy sets (Tang et al., 2022; Zhu et al., 2022) already maintain a scalar distribution over all entities after each execution step. Finally, GNN-based (Daza & Cochez, 2020; Alivanistos et al., 2022) and Transformer-based (Liu et al., 2022) processors explicitly include intermediate variables as nodes in the query graph (or tokens in the query sequence) and can therefore decode their representations to the entity space. The main drawback of all those methods is the lack of filtering mechanisms for the sets of intermediate variables *after* the leaf node has been identified. That is, in order to filter and project only those intermediate variables that lead to the final answer, some notion of *backward pass* is required. The first step in this direction is taken by QTO (Bai et al., 2023c) that runs the pruning backward pass after reaching the answer leaf node.

**Multiple Projected Variables.** The most general and complex case for queries is to have multiple projected variables as illustrated in Fig. 18 (right). In SPARQL, all projected variables are specified in the `SELECT` clause (with the possibility to project all variables in the query via `SELECT *`). In the logical form, a query has several target variables $q = ?v_1, ?v_2, ?v : student(\text{Stanford}, ?v_1) \wedge roommate(?v_1, ?v) \wedge student(\text{Stanford}, ?v_2) \wedge classmate(?v_2, ?v)$ such that the output bindings are organized in *tuples*. For example, one possible answer tuple is $\{?v_1 : F, ?v_2 : F, ?v : A\}$ denotes particular nodes (variable bindings) that satisfy the query pattern.

As shown in the previous paragraph about one-variable queries, some neural query processors have the means to keep track of the intermediate variables. However, none of them have the means to construct answer tuples with variables bindings and it remains an open challenge how to incorporate multiple projected variables

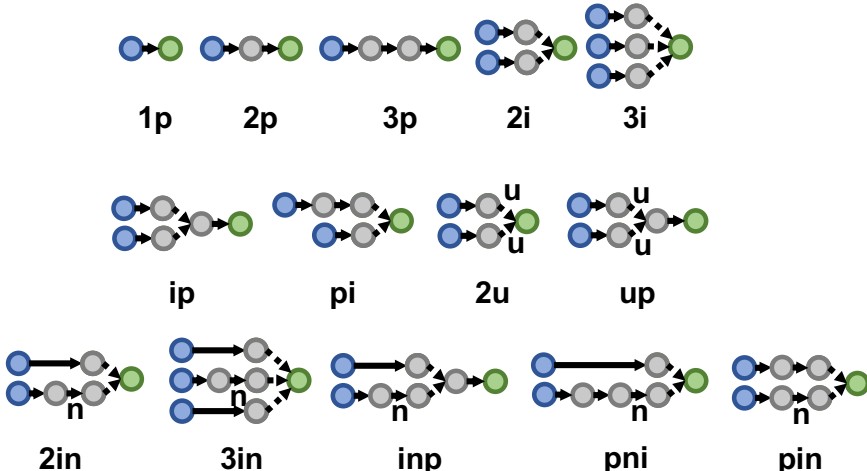

Figure 19: Standard query patterns with names, where $p$ is projection, $i$ is intersection, $u$ is union, $n$ is negation. In a pattern, blue node represents a non-variable entity, grey node represents a variable node, and the green node represents the answer node. In a typical training protocol, models are trained on 10 patterns (first and third rows) and evaluated on all patterns. In the hardest generalization case, models are only trained on *1p* queries. Some datasets further modify the patterns with additional features like qualifiers or temporal timestamps.

into such processors. Furthermore, some common caveats to be taken into account include (1) dealing with unbound variables that often emerge, for example, in `OPTIONAL` queries covered in Section 6.1, where answer tuples might contain an empty value ($\emptyset$ or `NULL`) for some variables; (2) the growing complexity issue where the answer set might potentially be polynomially large depending on the number of projected variables.

## 7 Datasets and Metrics

### 7.1 Evaluation Setup

Multiple datasets have been proposed for evaluation of query reasoning models. Here we introduce the common setup for CLQA task. Given a knowledge graph $\mathcal{G} = (\mathcal{E}, \mathcal{R}, \mathcal{S})$, the standard practice is to split $\mathcal{G}$ into a training graph $\mathcal{G}_{train}$, a validation graph $\mathcal{G}_{val}$ and a test graph $\mathcal{G}_{test}$ (simulating the unobserved complete graph $\hat{\mathcal{G}}$ from Section 2). The standard experiment protocol is to train a query reasoning model only on the training graph $\mathcal{G}_{train}$, and evaluate the model on answering queries over the validation graph $\mathcal{G}_{val}$ and the test graph $\mathcal{G}_{test}$. Given a query $q$, denote the answers of this query on training, validation and test graph as $[\![q]\!]_{\texttt{train}}$, $[\![q]\!]_{\texttt{val}}$ and $[\![q]\!]_{\texttt{test}}$. During evaluation, queries may have missing answers, *e.g.*, a validation query $q$ may have answers $[\![q]\!]_{\texttt{val}}$ that are not in $[\![q]\!]_{\texttt{train}}$, a test query $q$ may have answers $[\![q]\!]_{\texttt{test}}$ that are not in $[\![q]\!]_{\texttt{val}}$. The overall goal of CLQA task is to find these missing answers. The details of typical training queries, training protocol, inference and evaluation metrics are introduced in Section 7.2, Section 7.3, Section 7.4, and Section 7.5, respectively.

### 7.2 Query Types

The standard set of graph queries used in many datasets includes 14 types: *1p/2p/3p/2i/3i/ip/pi/2u/up/2in/3in/inp/pni/pin* where $p$ denotes relation projection, $i$ is intersection, $u$ is union, $n$ is negation, and a number denotes the number of hops for projection queries or number of branches to be merged by a logical operator. Fig. 19 illustrates common query patterns. For example, *3p* is a chain-like query of three consecutive relation projections, *2i* is an intersection of two relation projections, *3in* is an intersection of three relation projections where one of the branches contains negation, *up* is a union of two relation projections followed by another projection. The original GQE by Hamilton et al. (2018)

introduced 7 query patterns with projection and intersection *1p/2p/3p/2i/3i/ip/pi*, Query2Box (Ren et al., 2020) added union queries *2u/up*, and BetaE (Ren & Leskovec, 2020) added five types with negation.

Subsequent works modified the standard set of query types in several ways, *e.g.*, hyper-relational queries (Alivanistos et al., 2022; Luo et al., 2023) with entity-relation qualifiers on relation projections, or temporal operators on edges (Lin et al., 2023). New query patterns include queries with regular expressions of relations (property paths) (Adlakha et al., 2021), more tree-like queries (Kotnis et al., 2021), and more combinations of projections, intersections, and unions (Wang et al., 2021; Pflueger et al., 2022). We summarize existing query answering datasets and their properties in Table 14 covering supported query operators, inference setups, and additional features like temporal timestamps, class hierarchies, or complex ontological axioms.

Commonly, query datasets are sampled from different KGs to study model performance under different graph distributions, for example, BetaE datasets include sets of queries from denser Freebase (Bollacker et al., 2008) with average node degree of 18 and sparser WordNet (Miller, 1998) and NELL (Mitchell et al., 2015) with average node degree of 2. Hyper-relational datasets WD50K (Alivanistos et al., 2022) and WD50K-NFOL (Luo et al., 2023) were sampled from Wikidata (Vrandecic & Krötzsch, 2014) where qualifiers are natural. TAR datasets with class hierarchy (Tang et al., 2022) were sampled from YAGO 4 (Pellissier Tanon et al., 2020) and DBpedia (Lehmann et al., 2015) where class hierarchies are well-curated. Q2B Onto datasets with ontological axioms (Andresel et al., 2021) were sampled from LUBM (Guo et al., 2005) and NELL. Temporal TFLEX datasets (Lin et al., 2023) were sampled from ICEWS (Boschee et al., 2015) and GDELT (Leetaru & Schrodt, 2013) that maintain event information. InductiveQE datasets (Galkin et al., 2022b) were sampled from Freebase and Wikidata, while inductive GNNQ datasets (Pflueger et al., 2022) were sampled from the WatDiv benchmark (Aluç et al., 2014) and Freebase.

### 7.3 Training

Query reasoning methods are trained on the given $\mathcal{G}_{train}$ with different objectives/losses and different datasets. Following the standard protocol, methods are trained on 10 query patterns *1p/2p/3p/2i/3i/2in/3in/inp/pni/pin* and evaluated on all 14 patterns including generalization to unseen *ip/pi/2u/up* patterns. That is, the training protocol assumes that models trained on atomic logical operators would learn to compositionally generalize to patterns using several operators such as *ip* and *pi* queries that use both intersection and projection.

We summarize different training objectives in Table 13. Most methods that learn a representation of the queries and entities on the graph optimize a contrastive loss, *i.e.*, minimizing the distance between the representation of a query $q$ and its positive answers $e$ while maximizing that between the representation of a query and negative answers $e'$. Various objectives include: (1) max-margin loss (first column in Table 13) with the goal that the distance of negative answers should be larger than that of positive answers at least by the margin $\gamma$. Such loss is often of the form as the equation below.

$$\ell = \max(0, \gamma - \texttt{dist}(\mathbf{q}, \mathbf{e}) + \texttt{dist}(\mathbf{q}, \mathbf{e}'));$$

(2) LogSigmoid loss (second column in Table 13) with a similar goal that pushes the distance of negatives up and vice versa. Often the loss also includes a margin term and the gradient will gradually decrease when the margin is satisfied.

$$\ell = -\log \sigma(\gamma - \texttt{dist}(\mathbf{q}, \mathbf{e})) - \sum \frac{1}{k} \log \sigma(\texttt{dist}(\mathbf{q}, \mathbf{e}') - \gamma),$$

where $k$ is the number of negative answers. Other methods (third column in Table 13) that directly model a logit vector over all the nodes on the graph may optimize a cross entropy loss instead of a contrastive loss. Besides, methods such as the two variants of CQD (Arakelyan et al., 2021; 2023), QTO (Bai et al., 2023c), FIT (Yin et al., 2023b), and LitCQD (Demir et al., 2023) only optimize the link prediction loss since they do not learn a representation of the query.

Almost all the datasets including GQE (Hamilton et al., 2018), Q2B (Ren et al., 2020), BetaE (Ren & Leskovec, 2020), RegEx (Adlakha et al., 2021), BiQE (Kotnis et al., 2021), Query2Onto (Andresel et al., 2021), TAR (Tang et al., 2022), StarQE (Alivanistos et al., 2022), GNNQ (Pflueger et al., 2022), TeMP (Hu

Table 13: Complex Query Answering approaches categorized under *Loss*.

| Max Margin | LogSigmoid (Sun et al., 2019) | Cross Entropy |
|---|---|---|
| GQE (Hamilton et al., 2018), GQE w hash (Wang et al., 2019), CGA (Mai et al., 2019), MPQE (Daza & Cochez, 2020), HyPE (Choudhary et al., 2021b), Shv (Gebhart et al., 2023) | Query2Box (Ren et al., 2020), BetaE (Ren & Leskovec, 2020), RotatE-Box (Adlakha et al., 2021), ConE (Zhang et al., 2021b), NewLook (Liu et al., 2021), Q2B Onto (Andresel et al., 2021), PERM (Choudhary et al., 2021a), LogicE (Luus et al., 2021), MLPMix (Amayuelas et al., 2022), FuzzQE (Chen et al., 2022), FLEX (Lin et al., 2022), TFLEX (Lin et al., 2023), SMORE (Ren et al., 2022), LinE (Huang et al., 2022b), GammaE (Yang et al., 2022a), NMP-QEM (Long et al., 2022), ENeSy (Xu et al., 2022), RoMA (Xi et al., 2022), SignalE (Wang et al., 2022), Query2Geom (Sardina et al., 2023), RoConE (He et al., 2023), CylE (Nguyen et al., 2023b), WFRE (Wang et al., 2023d) | BiQE (Kotnis et al., 2021), NodePiece-QE (Galkin et al., 2022b), GNN-QE (Zhu et al., 2022), GNNQ (Pflueger et al., 2022), KGTrans (Liu et al., 2022), Query2Particles (Bai et al., 2022), EmQL (Sun et al., 2020), LMPNN (Wang et al., 2023e), StarQE (Alivanistos et al., 2022), NQE (Luo et al., 2023), CQD$^{\mathcal{A}}$ (Arakelyan et al., 2023), SQE (Bai et al., 2023b) |

et al., 2022), TFLEX (Lin et al., 2023), InductiveQE (Galkin et al., 2022b), SQE (Bai et al., 2023b) provide a set of training queries of given structures sampled from the $\mathcal{G}_{train}$. The benefit is that during training, methods do not need to sample queries online. However, it often means that only a portion of information is utilized from the $\mathcal{G}_{train}$ since exponentially more multi-hop queries exist on $\mathcal{G}_{train}$ and the dataset can never pre-generate all offline. SMORE (Ren et al., 2022) proposes a bidirectional online query sampler such that methods can directly do online sampling efficiently without the need to pre-generate a training set offline. Alternatively, methods that do not have parameterized ways to handle logical operations, *e.g.*, CQD (Arakelyan et al., 2021), only require one-hop edges to train the overall system.

## 7.4 Inference

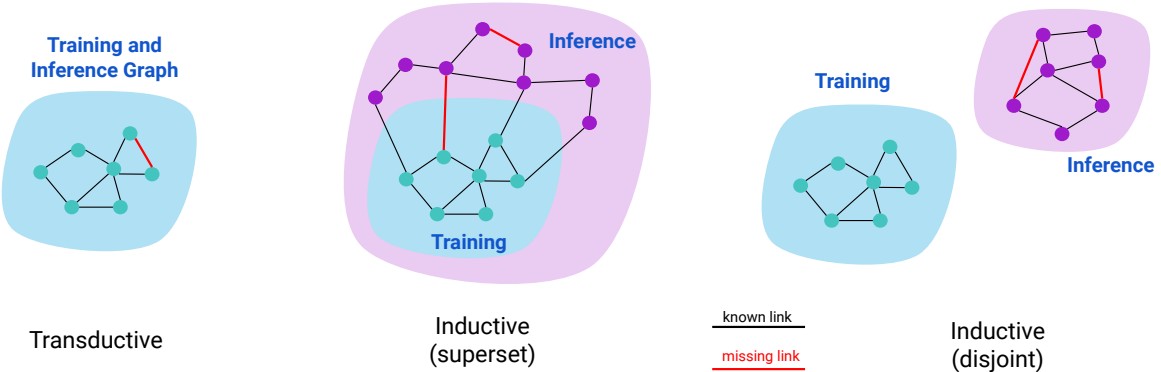

Figure 20: Inference Scenarios. In the *Transductive* case, training and inference graphs are the same and share the same nodes ($\mathcal{E}_{inf} = \mathcal{E}_{train}$). *Inductive* cases can be split into *superset* (or *semi-inductive*) where the inference graph extends the training one ($\mathcal{E}_{train} \subseteq \mathcal{E}_{inf}$, missing links cover both seen and unseen nodes) and *disjoint* (or *fully-inductive*) where the inference graph is disconnected ($\mathcal{E}_{train} \cap \mathcal{E}_{inf} = \emptyset$, missing links are among unseen nodes).

By *Inference* we understand testing scenarios on which a trained query answering model will be deployed and evaluated. Following the literature, we distinguish *Transductive* and *Inductive* inference (Fig. 20). In the transductive case, inference is performed on the graph with the same set of nodes and relation types as in training but with different edges. Any other scenario when either the number of nodes or relation types of an inference graph is different from that of the training is deemed inductive. The inference scenario plays a major role in designing query answering models, that is, transductive models can learn a shallow entity

Table 14: Existing logical query answering datasets classified along supported **query operators**, **inference** scenarios, and **domain** properties. + is partial support – some operators on literals can be treated as filters.

| Source | Dataset | Query Operators | | | | | Inference | | Domain | | | | | |
|---|---|---|---|---|---|---|---|---|---|---|---|---|---|---|
| | | Conjunctive | Union | Negation | Kleene Plus | Filter + Agg | Transductive | Inductive | Discrete | + Timestamps | + Continuous | Types | Rules | Qualifiers |
| Hamilton et al. (2018) | GQE datasets | ✓ | | | | | ✓ | | ✓ | | | | | |
| Ren et al. (2020) | Q2B datasets | ✓ | ✓ | | | | ✓ | | ✓ | | | | | |
| Ren & Leskovec (2020) | BetaE datasets | ✓ | ✓ | ✓ | | | ✓ | | ✓ | | | | | |
| Adlakha et al. (2021) | Regex queries | ✓ | ✓ | | ✓ | | ✓ | | ✓ | | | | | |
| Kotnis et al. (2021) | DAG queries | ✓ | | | | | ✓ | | ✓ | | | | | |
| Wang et al. (2021) | EFO-1 queries | ✓ | ✓ | ✓ | | | ✓ | | ✓ | | | | | |
| Andresel et al. (2021) | LUBM/NELL (type) | ✓ | ✓ | | | | ✓ | | ✓ | | | | ✓ | |
| Tang et al. (2022) | TAR datasets | ✓ | ✓ | | | | ✓ | | ✓ | | | ✓ | | |
| Ren et al. (2022) | SMORE datasets | ✓ | ✓ | | | | ✓ | | ✓ | | | | | |
| Alivanistos et al. (2022) | WD50K dataset | ✓ | | | | | ✓ | | ✓ | | | | | ✓ |
| Hu et al. (2022) | TeMP dataset | ✓ | ✓ | | | | | ✓ | ✓ | | | ✓ | | |
| Pflueger et al. (2022) | GNNQ dataset | ✓ | | | | | | ✓ | ✓ | | | | | |
| Lin et al. (2023) | TFLEX dataset | ✓ | ✓ | ✓ | | + | ✓ | | ✓ | ✓ | | | | |
| Galkin et al. (2022b) | InductiveQE dataset | ✓ | ✓ | ✓ | | | | ✓ | ✓ | | | | | |
| Luo et al. (2023) | WD50K-NFOL dataset | ✓ | ✓ | ✓ | | | ✓ | | ✓ | | | | | ✓ |
| Bai et al. (2023b) | SQE dataset | ✓ | ✓ | ✓ | | | ✓ | | ✓ | | | | | |
| Huang et al. (2022b) | WN18RR dataset | ✓ | ✓ | ✓ | | | ✓ | | ✓ | | | | | |
| Demir et al. (2023) | FB15k237 w/ literals | ✓ | ✓ | | | + | ✓ | | ✓ | | ✓ | | | |
| Yin et al. (2023b) | FIT dataset (w/ cycles) | ✓ | ✓ | ✓ | | | ✓ | | ✓ | | | | | |
| Bai et al. (2023a) | NRN datasets | ✓ | ✓ | | | | ✓ | | ✓ | | ✓ | | | |
| Cucumides et al. (2024) | Test set w/ cycles | ✓ | | | | | ✓ | | ✓ | | | | | |
| Yin et al. (2023a) | EFO$_k$ dataset | ✓ | ✓ | ✓ | | | ✓ | | ✓ | | | | | |
| Galkin et al. (2024) | WikiTopics-CLQA dataset | ✓ | ✓ | ✓ | | | | ✓ | ✓ | | | | | |

embedding matrix thanks to the fixed entity set whereas inductive models have to rely on other *invariances* available in the underlying graph in order to generalize to unseen entity/relation types. We discuss many transductive and inductive models in Section 5.2. Below, we categorize existing datasets from the *Inference* perspective. The overview of existing CLQA datasets is presented in Table 14 through the lens of supported query operators, inference scenario, graph domain, and other features like types or qualifiers.

**Transductive Inference.** Formally, given a training graph $\mathcal{G}_{train} = (\mathcal{E}_{train}, \mathcal{R}_{train}, \mathcal{S}_{train})$, the transductive inference graph $\mathcal{G}_{inf}$ [6] contains the same set of entities and relation types, that is, $\mathcal{E}_{train} = \mathcal{E}_{inf}$ and $\mathcal{R}_{train} = \mathcal{R}_{inf}$, while the edge set on $\mathcal{G}_{train}$ is a subset of that on the inference graph $\mathcal{G}_{inf}$, *i.e.*, $\mathcal{S}_{train} \subset \mathcal{S}_{inf}$. In this setup, query answering is performed on the same nodes and edges seen during training. From the entity set perspective, the prediction pattern is *seen-to-seen* – missing links are predicted between known entities.

Traditionally, KG link prediction focused more on the transductive task. In CLQA, therefore, the majority of existing datasets (Table 14) follow the transductive scenario. Starting from simple triple-based graphs with fixed query patterns in GQE datasets (Hamilton et al., 2018), Query2Box datasets (Ren et al., 2020), and BetaE datasets (Ren & Leskovec, 2020) that became de-facto standard benchmarks for query answering approaches, newer datasets include regex queries (Adlakha et al., 2021), wider set of query patterns (Kotnis et al., 2021; Wang et al., 2021; Bai et al., 2023b), entity type information (Tang et al., 2022), ontological axioms (Andresel et al., 2021), hyper-relational queries with qualifiers (Alivanistos et al., 2022; Luo et al., 2023), temporal queries (Lin et al., 2023), very large graphs up to 100M nodes (Ren et al., 2022), hierarchical

---

[6] Below we use $\mathcal{G}_{inf}$ to refer to the graphs we use during inference, it can be $\mathcal{G}_{val}$ or $\mathcal{G}_{test}$ without loss of generalization.

graphs (Huang et al., 2022b), queries with numerical literals (Demir et al., 2023; Bai et al., 2023a), or queries with cycles and multiedges (Yin et al., 2023b;a; Cucumides et al., 2024).

**Inductive Inference.** Formally, given a training graph $\mathcal{G}_{train} = (\mathcal{E}_{train}, \mathcal{R}_{train}, \mathcal{S}_{train})$, the inductive inference graph $\mathcal{G}_{inf} = (\mathcal{E}_{inf}, \mathcal{R}_{inf}, \mathcal{S}_{inf})$ is different from the training graph in either the entity set or the relation set or both. The nature of this difference explains several subtypes of inductive inference. First, the set of relations might or might not be shared at inference time, that is, $\mathcal{R}_{inf} \subseteq \mathcal{R}_{train}$ or $|\mathcal{R}_{inf} \setminus \mathcal{R}_{train}| > 0$. Most of the literature on inductive link prediction (Teru et al., 2020; Zhu et al., 2021; Galkin et al., 2022a) in KGs assumes the set of relations is shared whereas the setup where new relations appear at inference time is still highly non-trivial (Huang et al., 2022a; Gao et al., 2023; Chen et al., 2023).

On the other hand, the inference graph might be either a superset of the training graph after adding new nodes and edges, $\mathcal{E}_{train} \subseteq \mathcal{E}_{inf}$, or a disjoint graph with completely new entities as a disconnected component, $\mathcal{E}_{inf} \cap \mathcal{E}_{train} = \emptyset$ as illustrated in Fig. 20. From the node set perspective, the superset inductive inference case might contain both *unseen-to-seen* and *unseen-to-unseen* missing links whereas in the disjoint inference graph only *unseen-to-unseen* links are naturally appearing.

In CLQA, inductive reasoning is still an emerging area as it has a direct impact on the space of possible variables $\mathcal{V}$, constants $\mathcal{C}$, and answers $A$ that might now include entities unseen at training time. Several most recent works started to explore inductive query answering (Table 14). InductiveQE datasets (Galkin et al., 2022b) focus on the inductive superset case where a training graph can be extended with up to 500% new unseen nodes. Test queries start from unseen constants and answering therefore requires reasoning over both seen and unseen nodes. Similarly, training queries can have many new correct answers when answered against the extended inference graph. GNNQ datasets (Pflueger et al., 2022) focus on the disjoint inductive inference case where constants, variables, and answers all belong to a new entity set. TeMP datasets (Hu et al., 2022) focus on the disjoint inductive inference as well but offer to leverage an additional class hierarchy as a learnable *invariant*. That is, the set of classes at training and inference time does not change. The only suite of datasets for fully-inductive inference on both unseen entities and relations was introduced in UltraQuery (Galkin et al., 2024).

Inductive inference is crucial to enable running models over updatable graphs without retraining. We conjecture that inductive datasets and models are likely to be the major contribution area in the future work.

## 7.5 Metrics

Several metrics have been proposed to evaluate the performance of query reasoning models that can be broadly classified into **generalization**, **entailment**, and **query representation quality** metrics.

**Generalization Metrics.** Since the aim of query reasoning models is to perform reasoning over massive incomplete graphs, most metrics are designed to evaluate models' *generalization* capabilities in discovering missing answers, *i.e.*, $[\![q]\!]_{\text{test}} \setminus [\![q]\!]_{\text{val}}$ for a given test query $q$. As one of the first works in the field, GQE (Hamilton et al., 2018) proposes ROC-AUC and average percentile rank (APR). The idea is that for a given test query $q$, GQE calculates a score for all its missing answers $e \in [\![q]\!]_{\text{test}} \setminus [\![q]\!]_{\text{val}}$ and the negatives $e' \notin [\![q]\!]_{\text{test}}$. The model's performance is the ROC-AUC score and APR, where they rank a missing answer against at most 1000 randomly sampled negatives of the same entity type. Besides GQE, GQE+hashing (Wang et al., 2019), CGA (Mai et al., 2019) and TractOR (Friedman & Van den Broeck, 2020) use the same evaluation metrics.

However, the above metrics do not reflect the real world setting where we often have orders of magnitude more negatives than the missing answers. Instead of ROC-AUC or APR, Query2Box (Ren et al., 2020) proposes ranking-based metrics, such as mean reciprocal rank (MRR) and hits@$k$. Given a test query $q$, for each missing answer $e \in [\![q]\!]_{\text{test}} \setminus [\![q]\!]_{\text{val}}$, we rank it against all the other negatives $e' \notin [\![q]\!]_{\text{test}}$. Given the ranking $r$, MRR is calculated as $\frac{1}{r}$ and hits@$k$ is $1[r \leq k]$. This has been the most used metrics for the task. Note that the final rankings are computed only for the *hard* answers that require predicting at least one missing link. Rankings for *easy* answers reachable by edge traversal are usually discarded.

**Representation Quality Metrics.** Besides evaluating model's capability of finding missing answers, another aspect is to evaluate the quality of the learned query representation for all models. BetaE (Ren & Leskovec, 2020) proposes to evaluate whether the learned query representation can model the cardinality of a query's answer set, and view this as a proxy of the quality of the query representation. For models with a sense of "volume" (*e.g.*, differential entropy for Beta embeddings), the goal is to measure the Spearman's rank correlation coefficient and Pearson's correlation coefficient between the "volume" of a query (calculated from the query representation) and the cardinality of the answer set. BetaE also proposed to evaluate an ability to model queries without answers using ROC-AUC.

**Entailment Metrics.** The other evaluation protocol is about whether a model is also able to discover the existing answers, *e.g.*, $[\![q]\!]_{\texttt{val}}$ for test queries, that does not require inferring missing links but focuses on memorizing the graph structure (*easy* answers in the common terminology). This is referred to as faithfulness (or *entailment*) in EmQL (Sun et al., 2020). Natural for database querying tasks, it is expected that query answering models first recover *easy* answers already existing in the graph (reachable by edge traversal) and then enrich the answer set with predicted *hard* answers inferred with link prediction. A natural metric is therefore an ability to rank easy answers higher than hard answers – this was studied by InductiveQE (Galkin et al., 2022b) that proposed to use ROC-AUC as the main metric for this task.

Still, we would argue that existing metrics might not fully capture the nature of neural query answering and new metrics might be needed. For example, some under-explored but potentially useful metrics include (1) studying *reasonable* answers (in between easy and hard answers) that can be deduced by symbolic reasoners using a higher-level graph schema (ontology) (Andresel et al., 2021). A caveat in computing reasonable answers is a potentially infinite processing time of symbolic reasoners that have to be limited by time or expressiveness in order to complete in a finite time. Hence, the set of reasonable answers might still be incomplete; (2) evaluation in light of the *open-world assumption* (OWA) stating that unknown triples in the graph might not necessarily be false (as postulated by the standard *closed-world assumption* used a lot in link prediction). Practically, OWA means that even the test set might be incomplete and some high-rank predictions deemed incorrect by the test set might in fact be correct in the (possibly unobservable) complete graph. Initial experiments of Yang et al. (2022b) with OWA evaluation of link prediction explain the saturation of ranking metrics (*e.g.*, MRR) on common datasets by the performance of neural link predictors able to predict the answers from the complete graph missed in the test set. For example, saturated MRR of 0.4 on the test set might correspond to MRR of 0.9 on the true complete graph. Studying OWA evaluation in the query answering task in both transductive and inductive setups is a solid avenue for future work.

# 8 Applications

The framework of complex query answering is applied in a variety of graph-conditioned machine learning tasks. For example, SE-KGE (Mai et al., 2020) applies GQE to answer geospatial queries conditioned on numerical $\{x, y\}$ coordinates. The coordinates encoder fuses numerical representations with entity embeddings such that the prediction task is still entity ranking.

In case-based reasoning, CBR-SUBG (Das et al., 2022) is a method for question answering over KGs based on subgraph extraction and encoding. As a byproduct, CBR-SUBG is capable of answering conjunctive queries with projections and intersections. However, due to a non-standard evaluation protocol and custom synthetic dataset, its performance cannot be directly compared to CLQA models. Similarly, Wang et al. (2023b) merge a Query2Box-like model with a pre-trained language model to improve question answering performance. LEGO (Ren et al., 2021) also applies CLQA models for KG question answering. The idea is to simultaneously parse a natural language question as a query step and execute the step in the latent space with CLQA models.

LogiRec (Wu et al., 2022) frames product recommendation as a complex logical query such that source products are root nodes, combinations of multiple products form intersections, and non-similar products to be filtered out form negations. LogiRec employs BetaE as a query engine. Similarly, Syed et al. (2022) design an explainable recommender system based on Query2Box. Given a logical query, they first use Query2Box to generate a set of candidates and rerank them using neural collaborate filtering (He et al., 2017).

## 9    Summary and Future Opportunities

We proposed a deep and detailed review of Complex logical query answering (CLQA) methods based on the new taxonomy in Section 3 categorizing the existing approaches along three main areas: *Graphs*, *Modeling*, and *Queries* with their respective sub-areas. Going forward, there is still much room to unlock the full power of CLQA by addressing numerous open challenges. Adhering to the taxonomy, we summarize the challenges in three main areas: Graphs, Modeling, and Queries.

Along the **Graph** branch:

- **Modality:** Future systems need to handle a richer variety of graph structures and data types. For example, beyond conventional knowledge graphs that contain only triples of subject–predicate–object, we want to support hyper-relational graphs where edges can connect more than two nodes or have additional qualifiers. We also aim to include hypergraphs, which represent complex relationships with edges that join multiple entities simultaneously. In addition, we envision multimodal data sources where graph data is combined with textual descriptions, images, or other media. For instance, a system might incorporate a knowledge graph of people and places along with associated images and text captions, integrating all these modalities seamlessly.

- **Reasoning Domain:** Another goal is to facilitate logical reasoning and neural query answering over dynamic and continuous information within graphs. Many real-world knowledge graphs include temporal facts (such as events happening at specific times) and literal values (such as numerical measurements or textual strings). Because a large portion of a graph's information is stored as these literal values—like a numerical temperature reading or a textual label—a robust system should be able to answer queries that involve these literals. For example, the system might reason over a time-series of events to answer a query about when a particular relationship held, or interpret textual data attached to nodes to respond to a query that references an entity's description.

- **Background Semantics:** We want to incorporate complex background knowledge encoded by formal semantics and axioms. This means supporting not just direct entity-to-entity relationships but also higher-order relationships between classes of entities and their hierarchical structures. For example, the system should enable neural reasoning that respects the rules of description logics or subsets of the Web Ontology Language (OWL). In practice, this might allow the model to deduce that if an entity belongs to a particular subclass (like "Cat" within "Mammal"), it should inherit relationships and constraints from its superclass. A concrete example might be reasoning that if a graph knows "Fluffy is a Cat" and that "All Cats are Mammals," then the system can conclude "Fluffy is a Mammal." Supporting these complex axioms will allow more powerful and semantically rich reasoning over graph data.

In the **Modeling** branch:

- **Encoder:**

  We seek inductive encoders that can interpret new relationships in a knowledge graph even if they were unseen during training. For example, if a graph's schema is updated with a new relation type like "collaboratesWith," the encoder should handle queries involving this relation without requiring retraining. This capability supports two key goals: (1) Updatability: The neural database can quickly adapt to changes in the graph's schema or content. For instance, if new nodes and edges appear in a KG about scientific publications, the system can incorporate them on the fly. (2) Pretrain-Finetune Strategy: A model could be pre-trained on general graph structures and then fine-tuned to answer queries on a new, custom graph with a unique set of entity and relation types. Ultimately, we aspire to create a single CLQA model that can respond to queries about any unseen KG, regardless of the vocabulary of entities and relations it uses.

- **Processor:** We want processor networks that can execute a wide range of complex query operators similar to those in SPARQL or Cypher languages efficiently and effectively. For example, a system

should handle queries that require union, intersection, property paths, or filters over nodes and edges. Improving the sample efficiency of these neural processors is crucial—meaning the model should require less training data or time while preserving accuracy. For example, if training a model to interpret SPARQL "FILTER" operator, we want the model to learn from relatively few examples and still apply the filter operator correctly in new queries without extensive retraining.

- **Decoder:** Currently, neural query decoders typically return results as discrete graph nodes. We want to extend this to continuous outputs for queries that involve, for example, numerical attributes or categories not purely represented as nodes. For instance, if the query is "What is the average temperature of city X in July?" the decoder should produce a numeric result (like 29.5°C) rather than just pointing to a node. This ability is essential for real-world datasets where many queries involve combining discrete graph structures with numeric or categorical data.

- **Complexity:** Processor networks can face significant computational challenges, whether from the high dimensionality of their embeddings (in purely neural models) or the large number of graph nodes (in neuro-symbolic approaches). To handle massive KGs with billions of nodes and trillions of edges, we need more efficient algorithms for implementing neural logic operators (like those for union or negation) and for retrieving relevant information from the graph. For example, if a query asks about relationships among millions of nodes, the system should use optimized retrieval methods, ensuring that answering complex queries remains feasible at scale.

In the **Queries** branch:

- **Operators:** We aim to develop neural methods that can handle a wider variety of complex query operators found in declarative graph query languages. For example, a system should support the Kleene plus and star operators, which let queries match repeated patterns in a path (like "all ancestors" or "paths of any length connecting two nodes"). It should also manage property paths, where queries involve sequences of edges that fulfill certain property constraints (such as "entities connected by a path of type 'friendOf' repeated one or more times"). In addition, the system should handle filters that restrict results based on conditions, for example retrieving people from a graph who are over 30 years old and live in a specific city.

- **Patterns:** We want query answering systems that can handle patterns beyond simple tree structures. For instance, a query might involve a directed acyclic graph (DAG) pattern such as "Find topics that are prerequisites for both X and Y courses" or a cyclic graph pattern like "Identify people who follow each other on a social network." This complexity means the system must interpret and respond to queries where relationships loop back or branch in multiple directions, replicating the complexity of many real-world graph queries.

- **Projected Variables:** Instead of only returning a single final answer entity (like the name of a city), queries may require returning intermediate variables or entire tuples of variables. For example, a user might ask "Which actors and their respective directors collaborated on a film in 2020?" The answer requires projecting multiple variables—both the actor and the director—for each case. Similarly, a query might require returning a relationship as a variable, such as identifying not just the entities but also the nature of their relationship. The formalism presented in this survey supports this, and some papers present preliminary results, but none are evaluated on the quality of bindings to these intermediate variables.

- **Expressiveness:** We want systems to answer queries that go beyond simple fragments like EPFO (existential positive first-order) and EFO (existential first-order). This means supporting the union of conjunctive queries with negations (UCQ and UCQ$_{neg}$) and moving toward the expressiveness found in full database query languages. For instance, the system should handle a query like "Find all persons who are not authors of any paper in 2021 but who co-authored a paper in 2020." This query includes negation ("not authors") combined with conjunction ("and who co-authored"), illustrating advanced expressiveness.

- **Integrate KGQA:** Finally, we envision work on integrating integrating Knowledge Graph Question Answering and CLQA. KGQA systems interpret natural language questions and map them to complex graph queries. Suppose someone asks, "Where do the people who won the most prestigious prize in computer science work?". The system's capabilities will parse this question, interpret it as a graph query with appropriate operators and patterns, and execute the query on a knowledge graph that is assumed to be *complete*. Combining this with CLQA would lead to systems that can answer natural language questions with *incomplete* knowledge graphs. We elaborate on the similarities and differences between CLQA and KGQA in Appendix B.

In **Datasets** and **Evaluation**:

- We require larger, more varied benchmark datasets that reflect real-world complexity. For example, a new benchmark might include not just classic triple-based knowledge graphs but also hyper-relational graphs or graphs that combine textual and visual data. These benchmarks should present queries that use more expressive semantics—such as temporal conditions ("Which actors starred in movies after 2010?") or numeric filters ("Cities with population over one million?"). They should also incorporate a wider range of query operators (like union or intersection) and complex query patterns (such as nested queries). For instance, a robust benchmark might ask the system to handle a query like: "Find images of all institutions connected to entities that are both scientists and authors" requiring the system to interpret graph modalities, apply multiple operators, and return results from multiple data sources. Finally, we would need more insight into how difficult certain evaluation datasets really are. Currently, we cannot say whether the queries in one dataset are intrinsically harder to answer or whether it is just an artifact of how systems are designed that scores are different. This would also lead to more insight into what query answering system is expected to be better for what datasets and queries.

- Current evaluation methods often focus narrowly on whether a system predicts the correct "hard" answers. We need a more comprehensive evaluation framework that measures a system's performance across the entire query answering process. For example, this framework might include metrics that assess the efficiency of the query plan, how well the system handles uncertainty in queries, how it ranks potential answers by relevance, or how it deals with partial information. Concretely, beyond just checking if the system finds the correct city for a query like "What is the capital of France?" we might evaluate how it handles ambiguous queries ("What is the capital of a big European country near Germany?"), or how well it explains its reasoning steps. This means developing new metrics that reflect accuracy in retrieving possible answers ("hard" answers), correctness in reasoning about complex semantics, and clarity in explaining results, ensuring a more principled and nuanced evaluation of query answering systems.

Overall CLQA has demonstrated promise across a wide range of domains. In this survey we provide a detailed taxonomy to study CLQA. We envision CLQA to be among the core tasks and capabilities of the next generation graph databases.

## Acknowledgments

We thank Pavel Klinov (Stardog), Matthias Fey (Kumo.AI), Qian Li (Stanford) and Keshav Santhanam (Stanford) for providing valuable feedback on the draft. In addition, Michael wants to thank several people from the L&R group from the VU Amsterdam, including Daniel Daza, Dimitrios Alivanistos, Frank van Harmelen, Ruud van Bakel, Yannick Brunink, as well as Teodor Aleksiev (BSc. VU), Max Zwager (MSc. VU), and Patrick Koopmann (University Dresden) for discussions on various aspects of the work. Michael was partially funded by the Graph-Massivizer project, part of the Horizon Europe programme of the European Union (grant 101093202); his research was further made possible by a gift from Accenture, LLP.

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

# A  Definitions and Theoretical Foundations

## A.1  Types of Knowledge Graphs

Hyper-relational KGs generalize triple KGs by allowing edges to have relation-entity qualifiers. We define them as follows:

**Definition A.1 (Hyper-relational Knowledge Graph - derived from Alivanistos et al. (2022))**
*With $\mathcal{E}$ and $\mathcal{R}$ defined as in Definition 2.2, let $\mathfrak{Q} = 2^{(\mathcal{R} \times \mathcal{E})}$,[7] we define a hyper-relational knowledge graph $\mathcal{G} = (\mathcal{E}, \mathcal{R}, \mathcal{S})$. In this case, the edge set $\mathcal{S} \subset (\mathcal{E} \times \mathcal{R} \times \mathcal{E} \times \mathfrak{Q})$ consists of qualified statements, where each statement $s = (e_s, r, e_o, qp)$ includes $qp$ that represent contextual information for the main relation $r$. This set $qp = \{q_1, \ldots\} = \{(qr_1, qe_1), (qr_2, qe_2) \ldots\} \subset \mathcal{R} \times \mathcal{E}$ is the set of qualifier pairs, where $\{qr_1, qr_2, \ldots\}$ are the qualifier relations and $\{qe_1, qe_2, \ldots\}$ the qualifier entities.*

We note that if $\mathcal{E}$ and $\mathcal{R}$ are finite sets, then also $\mathfrak{Q}$ is finite and, there are a finite number of $(r, qp)$ combinations possible. As a consequence, we find that with these conditions, and by defining a canonical ordering over $\mathfrak{Q}$, we can represent the hyper-relational graph using first order logic by coining a new predicate $r_{qp}$ for each combination. The statement from the definition can then be written as $r_{qp}(e_s, e_o)$.

For example, one statement on a hyper-relational KG in (Fig. 2) is (`Hinton, education, Cambridge, {(degree, Bachelor)}`). The qualifier pair `{(degree, Bachelor)}` provides additional context to the main triple and helps to distinguish it from other `education` facts. If the conditions mentioned above are met, we can write this statement as $\text{education}_{\{(\texttt{degree:Bachelor})\}}(\texttt{Hinton}, \texttt{Cambridge})$.

Hyper-relational KGs can capture a more diverse set of facts by using the additional qualifiers for facts on the graph.

This concept is closely related (albeit with different theoretical foundations) to the RDF-star format introduced in Hartig et al. (2022). RDF-star explicitly separates metadata (context) from data (statements). RDF-star allows us to use a triple as the subject or object of another triple, which gives the same expressive power than hyper-relational graphs. A core difference, affecting the way data is modeled, is that the identity of a triple is solely defined by its constituents. This means that if a triple is used in two places, it is the same triple, which makes making complex statements about the triple more complicated (Fig. 21). For example, assume we want to express that `Max Welling` did his `BSc thesis` on `flat universe` in `Utrecht` and his `PhD degree` on `gravity` at the same university. With a hyper-relational graph we can make two edges. Each edge would state that Max Welling studied in Utrecht, but each would have two qualifier pairs with the corresponding topic and institute information. In RDF-star, we cannot model it with the same triple as subject of another triples, because RDF-star has no way to 'pair-up' the topic and degree level, and will instead think that the main triple has four pieces of information connected to it. The hyper-relational graphs introduced above do not have such issues. RDF-star further extends the values allowed in the object and qualifier value position to include literals, which we will discuss below in Definition A.3.

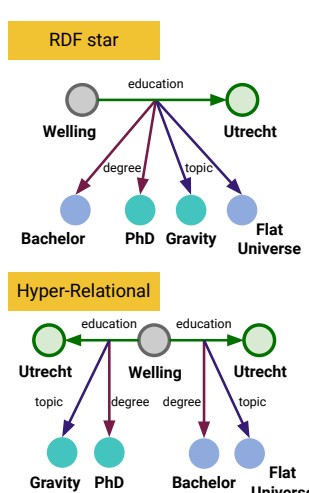

Figure 21: RDF-star and Hyper-relational Knowledge Graphs

**Definition A.2 (Hypergraph Knowledge Graph)** *With $\mathcal{E}$ and $\mathcal{R}$ defined as in Definition 2.2, a hypergraph KG is a graph $\mathcal{G} = (\mathcal{E}, \mathcal{R}, \mathcal{S})$ where statements $\mathcal{S} \subset (\mathcal{R} \times 2^{\mathcal{E}})$ are hyperedges. Such a hyperedge $s = (r, e) = (r, \{e_1, \ldots, e_k\})$ has one relation type $r$ and links the $k$ entities together, where the order of these entities is not significant. $k$, the size of $e$ is called the arity of the hyperedge.*

An example hyperedge in Fig. 2 is (`education_degree, {Hinton, Cambridge, Bachelor}`). This is a 3-ary statement (being comprised of three entities). In contrast to hyper-relational KGs, hyperedges consist of

---

[7]Given a set $S$, we use the notation $2^S$ to denote the powerset of S. Here, $\mathcal{R} \times \mathcal{E}$ denotes the Cartesian product of the set of relations and the set of entities. So, $2^{(\mathcal{R} \times \mathcal{E})}$ denotes the set containing all possible sets of pairs of a relation and an entity.

only one relation type and cannot naturally incorporate more fine-grained relational compositions. Instead, the type of the hyperedge is rather a merged set of statement relations and every composition leads to a new relation type on hypergraph KGs. It is thus not as compositional as the hyper-relational KG. That is, when describing, for instance, a `major` relation of the `education(Hinton, Cambridge)` fact, a hyper-relational KG can simply use it as a qualifier and retain both relations whereas a hypergraph model has to come up with a new hyperedge type `education_major`. With a growing number of relations, such a strategy of creating new hyperedge relation types might lead to a combinatorial explosion. However, the hypergraph model is more suitable for representing relationships of varying arity, with equal contributions of all entities such as `partnership(companyA, companyB, companyC)`.

Each of the graph types introduced above can be extended to also support literals. This happens by allowing nodes or qualifier values to contain a literal value. Literals contain numerical, categorical, discrete timestamps, or text data that cannot be easily discretized in an exhaustive set of possible values, like entities. Similarly to the Web Ontology Language (OWL, Motik et al. (2009)) that distinguishes *object properties* that connect two entities from *datatype properties* that connect an entity and a literal, it is common to use the term *relation* for an edge between two entities, and *attribute* for an edge between an entity and a literal.

**Definition A.3 (KG with Literals)** *With $\mathcal{E}$ and $\mathcal{R}$ as for one of the graph types above, a corresponding KG with literals $\mathcal{G}_{\mathcal{L}} = (\mathcal{E}, \mathcal{R}, \mathcal{S}_{\mathcal{L}}, \mathcal{L})$ has an additional set of literals $\mathcal{L} \subset$ `Con` representing numerical, categorical, textual, images, sound waves, or other continuous values ($\mathcal{L}$ is disjoint from $\mathcal{E}$ and $\mathcal{R}$). If we extend standard KGs to RDF graphs, literals can only be used in the objects position, that is $\mathcal{S}_{\mathcal{L}} \subset (\mathcal{E} \times \mathcal{R} \times (\mathcal{E} \cup \mathcal{L}))$. In RDF-star graphs, literals can be objects or qualifier values, that is, $\mathfrak{Q} = 2^{(\mathcal{R} \times (\mathcal{E} \cup \mathcal{L}))}$ and $\mathcal{S}_{\mathcal{L}} \subset (\mathcal{E} \times \mathcal{R} \times (\mathcal{E} \cup \mathcal{L}) \times \mathfrak{Q})$.[8] For both of these, we could also define graph types with literals in other positions of the triples, as necessary, or introduce more complex substructures in the elements of the triple (see e.g., Cochez (2012)). In hypergraph KGs, literals can be introduced in the elements of a hyperedge, $\mathcal{S}_{\mathcal{L}} \subset (\mathcal{R} \times 2^{\mathcal{E} \cup \mathcal{L}})$*

If the set of possible statements of the KG with literals has the same finiteness properties as the one without literals, then the properties regarding expressing it using first order logic do not change. Some graphs, like property graphs allow nodes to contain attributes, but this is equivalent with creating extra nodes with these attribute values and adding relations to those. In theory, also the edge type could be a literal, but since this can be modeled using a hyper-relational graph with literals, we exclude these from this work.

An example triple with a literal object (Fig. 5) is (`Cambridge, established, 1209`). In (`Hinton, education, Cambridge`), `education` is a relation, whereas in (`Cambridge, established, 1209`), `established` is an attribute.

The knowledge graph definitions above are not exhaustive. It is possible to create graphs with other properties. Examples include graphs with undirected edges, without edge labels, with time characteristics, with probabilistic or fuzzy relations, etc. Besides, it is also possible to have graphs or edges with combined characteristics. One could, for instance, define a hyperedge with qualifying information. Because of this plethora of options, we decided to limit ourselves to the options above. In the next section we introduce how to query these graphs.

## A.2 Basic Approximate Graph Query Answering

Until now, we have assumed that our KG is complete, *i.e.*, it is possible to exactly answer the queries. However, we are interested in the setting where we do not have the complete graph. The situation can be described as follows: Given a knowledge graph $\mathcal{G}$ (subset of a complete, but not observable graph $\hat{\mathcal{G}}$) and a basic graph query $\mathcal{Q}$. Basic Approximate Graph Query Answering is the task of answering $\mathcal{Q}$, without having access to $\hat{\mathcal{G}}$. Depending on the setting, an approach to this could have access to $\mathcal{G}$, or to a set of example (query, answer) pairs, which can be used to produce the answers (see also Section 7.3). In the example from

---

[8]Both RDF and RDF-star also allow blank nodes used to indicate entities without a specified identity in the subject and object position Brickley et al. (2014) . Besides, they also have support for named graphs (and in some cases for quadruples). We do not support these explicitly in our formalism, but all of these can be modeled using hyper-relational graphs.

Fig. 1, several edges are drawn with dashes; these are true edges, but only there in the non-observable part of the graph $\hat{\mathcal{G}}$.

The goal is to find the answer set as if $\hat{\mathcal{G}}$ was known. In this case, the complete answer set becomes $\{\mu_1, \{(\textit{?person}, \texttt{Bengio}), (\textit{?uni}, \texttt{UdeM})\}, \{(\textit{?person}, \texttt{LeCun}), (\textit{?uni}, \texttt{NYU})\}\}$, which includes the answer $\mu_1$ of the non-approximate version.

The query which includes the negation would have the following answers if our graph was complete: $\{\{(\textit{?person}, \texttt{Bengio}), (\textit{?uni}, \texttt{UdeM})\}, \{(\textit{?person}, \texttt{LeCun}), (\textit{?uni}, \texttt{NYU})\}\}$, which does not include $\mu_1$.

Approximate Query Answering Systems provide a score $\in \mathbb{R}$ for *every possible* mapping[9]. Hence, the answer provided by these systems is a function from mappings (*i.e.*, a $\mu$) to their corresponding score in $\mathbb{R}$.

**Definition A.4 (Basic Approximate Graph Query Answering)** *Given a knowledge graph $\mathcal{G}$ (sub-graph of a complete, but not observable knowledge graph $\hat{\mathcal{G}}$), a basic graph query $\mathcal{Q}$, and the scoring domain $\mathbb{R}$, a* basic approximate graph query answer *to the query $\mathcal{Q}$ is a function $f$ which maps every possible mapping ($\mu : \textit{Var}_Q \to \textit{Con}$) to $\mathbb{R}$.*

The objective is to make it such that the correct answers according to the graph $\hat{\mathcal{G}}$ get a better score (are ranked higher, have a higher probability, have a higher truth value) than those which are not correct answers. However, it is not guaranteed that answers which are in the observable graph $\mathcal{G}$ will get a high score.

For our example, a possible mapping is visualized in Table 15. Each row in the table corresponds to one mapping $\mu$. Ideally, all correct mappings should be ranked on top, and all others below. However, in this example we see several correct answers ranked high, but also two which are wrong.

Table 15: Ordered scored mappings for the example query. The two wrong mappings are in red.

| ?person | ?uni | score |
|---------|------|-------|
| Hinton | UofT | 40 |
| Bengio | UdeM | 35 |
| Welling | UofT | 34 |
| UdeM | Hinton | 33 |
| LeCun | NYU | 32 |
| . . . | . . . | . . . |

**A.3   Graph Query Answering**

In the previous sections, we introduced the basic graph query and how it can be answered either exactly, or in an approximate graph querying setting. However, there is variation in what types of queries methods can answer; some only support a subset of our basic graph queries, while others support more complicated queries. In this section we will again focus on *exact (non-approximate) query answering* and look at some possible restrictions and then at extensions.

**Definition A.5 (Graph Query Answering)** *Given a knowledge graph $\mathcal{G}$, a query formalism, and a graph query $\mathcal{Q}$ according to that formalism. An answer to the query is any mapping $\mu : \textit{Var}_Q \to \textit{Con}$ such that the requirements of the query formalism are met. The set of all answers is the set of all such mappings.*

For our basic graph queries introduced in Definition 2.4, the query formalism sets requirements as to what edges must and must not exist in the graph (Definition 2.5). In that context we already mentioned conjunctive queries, which exist either with ($\text{CQ}_{\text{neg}}$) or without (CQ) negation. If the conditions for writing our graphs using first order logic (FOL) hold, we can equivalently write our basic graph queries in first order logic. Each variable in the query becomes existentially quantified, and the formula becomes a conjunction of 1) the terms formed from the triples in $S'$, and 2) the negation of the terms formed from the triples in $\overline{S'}$. If there are variables on the edges of the query graph, then we can rewrite the second order formula in first order logic, by interpreting them as a disjunction over all possible predicates from the finite number of options. Our example query from above becomes the following FOL sentence.

$$q = \exists \textit{?person}, \textit{?uni} : \textit{win}(\texttt{TuringAward}, \textit{?person}) \land \textit{field}(\texttt{DeepLearning}, \textit{?person}) \land \textit{university}(\textit{?person}, \textit{?uni})$$

The answer to the query is the variable assignment function. For several of the more restrictive fragments, it is useful to formulate the queries using this FOL syntax.

---

[9]This score can indicate the rank of the mapping, a likelihood or a truth value, or be a binary indicating value. In some cases, only a partial mapping is provided. Anything not in this mapping has an implicit (zero or false) value.

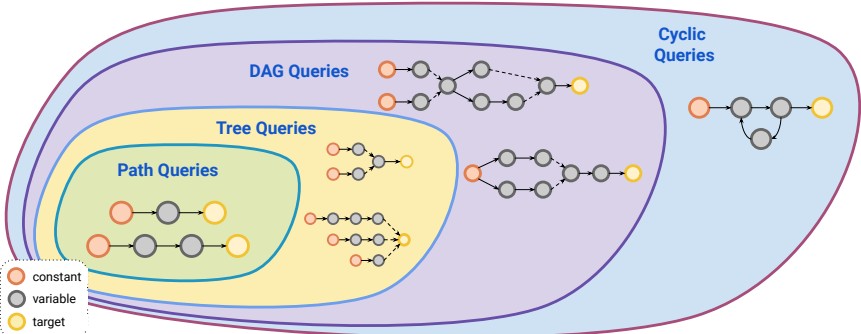

Figure 22: A space of query patterns and their relative expressiveness. Multi-hop reasoning systems tackle the simplest *Path* queries. Existing complex query answering systems support *Tree* queries whereas *DAG* and *Cyclic* queries remain unanswerable by current neural models.

**Restrictions**   The first restriction one can introduce are **multi-hop queries**, known in the NLP and KG literature for a long time (Guu et al., 2015; Das et al., 2017; Asai et al., 2020) mostly in the context of question answering. Formally, multi-hop queries (or *path* queries) are CQ which form a chain, where the tail of one projection is a head of the following projection, *i.e.*,

$$q_{\text{path}} = V_k, \exists V_1, \dots, V_{k-1} : r_1(v, V_1) \wedge r_2(V_1, V_2) \wedge \cdots \wedge r_k(V_{k-1}, V_k)$$

where $v \in \mathcal{E}$, $\forall i \in [1, k] : r_i \in \mathcal{R}, V_i \in \texttt{Var}$ and all $V_i$ are existentially quantified variables. In other words, path queries do not contain branch intersections and can be solved iteratively by fetching the neighbors of the nodes. One could also define multi-hop queries which allow negation.

Other ways of restricting CQ and CQ$_{\text{neg}}$, resulting in more expressive queries than the multi-hop ones exist. One can define families of logical queries shaped as a *Tree*, a *DAG*, and allowing *cyclic* parts. Illustrated in Fig. 22, path (multi-hop) queries form the least expressive set of logical queries. Tree queries add more reasoning branches connected by intersection operators, DAG queries drop the query tree requirement and allow queries to be directed acyclic graphs, and, finally, cyclic queries further drop the acyclic requirement. Note that these queries do not allow variables in the predicate position. Besides, all entities (in this context referred to as anchors) must occur before all variables in the topological ordering.

We elaborate more on these query types in Section 6.2 and note that the majority of surveyed neural CLQA methods in Section 5 are still limited to Tree-like queries, falling behind the expressiveness of many graph database query languages. Bridging this gap is an important avenue for future work.

**Extensions**   The first extension we introduce is the **union**.

**Definition A.6 (Union of Sets of Mappings)** *Given two sets of mappings (like μ from Definition 2.5), we can create a new set of mappings by taking their union. This union operator is commutative and associative, we can hence also talk about the union of three or more mappings. It is permissible that the domains of the mappings in the input sets are not the same.*

We can define a new type of query by applying the union operation on the outcomes of two or more underlying queries. If these underlying queries are basic graph queries, we will call this new type of queries **Unions of Conjunctive Queries with negation (UCQ$_{\text{neg}}$)** and if the basic queries did not include negation, as **Union of Conjunctive Queries (UCQ)**. These classes are also familiar from FOL, and indeed correspond to a disjunction of conjunctions.

As an example, the following query is in $\text{UCQ}_{\text{neg}}$ because it is a disjunction of conjunctive terms which consist of atoms $a_i$ that are relation projections or their negations:

$$q = v_?.\exists v_1, \ldots, v_n : \left( \underbrace{r_1(c, v_1)}_{a_1} \wedge \underbrace{r_2(v_1, v_2)}_{a_2} \right) \vee \left( \underbrace{\neg r_3(v_2, v_3)}_{a_3} \right) \vee \cdots \vee \underbrace{r_k(v_n, v_?)}_{a_m}$$

Moreover, there are FOL fragments that are equivalent to these fragments. Specifically, all queries in **EPFO**, which are Existential Positive First Order sentences, have an equivalent query in UCQ, and all queries in **EFO**, Existential First Order sentences, have an equivalent query in $\text{UCQ}_{\text{neg}}$. The reason is that EPFO and EFO sentences can be written in the Disjunctive Normal Form (DNF) as a union of conjunctive terms.

Some query languages, like SPARQL, allow an optional part to a query. In our formalism, we can define the **optional** part using the union operator. Assuming there are $n$ optional parts in the query, create $2^n$ different queries, in which other combinations of optional parts are removed. The answer to the query is then the union of the answers to all those queries. If the query language already allowed unions, then optionals do not make it more expressive.

Beyond these extensions, one could extend further to **all queries one could express with FOL**, which requires either universal quantification, or negation of conjunctions. These are, however, still not all possible graph queries. An example of interesting queries, which are not in FOL, are **conjunctive regular path queries**. These are akin to the path queries we discussed above, but without a specified length.

**Definition A.7 (Regular Path Query)** *A regular path query is a 3-tuple $(s, R, t)$, where $s \in$ `Term` the starting term of the path, $R \in$ `Term` the relation term of the path, and $t \in$ `Term` the ending term of the path. The query represents a path starting in $s$, traversing an arbitrary number of edges of type $R$ ending in $t$.*

Because this kind of query is a conjunction of an arbitrary length, it cannot be represented in FOL. If one wants to express paths with a fixed length, this would be a multi-hop path like the one described above. If one wants to express a maximum length, then this could be done using a union of all allowed lengths. For the two latter cases, the query can still be expressed in EPFO.

**Definition A.8 (Regular Path Query Answering)** *Given a knowledge graph $\mathcal{G}$ and a regular path query $\mathcal{Q} = (s, R, t)$. An answer to the query is any mapping $\mu : \textbf{Var}_Q \to \textbf{Con}$ , such that if we replace all variables $v$ in $\mathcal{Q}$ with $\mu(v)$, obtaining $(\hat{s}, \hat{R}, \hat{t})$, there exists a path in the graph that starts at node $\hat{s}$, then traverses one or more edges of type $\hat{R}$, and ends in $\hat{t}$. The set of all answers to the query is the set of all such mappings.*

There exist several variations on regular path queries and they can also be combined with the above query types to form new ones. In Fig. 23 we illustrate how the fragments relate to other query classes. Most methods fall strictly within the EFO fragment, *i.e.*, they only support a subset with restrictions as we discussed above. We will discuss further limitations of these methods in Section 6.

The two final aspects we want to highlight here are *projections* and *joins*.

**Definition A.9 (Projection of a Query Answer)** *Given a query answer $\mu$, and a set of variables $\mathcal{V} \in$ `Var`. The projection of the query answer on the variables $\mathcal{V}$ is $\{(var, val)|(var, val) \in \mu, var \in \mathcal{V}\}$.*

In other words, it is the answer but restricted to a specific set of variables. The query forms introduced above can all be augmented with a projection to only obtain values for specific variables. This corresponds to the `SELECT ?var` clause in SPARQL. Alternatively, it is possible to project all variables in $\mathcal{V}$ which is equivalent to the `SELECT *` clause. A query without any projected variable is a Boolean subgraph matching problem equivalent to the `ASK` query in SPARQL.

A join is used to combine the results of two separate queries into one. Specifically,

**Definition A.10 (Join of Query Answers)** *Given two query answers $\mu_A$ and $\mu_B$, and $\textbf{Var}_A$, and $\textbf{Var}_B$ the variables occurring in $\mu_A$ and $\mu_B$, respectively. The join of these two answers only exists in case they have the same value for all variables they have in common, i.e., $\forall var \in \textbf{Var}_A \cap \textbf{Var}_B : \mu_A(var) = \mu_b(var)$. In that case, $\text{join}(\mu_A, \mu_B) = \mu_A \cup \mu_B$.*

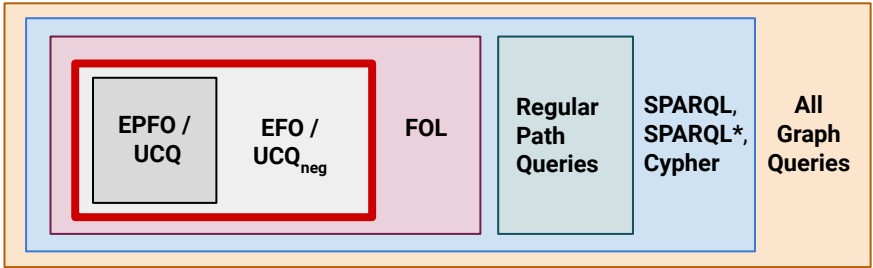

Figure 23: Current query answering models cover Existential Positive First Order (EPFO) and Existential First Order (EFO) logic fragments (marked in a red rectangle). EPFO and EFO are equivalent to unions of conjunctions (UCQ), and those with atomic negation ($UCQ_{neg}$), respectively. These, in turn, are a subset of first order logic (FOL). FOL queries, in turn, are only a subset of queries answerable by graph database languages. For example regular path queries cannot be expressed in FOL. Languages like SPARQL, SPARQL*, or Cypher, encompass all the query types and more.

Given two sets of answers, their join is defined as follows.

**Definition A.11 (Join of Query Answer Sets)** *Given two sets of query answers A and B, the join of these two is a new set* $\mathrm{join}(A, B) = \{\mathrm{join}(a, b) | a \in A, b \in B, and\ \mathrm{join}(a, b)\ exists\}$.

This operation enables us to combine multiple underlying queries, potentially of multiple types into a single one. For example, given the set of answers from our example basic graph query above:

$$A = \{\{(\mathit{?person}, \mathtt{Hinton}), (\mathit{?uni}, \mathtt{UofT})\}, \{(\mathit{?person}, \mathtt{Bengio}), (\mathit{?uni}, \mathtt{UdeM})\}, \{(\mathit{?person}, \mathtt{LeCun}), (\mathit{?uni}, \mathtt{NYU})\}\}$$

and another set of answers

$$B = \{\{(\mathit{?person}, \mathtt{Hinton}), (\mathit{?born}, \mathtt{1947})\}, \{(\mathit{?person}, \mathtt{Bengio}), (\mathit{?born}, \mathtt{1964})\}, \{(\mathit{?person}, \mathtt{Welling}), (\mathit{?born}, \mathtt{1968})\}\}$$

The join of these becomes:

$$\mathrm{join}(A, B) = \{\{(\mathit{?person}, \mathtt{Hinton}), (\mathit{?uni}, \mathtt{UofT}), (\mathit{?born}, \mathtt{1947})\}, \{(\mathit{?person}, \mathtt{Bengio}), (\mathit{?uni}, \mathtt{UdeM}), (\mathit{?born}, \mathtt{1964})\}\}$$

We will discuss joins further in Section 6.1, where we will use these basic building blocks to define a broader set of query operators, aiming to cover all operations that exist in SPARQL. This includes Kleene plus/star (+/*) for building property paths, FILTER, OPTIONAL, and different aggregation functions.

### A.4 Triangular Norms and Conorms

Answering logical queries implies execution of logical operators. Approximate query answering, in turn, implies continuous vector inputs and output truth values that are not necessarily binary. Besides, the methods often require that the logical operators are smooth and differentiable. Triangular norms (T-norms) and triangular conorms (T-conorms) define functions that generalize logical conjunction and disjunction, respectively, to the continuous space of truth values and implement fuzzy logical operations.

T-norm defines a continuous function $\top : [0, 1] \times [0, 1] \rightarrow [0, 1]$ with the following properties $\top(x, y) = \top(y, x)$ (commutativity), $\top(x, \top(y, z)) = \top(\top(x, y), z)$ (associativity), and $y \leq z \rightarrow \top(x, y) \leq \top(x, z)$ (monotonicity). Also, the identity element for $\top$ is 1, *i.e.*, $\top(x, 1) = x$. The goal of t-norms is to generalize logical conjunction with a continuous function. The T-conorm can be seen as a duality of a t-norm that similarly defines a function $\bot$ with the same domain and range $\bot : [0, 1] \times [0, 1] \rightarrow [0, 1]$. T-conorms use the continuous function $\bot$ to generalize disjunction to fuzzy logic. The function $\bot$ satisfies the same commutativity, associativity, and monotonicity properties as $\top$, but with 0 as the identity element, *i.e.*, $\bot(x, 0) = x$.

There exist many triangular norms, conorms, and fuzzy negations (Klement et al., 2013; van Krieken et al., 2022) that stem from corresponding logical formalisms, *e.g.*, (1) *Gödel logic* defines t-norm: $\top_{\min}(x, y) = \min(x, y)$, t-conorm: $\bot_{\max}(x, y) = \max(x, y)$; (2) *Product logic* with t-norm: $\top_{\text{prod}}(x, y) = x \cdot y$, t-conorm: $\bot_{\text{prod}}(x, y) = x + y - x \cdot y$; (3) in the *Łukasiewicz logic* t-norm: $\top_{\text{Łuk}}(x, y) = \max(x + y - 1, 0)$, t-conorm: $\bot_{\text{Łuk}}(x, y) = \min(x + y, 1)$. Using fuzzy negation, $N(x) = 1 - x$, one can verify that $\bot(x, y) = N(\top(N(x), N(y)))$ (De Morgan's laws) naturally obtaining a pair of $(\top, \bot)$.

### A.5 Graph Representation Learning

Graph Representation Learning (GRL) is a subfield of machine learning aiming at learning low-dimensional vector representations of graphs or their elements such as single nodes (Hamilton, 2020). For example, $\mathbf{h}_v \in \mathbb{R}^d$ denotes a $d$-dimensional vector associated with a node $v$. Conceptually, we want nodes that share certain structural or semantic features in the graph to have similar vector representations (where similarity is often measured by a distance function).

**Shallow Embeddings** The first GRL approaches focused on learning shallow node embeddings, that is, learning a unique vector per node directly used in the optimization task. For homogeneous (single-relation) graphs, DeepWalk (Perozzi et al., 2014) and node2vec (Grover & Leskovec, 2016) trained node embeddings on the task of predicting walks in the graph whereas in multi-relational graphs TransE (Bordes et al., 2013) trained node and edge type embeddings in the autoencoder fashion by reconstructing the adjacency matrix.

**Graph Neural Networks** The idea of *graph neural networks* (GNNs) (Scarselli et al., 2008) implies learning a network encoder with shared parameters on top of given (or learnable) node features by performing neighborhood aggregation. This framework can be generalized to *message passing* (Gilmer et al., 2017) where at each layer $t$ a node $v$ receives messages from its neighbors (possibly adding edge and graph features), aggregates the messages in a permutation-invariant way, and updates the representation:

$$\mathbf{h}_v^{(t)} = \text{UPDATE}\Big(\mathbf{h}_v^{(t-1)}, \text{AGGREGATE}_{u \in \mathcal{N}(v)}\big(\text{MESSAGE}(\mathbf{h}_v^{(t-1)}, \mathbf{h}_u^{(t-1)}, \mathbf{e}_{uv}))\big)\Big)$$

Here, $\mathbf{h}_u$ is a feature of the neighboring node $u$, $\mathbf{e}_{uv}$ is the edge feature, the MESSAGE function builds a message from node $u$ to node $v$ and can be parameterized with a neural network. As the set of neighbors $\mathcal{N}(v)$ is unordered, AGGREGATE is often a permutation-invariant function like *sum* or *mean*. The UPDATE function takes the previous node state and aggregated messages of the neighbors to produce the final state of the node $v$ at layer $t$ and can be parameterized with a neural network as well.

Classical GNN architectures like GCN (Kipf & Welling, 2017), GAT (Velickovic et al., 2018), and GIN (Xu et al., 2019) were designed to work with homogeneous, single-relation graphs. Later, several works have developed GNN architectures that work on heterogeneous graphs with multiple relations (Schlichtkrull et al., 2018; Vashishth et al., 2020; Zhu et al., 2021). GNNs and message passing paved the way for *Geometric Deep Learning* (Bronstein et al., 2021) that leverages symmetries and invariances in the input data as inductive bias for building deep learning models.

## B CLQA vs KGQA

KG-based question answering (KGQA) (Chakraborty et al., 2021) is the adjacent sub-field tackling natural language questions and multi-hop reasoning over structured data. The input of a typical KGQA model is a question posed in natural language, and, given a background graph, the output is a set of possible answer entities. While CLQA and KGQA are similar in terms of tackling complex multi-hop questions (and perhaps spanning over several modalities), KGQA approaches are rather different for the following reasons:

- Most often, the graphs in KGQA tasks are assumed to be *complete* and do not require predicting any missing links at inference time. This assumption significantly simplifies the prediction task and

enables two main branches of KGQA models, *i.e.*, (1) semantic parsing from a natural language question to a SPARQL query (Keysers et al., 2020) where the main challenge is compositional generalization to unseen combinations of relations; (2) retrieval-based LLM approaches (Yasunaga et al., 2021; Baek et al., 2023; He et al., 2024) where the main challenge is to maximize the recall of $k$ extracted candidate subgraphs and re-rank those to predict the answers. Some approaches that support predicting missing links on the fly rely on pre-trained shallow KG embeddings (Saxena et al., 2020) which, essentially, perform a *1p* relation projection step. As retrieval-based techniques are better amenable to other modalities, extending KGQA to support images and scene graphs is also possible (Wang et al., 2023c).

- The majority of KGQA benchmarks include only projection (such as *2p*) and intersection queries (*2i / 3i*) thus missing out on unions and negations.

- Since the query is posed in natural language, KGQA pipelines often rely on entity linking modules tailored to a specific graph (to map a string to an entity ID in the graph). Besides being an external component with possible error propagation, it hampers generalization capabilities of the models, *e.g.*, a KGQA pipeline with a tailored entity linker cannot generalize to unseen graphs. In contrast, the language-agnostic nature of CLQA allows to train graph reasoning models that can generalize to complex queries over completely unseen KGs at inference time (Galkin et al., 2024).

We believe that the synergy between CLQA and KGQA is achievable by combining the best from the both worlds, *e.g.*, inductive and generalizable reasoning over complex queries while predicting missing links (from CLQA), and deriving logical forms from the questions along with efficient retrieval mechanisms for faster scoring of relation projections (from KGQA).

