# OpenReview forum: "Neural Graph Reasoning: A Survey on Complex Logical Query Answering"
_TMLR — Accepted by TMLR_

### Review · Reviewer_dRok · 2024-06-07

**Summary Of Contributions:**

This paper surveys recent work on neural methods for complex query answering. It
captures the main aspects of (i) graph / data models, (ii) neural query
processing, (iii) query types, (iv) evaluation.

Generally, the survey is comprehensive and fills a gap in the literature. I feel
that such a survey is a valueable contribution to TMLR, and I'd ultimately would
like to see it accepted. At present, however, I feel that the survey needs
revision in two key directions: (W1) presentation and (W2) discussion.

**Audience:**

Yes

**Claims And Evidence:**

Yes

**Requested Changes:**

Address W1; W2+W3 to the extent feasible

**Strengths And Weaknesses:**

I'll briefly state the strengths/weaknesses of this survey, and will then focus
more on areas that I feel require work.

S1. Comprehensive: captures the quite large amount of recent work in this area

S2. Timely: this is a highly active reserach field at the moment

S3. In need: there is no such survey, as far as I know

S4. Useful taxonomy

S5. Many useful overview tables

W1. Presentation: let down by suboptimal structure/focus and, partly, lack of intuition

W2. Discussion: rarely provides thoughtful discussion or criticism

W3. Related areas ignored

On W1. There are three main issues with the presentation.

(i) The survey lacks focus. Most notably, there is a more than 15-page long disucssion on preliminaries, which is largely not needed (and especially in this level of formality) for the later chapters. In contrast, the discussion of the key techniques (Ch. 5) and training (Ch. 7.3) falls short. The amount of space allocated to a topic should match its relevance for this survey.

(ii) The structure of the presentation is not optimal. The presentation is strictly sequential (first all kinds of graph models, then all kinds of neural models, then all kinds of query models, then training/evaluation). While this may make sense at first glance, this makes the survey quite painful to read and quite repetitive. For example, when discussing neural methods in Ch. 5, the dicussion is very hard to follow without some of the notions introduced in Ch. 6 (e.g., that the models represent partial query results as embeddings). I feel that the discussion makes sense when the key points are already known, but it otherwise quite inaccessible. I feel that Ch. 2 can mostly go and Ch. 4--7 need to be more tightly integrated (and hence rewritten). It may help to start with plain KGs + tree-graphs and then expand from there, for example, or to start off with a much higher-level discussion of the key points for neural query processing before delving into details (e.g., in a new overview chapter with forward pointers).

(iii) Intuition party lacking. The paper quite often describes what is done, but does not convery the intuition behind the approaches or why certain things are done in certain ways. Although this is often clear to me (as I know some of the literature), it is not cleary exposed in the paper. As a simple example: why are relations translated by addition? Why does Query2Box use boxes and what do these boxes represent?

On W2. Many of the methods are presented in a descriptive style, but there is little comparitive discussion and little criticism and unclear takeways. After reading the survey, one wonders how all the methods compare to each other, what their respective strengths and weaknesses are, and where the general limitations lie. I understand that this is immensely difficult to do, but I do feel that a survey should strive for at least some insightful dicussion or identify limitations.

On W3. This work is related to KGQA (knowledge graph question answering, in particular, textual queries/features) and link prediction (single-hop queries). Many of the issues raised in this work also appear in these related areas (e.g., handling unseen entities or textual features such as entity names/descriptions). It's clear that this survey does (and needs to) focus on CQA, but it still should work out connections to these related lines of work.

Detailed comments:

D1. In principle, single-hop and rule-based methods can be used to "complete" the graph and perform CQA then. Clearly, this is limited but should be mentioned / related to CQA.

D2. The description of graphs is overly formal. That's not at all needed for the purpose of this survey.

D3. Hyper-Relational KGs and Hypergraph KGs are quite esoteric, as the tables of works that use them given in the survey also show. They have way too much focus in the presentation. (Also, I wonder about the concept of a "main relation" instead of simply using k-ary relations.)

D4. I suggest to always present intuition/high-level ideas before the details. E.g., for Def 2.7., first say that a query is a positive pattern and a negative pattern, then give the definition.

D5. Var_Q as defined on p. 7 does not contain the variables in the relation slots.

D6. The discusson of query expressiveness/types is scattered throughout chapter 2, but then also arises later on again. I feel that it would help to define the noteion of a query answer (exact, approximate) ONCE and generally; right now it's defined multiple times (e.g., Def 2.9 and 2.17) and answer and query types are mingled together.

D7. The answer semantics seems to use the CWA assumption, but it's not mentioend discussed. This matter in particular since the assumption is then broken later on.

D8. The query processing problem is not really well defined. Especially, G and Ghat clearly need to be related in some way. This is even more relevant when queries have negations, since Ghat can invalidate any query result on G.

D9. There is no diucssion around why approximate methods procude scores. Also, some methods may not; e.g., certain generative methods, and the practical implications of this choice ar not discussed. (It's clear that this is how the methods currently work, but a survey should provide context and point out potential limitations.)

D10. The mix of query languages (SPARQL, FOL, relation projections later on) is quite confusing.

D11. Approximate query answering raises all kind of questions around semantics and consistency. I think a key point is that there is often no well-defined semantics and answers can be inconsistent across queries, the methods just procude scores along Def. 2.17. This is very different to, say, probabilistic databases.

D12. The t-norm discusison is only used once much later, and would be better placed there.

D13. Multi-modal QA has been addressed in, say, the NLP and CV communities. It's weird that this field is empty in Tab. 2.

D14. TAb 2/3/4 do not feel particularly insightful, as few methods actually do this. The discussion around these tables feels more like a side note, but is placed very prominently before the actual neural methods are discussed. I'd prefer a survey that puts key points first and potential extensions/directions later.

D15. Fig 11: use subfigures, the boxes are hard to spot.

D16. The difference between "shallow embedding" and "transductive encoder" is unclear to me.

D17. Generally, it does not become sufficently clear in Fig. 11/Sec. 5.1 what the output of the encoder actually is (or can be). This relates to point W1(ii).

D18. Tables like Tab 5 are hard to read. I'd find it more insightful if rows were methods and columns properties. This way, multiple tables (e.g., Tab 5 and Tab 6) could be integrated and provide a more holistic overview of the considered methods.

D19. More generaelly, the methods are often mentioned "in passing", but there is no accessible general overview.

D20. Downside (1) of shallow methods is unclear to me.

D21. The distinction between neural and neural-symbolic processors is not clear to me.

D22. Tables on top, not in text, improve readability.

D23. The survey argues that union and negation are not well-defiend in latent space, hence many methods only support relation projection and intersection. But are those well-defined in latent space? How?

D24. Fig 12 (esp. left) is confusing and not sufficiently discussed in text.

D25. When talking about neural processors, relation projection and, say, conversion to DNF have not been discussed yet, but are already used throughout. This is an example of suboptimal structure.

D26. Tab 7, DistMult-m/ComplEx-m -- relation projection correct? (L2Norm would destroy the embedding and produce a single real)

D27. In Tab 7, cklickable refs + conference would be helpful.

D28. Before discussion geometric/probaiblistic processors, the notion of representing possible values of a variable needs to be present, but that's (again) discussed only later.

D29. Unclear to me: "probabilitic processors are similar to geometric processors"

D30. Ch. 5.3 uses the word "non-parametric" in a non-standard fashion. What is meant is "learned/pre-defined" instead of "paramteric/non-parameteric".

D31. At the end of Ch. 5, I wondered: so what? There is no discussion about the expressibity/quality/cost of these methods.

D32. For discustions, the theorem of Ren et al should be paraphrased to be self-contained. Also, what is M? (Defined in Ren et al., but not here.)

D33. Footnote 7 is incomplete.

D34. In principle, regular path queries (i.e., a single relation type) can be handled by all methods by adding a relation R+ for the transitive closure of R.

D35. Fig 19 has the right level of detail/formalism required for query types.

D36. It's stated that neural methods "assume that the underlying graph is incomplete and queries may have some missing answers". I am not sure I'd agree: It appears that quite some of these methods are trained with CWA, but have limited capacity and hence can generalize.

D37. Fig 19/20/21 can be merged.

D38. The common term for "zero projected variabels" is Boolean queries. Also, at least in principle, all other quries can be reduced to Boolean queries by "trying" all possible answers.

D39. Ch. 7 introduces a new notation for answers. Why not use this before?

D40. Training comes very hort, just a paragraph. Isn't there more to say? Are there differences in training methods other than loss (e.g., query sampling stategies, negative sampling) and what's the general cost?

D41. Inductive (superset) / inductive (disjoint) is termed semi-inductive / fully-inductive in the literature.

D42. Tab 15 is a useful overview of datasets. An overview like this for methods would be great (see comment D18).

D43. Tab 16 appears incomplete. For example, there are methods to handle unseen entities in KGQA/single-hop methods.

D44. The application section is disappointing. The methods are claerly not used so far. That's ok, but: why not? what's missing?

D45. Likewise, the summary/conclusion falls short. It list mainly features that would be nice to have, but does not work out / ask to address key weaknesses of current methods.

---

> ### Author Response · Authors · 2024-08-23
> **Response**
>
> Thank you for the detailed comments. We uploaded a revision addressing the weaknesses and comments. In particular:
>
> We moved a significant portion of theory in Section 2 to the Appendix only keeping the most important definitions in the main part
> We added Appendix B linking a tangentially related KGQA in the NLP area. Generally, we deem KGQA as a different task in the different subfield where the queries are posed in the natural language and the graphs are assumed to be complete (no link prediction needed).
> Addressed a lot of your smaller comments, thanks for the suggestions
>
> In this survey, we strived for a general overview of CLQA theory, methods, and datasets without going into the details of each method – mostly due to the size concerns: the manuscript is already 50+ pages of content.
>
>
> Comments on issues not addressed by the changes mentioned above:
>
> D1. Addressed in the introduction
>
> D2 - D3. We moved a significant portion of Section 2 in the appendix. Indeed, there are not that many works on hypergraph KGs, but we believe that hyper-relational KGs are of higher importance - at least because (1) Wikidata uses this representation formalism with main facts and qualifiers; (2) the upcoming standards RDF-star and SPARQL-star will increase the adoption of hyper-relational KGs in industry
>
> D4/ D6  We moved much of the details to the appendix with the intention that the reader gets now less confused by them. Those interested can then still find them there.
>
> D5.  This was a mistake indeed.
> $\texttt{Var}_Q = \gE' \cap \texttt{Var}$ should have been   $\texttt{Var}_Q = ( \gE' \cup \gR' ) \cap \texttt{Var}$ instead.
>
> D7. This is a tricky point, indeed. The answer semantics are indeed according to a closed world assumption. In a way, we expect the system to take a graph with unknown parts, thus according to OWA, and coerce it into making a decision about the truthfulness of the unknown, in the context of the query, and then provide the answers. The query could also provide (or rather assert) information; it could include statements which are not in the knowledge graph.
>
> D8. We elaborate on visible (training) and complete graphs in Section 5.2 (Processors)
>
> D10. SPARQL is a standard query language for KGs and we used it more for mapping to practical cases. FOL, on the other hand, is more for the definitions and mathematic rigor.
>
> D15. Added more visual separation to the figure and caption, thanks for the suggestion.
>
> D16. Shallow encoders do not explicitly encode the graph structure and operate on directly on vectors corresponding to entities and relations. Technically, shallow encoders are a subset of transductive encoders but due to the sheer volume of related works we put them into a separate category.
>
> D17. The output for all CLQA methods is a scalar distribution over all entities that indicates the probability of an entity to be a correct answer (Section 5.3)
>
> D18/D42. We think it is more of a style preference. In the suggested way, you would get very sparse checkmarks as almost nothing belongs in multiple categories.
>
> D20. In simpler words, shallow methods need much longer time to encode the graph structure in the representations. In transductive encoders GNNs provide this inductive bias from the start.
>
> D25. DNF conversion has been introduced in the definitions section (prev. 2.4, now in the Appendix)
>
> D26. This refers to element-wise vector normalization using the L2 norm: $v_i / ||v|| $
>
> D27. We cite these already in the text and in the other table. Adding these citations here again makes the table really crowded. We could make the names clickable and make them navigate to the relevant citations, but that worked rather confusingly.
>
>
> D29. Reworded in the paper

---

> > ### Author Response · Authors · 2024-08-23
> > **Response cont.**
> >
> > D32. We removed the use of $M$, it was not necessary in this context. The substance of the Theorem 1 from Ren et al is summarized in the next sentence after we mention it: “The theorem proves that we need the VC dimension of the function class of the distance function to be around the number of entities on the graph. ”
> >
> > D33. We removed the footnote as it was not adding more clarity to the paragraph.
> >
> > D34. In principle, regular path queries (i.e., a single relation type) can be handled by all methods by adding a relation R+ for the transitive closure of R.
> >
> > We might be misinterpreting the comment. We agree one could attempt to train the existing systems with a graph that contains the transitive closure for the known parts of the graph. However, none of the papers have demonstrated that so far.
> >
> > D37. We looked into the suggestion, but doing this causes a large gap between the figures and the place where they would be explained.
> >
> > D38. Yes, we spell it explicitly in the first paragraph of the Zero-projected Variables subsection that those queries are Boolean and are equivalent to ASK in SPARQL.
> >
> > D41. We have added this terminology, we have encountered it in the link prediction literature, but not in the papers on CLQA.
> >
> > D43. We removed Table 16 as it was overlapping in content with Table 15.
> >
> > D45. We substantially revised the Future Work section and elaborated on the open challenges.

---

### Review · Reviewer_9uMg · 2024-07-08

**Summary Of Contributions:**

This paper provides a comprehensive survey on complex logical query answering (CLQA), a task of graph machine learning that goes beyond one-hop link prediction to multi-hop logical reasoning over large-scale and potentially incomplete graphs. This paper discusses literature in this area from different perspectives, including graph formalism, modeling focuses, types of queries, data and metrics, and applications. The final section points out several directions for future research.

**Audience:**

Yes

**Broader Impact Concerns:**

No concerns for broad impact.

**Claims And Evidence:**

Yes

**Requested Changes:**

I would recommend that the authors
- discuss multi-hop reasoning work in NLP and their relations / differences with the main focus of this survey
- be more specific and provide technical insights for meaningful future directions
- fix their grammatical errors

**Strengths And Weaknesses:**

Strengths

The paper is comprehensive from graph ML perspective. It discusses literature on different types of graphs, different modeling focuses (e.g., encoding, decoding), different types of queries, and different benchmarks and metrics. The papers discussed in the survey are up-to-date.

The paper is generally well-written. The technical discussion is often accompanied with clear visualization.

Weaknesses

Multi-hop reasoning is also a popular and long-standing (not "emerging") topic in NLP. But this connection is not discussed. Arguably, the "graphs" in NLP are largely unstructured or less-structured (e.g., wikipedia articles connected by hyperlinks or lexical matches), so it is not at the core of this survey area, but it is highly related.

The discussion of future research directions are vague and generic. It merely lists some possible tasks / problems that past work has not covered yet, without providing insights about why it is not covered, or why it is a meaningful future direction. E.g., one future direction is to consider "hypergraphs", but section 2.1 implies that this formalism is restricted (and thus not very useful?), making me wonder if this direction is actually meaningful. This concern applies to other items in this section.

The writing has some grammatical errors. E.g., in page-5, "This means that is a triple is used in two places"...

---

> ### Author Response · Authors · 2024-08-23
> **Response**
>
> Thank you for the comments and suggestions. We are delighted to see the work is deemed comprehensive from the graph ML perspective highlighting writing and visualization quality.
>
> In the newly uploaded revision, we addressed the raised weaknesses:
>
> > Multi-hop reasoning is also a popular and long-standing topic in NLP. But this connection is not discussed.
>
> We elaborated on the similarities and differences between CLQA and KGQA in Appendix B, and highlight the integration of these two areas of research as a future research direction.
>
> > The discussion of future research directions are vague and generic.
>
> We expanded on the future work directions in the relevant section.
>
> > Typos
>
> We fixed the remaining typos, thanks for the pointers.
>
> Please let us know if you have any further concerns.

---

> > ### Comment · Reviewer_9uMg · 2024-10-29
> >
> > Thank you for the improvements.

---

### Review · Reviewer_qV59 · 2024-08-17

**Summary Of Contributions:**

This work presents a comprehensive survey of query reasoning methods over graph databases. The survey delineates existing works (~50 papers) along three main axes: Graphs (e.g., underlying structure), Modeling (e.g., query/graph encoders, decoding strategies), and Queries (e.g., operators' expressiveness, patterns complexity). The paper also touches on the evaluation metrics commonly used and ends on what are the open challenges for each of presented axes.

**Audience:**

Yes

**Broader Impact Concerns:**

I don't see any particular concerns on the ethical implications of this survey paper.

**Claims And Evidence:**

Yes

**Requested Changes:**

Irrespective of the highlighted weakness, I like the current version of the submission. Here is two minor changes the authors might wish to consider:
- I'm not sure I understand the purpose of Table 16. It seems to be all the information are already present in Table 15?
- In section 2, binary sets are defined according to other sets instead of their cardinality. Maybe it is a valid notation that I'm unaware of? For instance,
  - p.5: $\mathfrak{Q}=2^{(\mathcal{R}\times \mathcal{E})}$ vs. $\mathfrak{Q}=2^{|\mathcal{R}\times \mathcal{E}|}$

Also, here are some typos:
- p.5: "which gives is the same" -> "which gives the same"
- p.5: "This means that is a triple" -> "This means that a triple"
- p.8: Just before Definition 2.10, "to FOL fragments". FOL is only defined two paragraphs later.
- p.9: "restricting CQ and on CQ_neg" ->  "... and CQ_neg" ?
- p.12: "the identity element for T is 1, i.e., T(x,1)=1" -> T(x,1)=x ?
- p.12: "on can verify" -> "one can verify"
- p.21: "... Section 7.3) However..." <- missing point.
- p.34, Footnote 7: Missing end of sentence.
- p.35: "strong closture" -> closure
- p.37: Unbalanced parenthesis in "the clause COUNT (?book) as ?n)"
- p.45: "from the Inferece" -> Inference
- p.48: "open challenges Adhering" <- missing point
- p.49: Unbalanced parenthesis in "... negations (UCQ and UCQ_neg and aiming ..."

**Strengths And Weaknesses:**

**Strengths**
- I can see this work as a go to survey paper for neural graph research. The literature review is extensive and I was pleased to see that the authors kept it up to date from their previous submission.

- The paper is well written and structured. I really enjoyed seeing the delineation of the related work given the different axes. I also appreciated the authors' effort to provide a comprehensive background section with definitions for neural graph and graph query answering.

- The open challenges section at the end was clear and a valuable to any researchers wanting to jump into this field.

**Weaknesses**
- I would have appreciated to see some concrete performance results to see which methods are better than others in the different settings. For instance, if I have a particular setting in mind, I wouldn't know which methods to consider in the first place. That said, I understand that this is a survey paper, and the authors might have chosen to focus on the literature review. I can only hope for a follow up paper that would provide such results (maybe a benchmark? that would be a great contribution).

---

> ### Author Response · Authors · 2024-08-23
> **Response**
>
> Thank you for appreciating our work highlighting the clarity, structure, and the go-to nature for the CLQA field.
>
>
> > I would have appreciated to see some concrete performance results
>
>
> As you correctly pointed, the focus of this manuscript is to provide some formal foundations and a literature review of CLQA. Experimental evaluation and benchmarking of dozens of methods is indeed the subject of our future work that deserves a separate paper
>
>
> > I'm not sure I understand the purpose of Table 16. It seems that all the information are already present in Table 15?
>
>
> You are right - originally, for Table 16, we wanted to highlight the inductive setup and datasets scarcity for it, but it indeed largely overlaps with Table 15. We removed Table 16 in the revision and link the text to Table 15 now.
>
>
> > In section 2, binary sets are defined according to other sets instead of their cardinality.
>
>
> This notation denotes a powerset - we thought it is a rather standard notation but we clarified explicitly in the text that it is a powerset, as well as expanded on the meaning of $2^{(\gR \times \gE)}$.
>
>
> Thanks for noticing other typos, we fixed all of them in the revision.

---

> > ### Comment · Reviewer_qV59 · 2024-08-23
> > **Ack response**
> >
> > Thank you for the clarification about the powerset notation. Great survey.

---

### Decision · Action_Editor_KBB5 · 2024-10-30

**Recommendation:** Accept as is

**Comment:**

- The paper is comprehensive and timely, capturing a large amount of recent work in the field.

- It provides a useful taxonomy and many overview tables.

- It is well-written and structured, with clear visualizations.

- It is considered a go-to survey for neural graph research

All the reviewers are positive about the paper. I recommend acceptance.

**Audience:**

Yes

**Claims And Evidence:**

Yes